# Neural Collapse by Design: Learning Class Prototypes on the Hypersphere

## Abstract

Neural Collapse (NC) describes the global optimum of supervised learning, yet standard cross-entropy (CE) training rarely attains its geometry in practice. This is due to unconstrained radial degrees of freedom: cross-entropy is invariant to joint rescaling of features and weights, leaving radial directions underconstrained thus preventing convergence to a unique geometry. We show that constraining optimization to the unit hypersphere removes this degeneracy and reveals a unifying view of normalized softmax classifier learning (CL) and supervised contrastive learning (SCL) as the same prototype-contrast principle: both optimize angular similarity to class prototypes, using explicit learned weights for normalized softmax and implicit class means for SCL. Despite this shared foundation, existing objectives suffer from small effective negative sets and interference between positive and negative terms, which slows convergence to NC. We address these issues with two objectives: NTCE, which contrasts class prototypes against all batch instances to expand the negative set from K classes to M samples; and NONL, which normalizes only over negatives to decouple intra-class alignment from inter-class repulsion. Theoretically, we prove that SCL already learns an optimal prototype classifier under NC, eliminating the need for post-hoc typically hours-scale linear probing. Empirically, across four benchmarks including ImageNet-1K, our methods surpass CE accuracy, reach $\geq 95\%$ on NC metrics, and match NC structure with substantially fewer iterations. Moreover, SCL with class-mean prototypes matches linear-probing accuracy while requiring no training. These results reframe supervised learning as prototype-based classification on the hypersphere, closing the theory–practice gap while simplifying training and accelerating convergence.

## 1 Introduction

Despite theoretical proofs that Neural Collapse (NC) is the global optimum of supervised learning objectives (Lu & Steinerberger, 2022; Zhou et al., 2022a; Graf et al., 2021), standard training with cross-entropy rarely achieves this configuration in practice. This failure is particularly striking because *NC delivers precisely the properties we seek*: when neural networks do approach this geometric configuration, where within-class representations collapse to their means, class means form an equiangular tight frame (ETF), and classifier weights align with these prototypes, they demonstrate improved generalization (Papyan et al., 2020; Bartlett et al., 2017; Neyshabur et al., 2018), adversarial robustness (Fawzi et al., 2016; Ding et al., 2020), enhanced transfer learning (Kornblith et al., 2019; Khosla et al., 2020), and converge toward max-margin classifiers (Soudry et al., 2018) with stronger robustness guarantees (Hein & Andriushchenko, 2017). *If NC is provably optimal and empirically beneficial, why does standard training consistently fail to achieve it?*

We identify the core issue as *unconstrained radial degrees of freedom*. Cross-entropy optimization allows features and weights to be jointly rescaled without changing predictions (Soudry et al., 2018). This leaves radial directions underconstrained, preventing convergence to a unique geometry. While explicit regularization of features, weights, and biases may theoretically resolve this (Zhu et al., 2021), it introduces multiple hyperparameters that complicate practical implementation. A more principled solution is to eliminate radial freedom entirely by constraining optimization to the unit hypersphere, where NC becomes the *unique* global optimum (Yaras et al., 2022).

This geometric perspective reveals a surprising *unity* between two learning paradigms traditionally viewed as fundamentally different. Classifier learning (CL) with normalized softmax (Wang et al., 2017) has been understood as directly learning decision boundaries through weight vectors, while supervised contrastive learning (SCL) (Khosla et al., 2020) has been viewed as learning representations through instance-to-instance comparisons followed by a separate classifier training phase. We show both are actually *prototype-contrast methods on the hypersphere*: normalized softmax optimizes angular similarity between normalized features and explicit class weight vectors serving as prototypes, while SCL optimizes angular similarity among normalized instances using implicit class-mean embeddings as prototypes.

Despite this shared geometric foundation, these methods inherit computational limitations that prevent efficient NC convergence. CL suffers from a small effective negative set (denominator contrasts against only $K$ class weights) (He et al., 2020), while both paradigms couple positive and negative similarity terms through shared normalization (Yeh et al., 2022), creating interference that slows convergence to optimal geometry. These limitations suggest that achieving NC requires not just hyperspherical constraints but also algorithmic innovations in *how prototypes are contrasted*.

Building on the insight that both paradigms are prototype-based but computationally limited, we make four five contributions:

1. We **unify normalized softmax and SCL** under a single geometric framework, revealing both as *prototype-contrast methods on the unit hypersphere* that differ only in whether prototypes are explicit (learned weights) or implicit (class means). This framework explains why both can achieve NC while standard cross-entropy cannot.

2. We propose **two supervised objectives** that overcome existing computational limitations. **NTCE** (Normalized Temperature-scaled Cross Entropy) increases the effective number of negatives from $K$ classes to $M$ batch samples by contrasting the class prototype against all instances in the batch, strengthening inter-class separation. **NONL** (Negatives-Only Normalization Loss) eliminates interference between intra-class alignment and inter-class repulsion by normalizing only over negatives, accelerating NC convergence.

3. We prove that *SCL already learns an optimal classifier during pretraining*, **eliminating the need for linear probing**. The class-mean embeddings learned by SCL form an ETF-aligned prototype classifier under NC, implementing the self-duality condition by construction and yielding equivalent accuracy without incurring the computational cost of post-training probing.

4. We validate our approach across four benchmarks including ImageNet-1K. NTCE and NONL achieve $\geq 95\%$ *on NC metrics* while *surpassing standard cross-entropy accuracy*, and match cross-entropy's NC metrics with *substantially fewer training iterations*. Our prototype classifier maintains SCL's accuracy while eliminating hours of linear probing computation, a significant *practical saving for large-scale deployments*.

5. We empirically demonstrate that the representations learned by our objectives translate into **practical benefits**, yielding improved performance on *transfer learning* (e.g., +5.5% mean relative improvement), *long-tailed classification* (up to +8.7% relative improvement), and *robustness* (lower mCE).

These results suggest a **fundamental shift** in how supervised learning should be understood: not as unconstrained optimization in Euclidean space, but as *prototype-based classification on the hypersphere*. By making this geometry explicit, we close the theory–practice gap, simplify training, accelerate convergence, and yield interpretable models that provably realize their optimal NC structure. The *practical impact is substantial*: faster training, elimination of extra compute phases, and models that reach the theoretical optimum. The *theoretical insight* provides a principled foundation for future advances in supervised learning.

## 2 RELATED WORK

**Neural Collapse.** Neural Collapse (NC) describes a limiting geometry in which within-class features collapse to their means (NC1), class means form a centered simplex ETF (NC2), classifier weights align with the means (NC3), and biases collapse (NC4) (Papyan et al., 2020). Variants of

this structure characterize global minimizers for several objectives and modeling assumptions, including MSE (Han et al., 2022; Zhou et al., 2022a), cross-entropy (CE) (Lu & Steinerberger, 2022), supervised contrastive learning (SCL) (Graf et al., 2021), and CE variants such as label smoothing and focal loss (Zhou et al., 2022b). In finite training, however, standard CE with weight decay often fails to realize the optimal geometry: the loss is *scale-noncoercive* and can be driven toward zero by inflating logit magnitudes without improving angular structure (Albert & Anderson, 1984; Soudry et al., 2018). Class imbalance further distorts the ETF and slows convergence (Thrampoulidis et al., 2022; Hong & Ling, 2024); free bias terms obstruct NC4 and can exacerbate miscalibration unless controlled (e.g., logit adjustment) (Menon et al., 2021). While simultaneously penalizing features, weights, and biases can restore coercivity and yield NC in principle (Zhu et al., 2021; Zhou et al., 2022a), tuning multiple regularizers is brittle. *We show that contrasting instances against class prototypes on the hypersphere operationalizes NC in practice.*

**Learning on the hypersphere.** Constraining radial freedom is a principled route to NC. When both features and classifier lie on the unit hypersphere, CE over the product of spheres exhibits a benign strict-saddle landscape whose minima realize perfect NC (Yaras et al., 2022). Related evidence appears in contrastive objectives: SCL yields within-class collapse and simplex class means (Graf et al., 2021), while in self-supervised contrastive learning batch-level optima form a simplex ETF (Koromilas et al., 2024). A long line of face-recognition work, including SphereFace, CosFace, ArcFace, and NormFace (Liu et al., 2017; Wang et al., 2018; Deng et al., 2019; Wang et al., 2017), operationalizes direction-only discrimination by using angular/cosine margins. *We unify these appraches by showing that both normalized softmax and SCL perform prototype contrast on the hypersphere.* Building on this bridge, we extend normalized softmax with NTCE/NONL to import desirable properties.

**Prototype-based classification and ETF classifiers.** Prototype methods classify via distances to learned representatives (Snell et al., 2017). Motivated by NC, several works fix or guide the classifier toward ETF-like prototypes and learn only the encoder, for example by (i) fixing a simplex ETF head and training the backbone (ETF+DR) (Yang et al., 2022), (ii) using hyperspherical prototype networks (Mettes et al., 2019), or (iii) constructing equiangular basis vectors (EBVs) (Shen et al., 2023). Other approaches enforce (non-negative) orthogonality (Kim & Kim, 2024) or guide the classifier toward the nearest ETF via a Riemannian inner optimization (Markou et al., 2024). Recently NC structure has been exploited in a teacher–student setting (Zhang et al., 2025): given a trained teacher that already exhibits NC, they compute *teacher* class centroids and use them as an NC3-inspired classifier for the *student*. *Our perspective is that CL and SCL already operate with prototypes: we modify the objectives to realize NC in practice, and we show that SCL's class-mean prototypes form an effective classifier, making linear probing unnecessary.*

## 3 PRELIMINARIES

**Notation.** Scalars are denoted by lowercase letters $u$, vectors by lowercase bold letters $\boldsymbol{u}$, and matrices by uppercase bold letters $\boldsymbol{U}$. Sets are represented by uppercase caligraphic letters $\mathcal{U}$. Individual elements are accessed using subscript notation: $u_i$ for the $i$-th element of vector $\boldsymbol{u}$ and $U_{i,j}$ for the element at row $i$ and column $j$ of matrix $\boldsymbol{U}$. To denote vertical (row-wise) concatenation of matrices $\mathbf{X}$ and $\mathbf{Y}$, we use $[\mathbf{X}; \mathbf{Y}]$. We denote normalized vectors with $\hat{\boldsymbol{u}}_j = \boldsymbol{u}_j / \|\boldsymbol{u}_j\|$.

### 3.1 LEARNING PARADIGMS

**Classifier Learning with Cross-Entropy.** The cross-entropy loss is the standard Classifier Learning (CL) objective, optimizing representations and classifier weights simultaneously. An encoder $f_{\boldsymbol{\theta}} : \mathcal{X} \to \mathcal{Z}$, parameterized by $\boldsymbol{\theta} \in \Theta$, maps an input $\mathbf{x} \in \mathcal{X}$ to its representation $\mathbf{z} = f_{\boldsymbol{\theta}}(\mathbf{x}) \in \mathcal{Z}$. For a $K$-class task, $y_i$ denotes the class assignment of sample $\mathbf{x}_i$. A linear classifier is placed on top of the encoder, with weight matrix $\mathbf{W} \in \mathbb{R}^{K \times h}$ and bias $\mathbf{b} \in \mathbb{R}^K$, where $h$ is the embedding dimension. For a mini-batch of $M$ samples with $\{\mathbf{z}_i\}_{i=1}^M$, the cross-entropy loss is defined as

$$\mathcal{L}_{\mathrm{CE}}(\mathbf{Z}, \mathbf{W}) = \frac{1}{M} \sum_{i=1}^M - \log \left( \frac{e^{\mathbf{z}_i^\top \mathbf{w}_{y_i} + b_{y_i}}}{\sum_{j=1}^K e^{\mathbf{z}_i^\top \mathbf{w}_j + b_j}} \right), \tag{1}$$

where $\mathbf{w}_j$ denotes the $j$-th row of $\mathbf{W}$ and $b_j$ the $j$-th component of $\mathbf{b}$.

**Supervised Contrastive Learning.** *Supervised Contrastive Learning (SCL)* takes a seemingly different direction: it learns representations by exploiting similarities between instances to learn class-invariant representations. Building on our notation, the contrastive framework augments the encoder $f_{\boldsymbol{\theta}} : \mathcal{X} \to \mathcal{Z}$ with a projection head $g_{\boldsymbol{\phi}} : \mathcal{Z} \to \mathcal{U}$, parameterized by $\boldsymbol{\phi} \in \Phi$, which maps representations onto the unit hypersphere, $\mathcal{U} = \mathbb{S}^{d-1} = \{\boldsymbol{u} \in \mathbb{R}^d \mid \|\boldsymbol{u}\| = 1\}$. We denote the projected representations as $\boldsymbol{u}, \boldsymbol{v} \in \mathcal{U}$, where $\boldsymbol{u}_i$ comes from instance $\boldsymbol{x}_i$ and $\boldsymbol{v}_i$ from its alternative view produced via augmentation, a typical process in contrastive learning.

For SCL the objective is to pull together positive pairs while pushing apart negative pairs in the projection space. Typically alternative views of the same data point that originate from augmentation are considered as new data points, *i.e.* $\boldsymbol{A} = [\boldsymbol{U}; \boldsymbol{V}]$, and the supervised contrastive loss becomes:

$$\mathcal{L}_{\text{SCL}}(\boldsymbol{A}) = \frac{1}{2M} \sum_{i=1}^{2M} -\frac{1}{|\mathcal{C}(i)|} \sum_{l \in \mathcal{C}(i)} \log \left( \frac{e^{\boldsymbol{a}_i^\top \boldsymbol{a}_l / \tau}}{\sum_{\substack{j=1 \\ j \neq i}}^{2M} e^{\boldsymbol{a}_i^\top \boldsymbol{a}_j / \tau}} \right), \qquad (2)$$

where $\mathcal{C}(i)$ denotes the set of indices corresponding to positive examples sharing the same class as $\boldsymbol{x}_i$ and $\tau > 0$ is a temperature parameter that controls the concentration of the distribution.

A crucial distinction emerges post-training: while learning with cross-entropy directly produces a classifier, contrastive learning requires an additional step. After optimizing Equation (2), the projection head is discarded and a linear classifier $\boldsymbol{W}, \boldsymbol{b}$ is trained on the frozen encoder representations $\boldsymbol{z}$ using Equation (1), a process known as **linear probing**.

### 3.2 Neural Collapse (NC).

*Neural Collapse* Papyan et al. (2020)is the late-training regime (on balanced data) where last-layer features and the linear classifier converge to a highly structured limit. Let $\boldsymbol{z}_i = f(\boldsymbol{x}_i) \in \mathbb{R}^h$, class means $\boldsymbol{\mu}_c = \frac{1}{n_c} \sum_{i : y_i = c} \boldsymbol{z}_i$, weights $\boldsymbol{w}_c$, and bias $\boldsymbol{b}$. NC holds when, up to common scalings:

(NC1) **Within-class collapse:** $\boldsymbol{z}_i = \boldsymbol{\mu}_{y_i}$ for all $i$.

(NC2) **Simplex ETF of class means:** the centered means $\tilde{\boldsymbol{\mu}}_c = \boldsymbol{\mu}_c - \frac{1}{K} \sum_{k=1}^{K} \boldsymbol{\mu}_k$ have equal norms and equal pairwise angles so the means span a centered $(K-1)$-simplex ETF.

(NC3) **Alignment of Class Representation and Classifier:** classifier columns align with the class means, $\boldsymbol{w}_c \parallel \boldsymbol{\mu}_c$ (there exists $\gamma > 0$ with $\boldsymbol{w}_c = \gamma \boldsymbol{\mu}_c$).

(NC4) **Bias collapse:** $\boldsymbol{b} = \beta \mathbf{1}$ for some scalar $\beta$.

Under NC, the decision rule reduces to nearest-class-mean classification. We assume balanced classes and $h \geq K - 1$ so a centered simplex ETF is feasible (Lu & Steinerberger, 2022).

**Practical Challenges in reaching Neural Collapse** Neural Collapse (NC) is now well documented in deep nets (Papyan et al., 2020) and characterizes global minima of balanced cross-entropy (Lu & Steinerberger, 2022). However standard pipelines does not enforce it in practice. For the typical paradigm of cross-entropy and classifier weight decay, the objective admits an *unbounded rescaling direction*: shrinking the classifier while amplifying features leaves logits unchanged, reduces the penalty, and drives the loss toward zero without achieving NC (Soudry et al., 2018; Albert & Anderson, 1984). It is shown by Zhu et al. (2021) that a well-posed objective arises when all radial degrees of freedom are constrained by penalizing weights, features, and biases simultaneously (Zhu et al., 2021). However this is practically brittle due to multiple regularizers to tune.

Supervised contrastive training on the other hand can drive representations toward NC geometry (Graf et al., 2021). However, the subsequent *linear probing* step typically fits a softmax classifier with cross-entropy on *frozen* features, allowing free weight magnitudes and biases. This reintroduces the same scale and bias pathologies as cross-entropy even when training has already reached an NC.

## 4 Supervised Learning on the Hypersphere

In this section we present a common view-point bridging classifier learning and contrastive learning to accelerate neural collapse. Our approach leverages similarity-based optimization while elimi-

nating radial degrees of freedom by constraining both feature and classifier norms to the hypersphere. This constraint transforms the optimization landscape into a benign geometry where all critical points become global optima (Yaras et al., 2022), enabling direct convergence to NC.

### 4.1 REVISITING CROSS ENTROPY: CONTRASTING CLASS PROTOTYPES TO INSTANCES

The weight matrix of the final linear classifier in CL methods can be expressed as $\boldsymbol{W} = [\boldsymbol{w}_1; \boldsymbol{w}_2; \ldots; \boldsymbol{w}_K] \in \mathbb{R}^{K \times h}$, where each $\boldsymbol{w}_c$ represents a learnable class prototype. This formulation reveals an important insight: we can treat the classifier weights as *learnable prototypes* that evolve through gradient descent to capture class-specific geometric structures. Building on this we design objectives that leverage such prototypes to help arrive at the optimal NC geometry.

**Normalized Softmax Losses.** Standard cross-entropy and contrastive learning represent two seemingly distinct paradigms: the former discriminates through learned magnitudes and biases in unconstrained space, while the latter operates purely on angular similarities on the hypersphere. This fundamental difference leads to a critical inefficiency: while both methods theoretically converge to neural collapse configurations, cross-entropy introduces unnecessary radial degrees of freedom that slow convergence to this optimal geometry (Yaras et al., 2022; Zhu et al., 2021).

Normalized softmax losses resolve this inefficiency by reformulating cross-entropy as a pure geometric objective. NormFace (Wang et al., 2017), a prominent example, achieves this through three coordinated modifications: (i) eliminating biases that merely translate decision boundaries without encoding semantic structure, (ii) projecting representations onto the hypersphere to focus exclusively on angular geometry, and (iii) introducing temperature scaling to control concentration of the softmax distribution. Formally, with $\boldsymbol{u}_i = \boldsymbol{z}_i / \|\boldsymbol{z}_i\|_2$ as the normalized representation and $\hat{\boldsymbol{w}}_j = \boldsymbol{w}_j / \|\boldsymbol{w}_j\|_2$ as the normalized classifier weight for class $j$, NormFace minimizes:

$$L_{\text{NormFace}}(\boldsymbol{U}, \boldsymbol{W}) = -\frac{1}{M} \sum_{i=1}^{M} \log \left( \frac{e^{\boldsymbol{u}_i^\top \hat{\boldsymbol{w}}_{y_i} / \tau}}{\sum_{j=1}^{K} e^{\boldsymbol{u}_i^\top \hat{\boldsymbol{w}}_j / \tau}} \right). \tag{3}$$

This reformulation transforms classification into contrastive learning between data instances and learnable class prototypes while maintaining cross-entropy's computational efficiency.

**Normalized Temperature-scaled Cross Entropy (NTCE)**

When utilizing NormFace to view CL from a contrastive learning perspective we end up with an inherent limitation of cross entropy: the number of negatives in the objective is limited to $K$, the number of class prototypes. It is very well investigated that contrastive objectives need very large numbers of negatives in order to converge (He et al., 2020). This is mostly due to the fact that fewer negatives provide a worse estimate to the expectation of the actual contrastive objective (Koromilas et al., 2024).

By inverting the contrastive direction from instance-to-class to class-to-instance discrimination we address this limitation through the Normalized Temperature-scaled Cross Entropy (NTCE). This modification fundamentally alters the learning dynamics: rather than each instance contrasting against $K$ class prototypes, each class prototype now contrasts against $M$ batch representations.

The key insight underlying NTCE is that *class prototypes themselves can serve as anchors* in the contrastive formulation. By anchoring on the class weight vector corresponding to each instance's ground-truth label and contrasting it against all batch representations, we dramatically expand the negative sampling space. Formally, NTCE takes the form:

$$L_{\text{NTCE}}(\boldsymbol{U}, \boldsymbol{W}) = \frac{1}{M} \sum_{i=1}^{M} -\log \left( \frac{e^{\hat{\boldsymbol{w}}_{y_i}^\top \boldsymbol{u}_i / \tau}}{\sum_{j=1}^{M} e^{\hat{\boldsymbol{w}}_{y_i}^\top \boldsymbol{u}_j / \tau}} \right), \tag{4}$$

where $\hat{\boldsymbol{w}}_{y_i}$ serves as the anchor for instance $i$, and critically, the denominator sums over all $M$ instances in the batch rather than over $K$ classes.

**Negatives Only Normalization Loss.** NTCE adds enhanced negative sampling on top of NormFace to directly transfer the principles of contrastive learning to cross entropy training. However, it also brings a fundamental drawback of popular contrastive objectives that compromises its optimization dynamics. The denominator in Equation (4) indiscriminately aggregates all instances sharing the

same class anchor. That is the denominator, also known as the uniformity term, is optimized when instances of the same class have maximum distance (Wang & Isola, 2020), which contradicts the optimality of the numerator (alignment terms). More specifically, positive pairs explicitly appear as negative samples in the normalization term, generating gradients that actively repel instances from their own class prototype. When instance $i$ and instance $j$ share class $y_i = y_j$, the term $e^{\hat{\boldsymbol{w}}_{y_i}^\top \boldsymbol{u}_j / \tau}$ in the denominator produces gradients that decrease $\hat{\boldsymbol{w}}_{y_i}^\top \boldsymbol{u}_j$, directly opposing the alignment objective. This is a known behavior that is called *alignment-uniformity coupling* (Yeh et al., 2022).

In order to resolve this conflict we introduce the Negatives-Only Normalization Loss (NONL), which explicitly excludes same-class instances from the denominator:

$$L_{\text{NONL}}(\boldsymbol{U}, \boldsymbol{W}) = \frac{1}{M} \sum_{i=1}^{M} - \log \left( \frac{e^{\hat{\boldsymbol{w}}_{y_i}^\top \boldsymbol{u}_i / \tau}}{\sum_{\substack{j=1 \\ j \notin \mathcal{C}(i)}}^{M} e^{\hat{\boldsymbol{w}}_{y_i}^\top \boldsymbol{u}_j / \tau}} \right). \tag{5}$$

**NC optimality of normalized objectives.** In in Theorem 4.1 we show that, in the balanced UFM/LPM setting (Tirer & Bruna, 2022; Yaras et al., 2022), the three normalized losses (Norm-Face, NTCE, and NONL) are globally optimized by Neural Collapse (NC) geometry. The proof is deferred to Section A.1 and proceeds via a unified three-step reduction for NTCE and NONL: (i) we reduce the sample-level losses to class-mean based objectives using Jensen's inequality with explicit tightness conditions, (ii) we show these class-level objectives are contrastive losses whose global minimizers exhibit a centered simplex ETF structure where class-means align with the classifier, and (iii) we invoke the tightness conditions to prove any global minimizer must collapse each class to a single feature vector, recovering NC at the sample level. For NormFace, we establish an exact equivalence with the constrained UFM formulation of Yaras et al. (2022), directly invoking their Neural Collapse theorem.

**Theorem 4.1** (Neural Collapse optimality of normalized losses)**.** *In the balanced UFM/LPM setting above with $d \geq K$, every global minimizer of $\mathcal{L}_{\text{NF}}$, $\mathcal{L}_{\text{NTCE}}$, and $\mathcal{L}_{\text{NONL}}$ satisfies NC1–NC3 (within-class collapse, simplex ETF class means, and classifier–feature alignment), up to a global rotation and permutation of class labels.*

## 4.2 REVISITING SUPERVISED CONTRASTIVE LEARNING: CONTRASTING MEAN-CLASS PROTOTYPES TO INSTANCES

**SCL implicitly learns prototype classifiers.** We follow Equation (2) to treat alternative views produced by data augmentation as distinct samples, i.e., $\boldsymbol{A} = [\boldsymbol{U}; \boldsymbol{V}]$. Let $\mathcal{B}_c = \{j \in [2M] : y_j = c\}$ denote the within-batch index set for class $c$, $n_c = |\mathcal{B}_c|$, and $\hat{\boldsymbol{\mu}}_c = \frac{1}{n_c} \sum_{j \in \mathcal{B}_c} \boldsymbol{a}_j$ the corresponding batch prototype (class mean). We define the *prototype loss*:

$$L_{\text{proto}}(\boldsymbol{A}) = -\frac{1}{2M} \sum_{i=1}^{2M} \log \left( \frac{e^{\boldsymbol{a}_i^\top \hat{\boldsymbol{\mu}}_{y_i} / \tau}}{-e^{\boldsymbol{a}_i^\top \hat{\boldsymbol{\mu}}_{y_i} / \tau} + \sum_{c=1}^{K} n_c \cdot e^{\boldsymbol{a}_i^\top \hat{\boldsymbol{\mu}}_c / \tau}} \right), \tag{6}$$

where the numerator encourages alignment with the correct class prototype, while the denominator includes both positive and negative prototypes weighted by their batch frequencies $n_c$. Theorem 4.2 connects the optima of this loss to the ones of SCL. The proof can be found in Section A.2.

**Theorem 4.2** (Equivalence of SCL and prototype–softmax minimizers)**.** *For unit-norm representations and balanced labels the supervised contrastive loss $L_{SCL}$ and the prototype loss $L_{proto}$ in Equation (6) share the same set of global minimizers (up to rotation and label permutation). In particular, at every global minimizer the representations exhibit in-class collapse and the class means form a centered simplex ETF.*

This result clarifies our understanding of SCL: rather than merely learning good representations for classification, *SCL directly optimizes for classifier-feature alignment* through its contrastive objective. The learned prototypes are not just byproducts but the optimal classifiers themselves.

**Connection to Classifier Learning.** The $n_c$ weighting in the denominator of Equation (6) captures the effect of utilizing multiple negative instances, matching the structure of Equation (4). When discarding the $n_c$ weights, this loss reduces to Equation (3), establishing a direct **correspondence**

Table 1: Performance comparison of learning paradigms and objectives across datasets. **Bold**: best within method family; **green**: overall best per dataset.

**(I) Classifier Learning Methods**

| Loss | CIFAR-10 | CIFAR-100 | ImageNet-100 | ImageNet-1K |
|---|---|---|---|---|
| CE | 94.6 | 72.1 | 84.4 | 75.4 |
| ETF + DR | 94.4 | 72.1 | 84.5 | 75.4 |
| NORMFACE | 94.8 | 72.4 | 84.4 | 76.4 |
| NTCE (ours) | 94.7 | 72.9 | 84.7 | **76.7** |
| NONL (ours) | **94.9** | **73.6** | **84.9** | 76.5 |

**(II) Supervised Contrastive Learning Methods**

| Classifier Learning | Loss | Forward Passes | CIFAR-10 | CIFAR-100 | ImageNet-100 | ImageNet-1K |
|---|---|---|---|---|---|---|
| LINEAR PROBING | SCL | $T \times N$ | **95.0** | **73.9** | 84.8 | **75.1** |
| NORMALIZED LINEAR PROBING | SCL | $T \times N$ | 94.9 | 73.6 | 84.8 | **75.1** |
| FIXED PROTOTYPES | SCL | $N$ | **95.0** | **73.9** | **86.8** | **75.1** |

**between the prototype weights and class means**. Adding that the optimal solution of Equation (3) holds when $\boldsymbol{w}_c = \hat{\boldsymbol{\mu}}_c$ (Yaras et al., 2022) the connection becomes even more prevalent.

In other words, despite SCL converging to collapsed class representations forming an ETF, its optima can also be attained by contrasting instances to class-mean prototypes. This connects CL techniques to SCL, where the learnable classifier weights in the former are free parameters while in the latter they emerge implicitly from the learned representations.

**Why linear probing fails for SCL features.** In practice linear probing is used to train a classifier for the learned SCL representations. This approach introduces **unnecessary degrees of freedom** that disrupt the geometric optimality achieved by SCL. Specifically this process introduces: (i) **geometric mismatch**: SCL features live on the hypersphere with collapsed, ETF-structured class means. Linear probing operates in unconstrained Euclidean space, allowing weight rescaling and bias shifts that break classifier-feature alignment (Soudry et al., 2018). (ii) **Redundancy:** Our theorem shows SCL has already learned optimal classifier weights, *i.e.* the class prototypes themselves.

**Class-mean Prototypes inplace of Linear Probing.** We observe that class prototypes $\hat{\boldsymbol{\mu}}_c$ serve as natural classifier weights that satisfy NC3 (classifier-feature alignment) by construction. Rather than retrofitting a linear head to pre-collapsed features, we directly impose the NC-optimal classifier from the learned geometry. We *discard linear probing entirely* and set the classifier weights to the learned prototypes: $\boldsymbol{w}_c = \hat{\boldsymbol{\mu}}_c$. Doing so, we alleviate the need for an extra training phase, and we show empirically (Section 5) that this prototype-based classification matches linear probing performance.

## 5 EXPERIMENTS

In this section we empirically validate our methods against cross-entropy (CE), ETF + DR (Yang et al., 2022), and NormFace(Wang et al., 2017) for Classifier Learning paradigms and supervised contrastive learning (SCL), evaluating: (i) classification accuracy, (ii) proximity to neural collapse geometry, and (iii) NC convergence speed. Experiments are conducted on four standard datasets: *CIFAR10, CIFAR100, ImageNet-100, and ImageNet1K*, following common representation learning benchmarking practices (Khosla et al., 2020; Markou et al., 2024; Wang et al., 2021; Yeh et al., 2022). We use ResNet50 for ImageNet datasets and ResNet18 for CIFAR. Implementation details are provided in Section A.3.

### 5.1 CLASSIFICATION PERFORMANCE

**Classifier Learning Methods.** As can be inferred from Table 1(I), normalized losses *outperform cross-entropy* (CE) in all cases, while also our losses further outperform NormFace. NONL achieves the strongest gains on datasets with few (10) to medium (100) number of classes while it has the

second best score on ImageNet-1K. Here we have to note that ImageNet-1K, our objectives exhibit the typical behavior of contrastive-style objectives: they benefit from larger batch sizes, whereas using smaller batches leads to degraded performance due to insufficient in-batch negatives (see Section A.5).

**Supervised Contrastive Learning Methods.** The accuracy from three classifier learning strategies on SCL representations is presented in Table 1(II): (i) standard linear probing with learnable weights and bias, (ii) normalized linear probing using NormFace loss, and (iii) fixed prototypes computed as class-mean embeddings. Fixed prototypes match linear probing performance on 3 of 4 datasets, and mark a considerable +2.0% improvement on ImageNet-100 **requiring only $N$ forward passes versus** $T \times N$ for training-based methods, where $T$ is the number of epochs. Normalized linear probing achieves comparable accuracy to standard linear probing, validating that the *discriminative information in SCL features resides primarily in their angular structure* rather than magnitude or biases. These findings validate that angular structure alone suffices for discrimination in well-trained representations, enabling *training-free classification* in SCL via fixed prototypes that **eliminate huge computational costs** by discarding a, typically hours long, training phase.

## 5.2 QUANTIFYING NEURAL COLLAPSE

We quantify NC1–NC3 with complementary, condition-specific metrics; we omit NC4 (bias collapse) as our models enforce zero bias by design.

**Effective Rank (NC1, NC2).** For matrix $\mathbf{A}$ with singular values $\{\sigma_i\}$ the effective rank (Roy & Vetterli, 2007) is defined as $\mathrm{erank}(\mathbf{A}) = \exp\{-\sum_i p_i \log p_i\}$ where $p_i = \sigma_i / \sum_j \sigma_j$. We compute the intra and inter class effective ranks (Zhang et al., 2024) as: $\mathrm{erank}_{\mathrm{intra}} = \frac{1}{K} \sum_{c=1}^{K} \mathrm{erank}(\mathrm{Cov}[\mathbf{z}_i - \mu_c \mid y_i = c])$ and $\mathrm{erank}_{\mathrm{inter}} = \mathrm{erank}(\mathrm{Cov}[\mu_c - \mu_G])$, where Cov is the covariance matrix. These metrics quantify **NC1** (within-class variability collapse): $\mathrm{erank}_{\mathrm{intra}} \to 0$ indicates $\mathbf{z}_i \to \mu_{y_i}$, and **NC2** (ETF structure): Zhang et al. (2024) proved that when $\mathrm{erank}_{\mathrm{inter}} = K - 1$ the class means form a simplex with equal pairwise angles. We also report $\mathrm{erank}(\mathbf{W})$ to assess whether classifier weights approximate an equiangular tight frame (ETF).

**Alignment (NC3).** We quantify feature–classifier alignment by $\frac{1}{N} \sum_{i=1}^{N} \|\mathbf{z}_i - \mathbf{w}_{y_i}\|_2^2$ and also report instance-to-instance alignment to probe per-class collapse.

**Information Metrics (NC2, NC3).** For normalized Gram matrices $\mathbf{G}_W$ (weights), $\mathbf{G}_M$ (class means) and H being the matrix entropy, Song et al. (2024) connects Neural Collapse to the metrics:

$$\mathrm{MIR} = \frac{\mathrm{H}(\mathbf{G}_W) + \mathrm{H}(\mathbf{G}_M) - \mathrm{H}(\mathbf{G}_W \odot \mathbf{G}_M)}{\min\{\mathrm{H}(\mathbf{G}_W), \mathrm{H}(\mathbf{G}_M)\}}, \qquad \mathrm{HDR} = \frac{|\mathrm{H}(\mathbf{G}_W) - \mathrm{H}(\mathbf{G}_M)|}{\max\{\mathrm{H}(\mathbf{G}_W), \mathrm{H}(\mathbf{G}_M)\}} \quad (7)$$

These capture the information-theoretic signatures of **NC2** and **NC3** where under full collapse $\mathrm{MIR} \to 1$ and $\mathrm{HDR} \to 0$, reflecting perfect structural alignment.

In Table 2 four key findings are revealed: **(i) CE fails to achieve NC:** high intra-class variance (erank 22.5/96.4), suboptimal inter-class separation (erank 8.6/57.1 vs. theoretical K-1=9/99), and poor weight-feature alignment (w-inst 0.59/0.83, inst-inst 0.69/1.05). **(ii) Normalized softmax losses satisfy NC2-NC3** since they achieve perfect inter-class separation (erank 9.0/99.0), near-zero alignment errors (NTCE: w-inst 0.08/0.01, inst-inst 0.10/0.05), and optimal weight dimensionality matching the simplex ETF, with **NONL being the overall best** mostly due to its better intra class structure. **(iii) SCL with linear probing violates NC3:** despite superior within-class collapse (erank 4.5/7.5), inter-class structure degrades (erank 9.0/66.7) and classifier-feature alignment fails (w-inst 0.99/1.03). **(iv) Fixed prototypes restore NC3 in SCL:** removing the trainable classifier enforces perfect alignment by construction, though inter-class separation remains suboptimal.

While CE and ETF+ DR attain slightly better MIR/HDR values than our normalized losses, these information-theoretic metrics primarily reflect the overall entropy/redundancy of the representation, not the NC geometry itself. In our case, CE appears to preserve a bit more raw variability, but organizes it in a less NC-like, less prototype-structured way (higher intra-class effective rank, weaker alignment), whereas our normalized losses reshape the same information into a cleaner NC geometry. As our downstream experiments show in Section 5.3, this structured organization is more beneficial for transfer, long-tailed performance, and robustness, even though CE may capture slightly more "information" by these metrics.

Table 2: NC metrics on CIFAR-10/100 (training). **Bold** marks the best within each learning family; green marks the overall best per dataset. Theoretical optima: Intra ER 0/0, Inter ER 9/99, Weights ER 9/99, Weight Align 0/0, Instance Align 0/0, MIR 1/1, HDR 0/0.

| Learning Family | Method | Effective Rank | | | Alignment | | Information Theory Metrics | |
|---|---|---|---|---|---|---|---|---|
| | | Intra ↓ | Inter ↑ | Weights ↑ | Weight ↓ | Instance ↓ | MIR ↑ | HDR ↓ |
| | CE | 22.5 / 96.4 | 8.6 / 57.1 | 8.9 / 89.7 | 0.59 / 0.83 | 0.69 / 1.05 | **0.98** / 0.97 | 0.03 / 0.13 |
| | ETF + DR | 9.00 / 18.4 | 8.90 / 94.8 | **9.00 / 99.00** | 0.58 / 0.59 | 0.59 / 0.61 | **0.98** / **1.00** | **0.02** / **0.11** |
| CLASSIFIER | NormFace | 10.5 / 13.6 | **9.0** / 96.2 | **9.0** / 96.1 | 0.12 / **0.01** | 0.14 / 0.06 | 0.95 / **1.00** | 0.04 / 0.30 |
| LEARNING | NTCE | 9.0 / 12.6 | **9.0 / 99.0** | 8.9 / 98.9 | **0.08** / **0.01** | **0.10 / 0.05** | 0.96 / **1.00** | 0.05 / 0.30 |
| | NONL | **4.0 / 11.4** | **9.0 / 99.0** | **9.0 / 99.0** | 0.11 / **0.01** | 0.16 / 0.06 | 0.95 / **1.00** | 0.05 / 0.30 |
| CONTRASTIVE | SCL (w probing) | **4.5 / 7.5** | **9.0** / 66.7 | 8.3 / **77.8** | 0.99 / 1.03 | **0.10** / 0.34 | 0.99 / **0.95** | **0.07 / 0.11** |
| LEARNING | SCL (w/o probing) | **4.5 / 7.5** | **9.0** / 66.7 | **9.0** / 66.5 | **0.00 / 0.00** | **0.10** / 0.34 | **1.00** / 0.87 | 0.09 / 0.14 |

Table 3: Convergence speed (% of training iters): **(I)** time to reach the 95% NC threshold; **(II)** time to match CE's final value; "0%" indicates the target is met at the first logged eval.

| Method | Instance alignment | Weight alignment | Weights erank | Intra erank | Inter erank |
|---|---|---|---|---|---|
| **(I) NC convergence to 95% threshold (ratio to max iterations)** | | | | | |
| NormFace | 79.4% | 8.2% | 52.6% | 45.4% | 56.2% |
| NTCE | 79.4% | 6.8% | 56.4% | 36.6% | 52.4% |
| NONL | 79.4% | 7.4% | 34.6% | 14.6% | 47.2% |
| **(II) CE convergence to converged value (ratio to CE converged iteration)** | | | | | |
| NormFace | 2.2% | 2.0% | 66.3% | 0% | 7.4% |
| NTCE | 2.2% | 1.8% | 73.9% | 0% | 7.4% |
| NONL | 2.2% | 1.8% | 35.4% | 0% | 6.0% |

**Convergence Dynamics.** On CIFAR-100, we track NC metrics and define convergence as the earliest iteration where the exponentially-weighted moving average enters and remains within a metric-specific tolerance around the 95% NC threshold.

In Table 3(I) the convergence speed to 95% of theoretical NC thresholds is quantified. Normalized losses reach these thresholds, *typically early in training*. NONL converges faster with **gains over NormFace for the rank metrics** (1.2-3.1 speedup), benefiting from simplified optimization without competing terms. Table 3(II) benchmarks against CE's converged values. The acceleration is dramatic: normalized losses reach CE-equivalent values in under 7.5% of CE's required iterations across 4/5 metrics, while **NONL converges faster**. This demonstrates that normalized losses fundamentally restructure the optimization landscape, *enabling direct paths to neural collapse*.

## 5.3 PRACTICAL BENEFITS OF COLLAPSED REPRESENTATIONS

**Transfer learning.** We first ask whether representations that lie closer to the NC regime are more generalizable to unseen tasks. Following typical pipelines (Khosla et al., 2020), we freeze the pre-trained encoder for each loss and train a linear classifier (or detection head for VOC07) on eight

Table 4: Transfer learning results. Numbers are top-1 accuracy (%) for all datasets except VOC2007 (mAP). Best per column in green. The last row reports NONL's relative improvement over CE.

| Method | Food | CIFAR10 | CIFAR100 | Cars | DTD | Pets | Flowers | VOC2007 | Mean |
|---|---|---|---|---|---|---|---|---|---|
| CE | 68.0 | 88.6 | 67.7 | 25.9 | 69.9 | 67.0 | 81.4 | 67.1 | **67.0** |
| ETF + DR | 57.2 | 83.9 | 54.3 | 9.8 | 64.0 | 49.2 | 58.3 | 60.0 | **54.6** |
| NormFace | 69.8 | 89.8 | 69.7 | 29.7 | **70.1** | 69.7 | 83.2 | 67.9 | **68.7** |
| NTCE | 69.3 | 89.9 | 69.8 | 28.1 | 70.0 | 69.4 | 81.9 | 67.8 | **68.3** |
| NONL | **70.7** | **90.0** | **71.0** | **38.1** | 69.4 | **72.9** | **85.2** | **68.3** | **70.7** |
| Δ(NONL−CE) | +4.0% | +1.6% | +4.9% | +47.1% | −0.7% | +8.8% | +4.7% | +1.8% | **+5.5%** |

Table 5: Performance under class imbalance on CIFAR-10-LT/100-LT vs. imbalance ratio $\tau$ (Yang et al., 2022). Best per column in **green**; last row shows NONL's relative improvement over CE.

| | CIFAR-10-LT | | | CIFAR-100-LT | | |
|---|---|---|---|---|---|---|
| Method | $\tau = 0.1$ | 0.02 | 0.01 | $\tau = 0.1$ | 0.02 | 0.01 |
| CE | 88.1 | 76.8 | 70.2 | 55.5 | 41.5 | 37.4 |
| ETF + DR | 88.0 | 77.8 | 71.3 | 54.4 | 40.6 | 36.2 |
| NormFace | 88.0 | 79.0 | 74.3 | 54.6 | 40.1 | 35.9 |
| NTCE | 88.8 | **80.8** | **77.3** | 56.8 | 43.7 | 39.0 |
| NONL | **89.2** | 80.5 | 76.3 | **57.4** | **44.7** | **40.0** |
| $\Delta$(NONL − CE) | +1.2% | +4.8% | +8.7% | +3.4% | +7.7% | +7.0% |

Table 6: Clean error, mCE, and corruption error (%) on ImageNet-C. Best per column in **green**.

| Network | Error | mCE | Noise | | | Blur | | | | Weather | | | | Digital | | | |
|---|---|---|---|---|---|---|---|---|---|---|---|---|---|---|---|---|---|
| | | | Gauss | Shot | Impulse | Defoc | Glass | Motion | Zoom | Snow | Frost | Fog | Bright | Contrast | Elastic | Pixel | JPEG |
| CE | 25.0 | 80.1 | 75 | 78 | 83 | **84** | 94 | 86 | 88 | 80 | 78 | 68 | 62 | 66 | 96 | 84 | 80 |
| ETF + DR | 24.6 | 79.2 | 75 | 78 | 82 | 86 | 94 | 86 | 86 | **76** | 77 | 66 | 62 | 67 | 93 | 80 | 82 |
| NormFace | 23.6 | 77.8 | 74 | 76 | 82 | **84** | **93** | **81** | 85 | **76** | 75 | 64 | 60 | **65** | **91** | 81 | 80 |
| NTCE | **23.3** | **77.6** | 73 | 76 | **80** | **84** | **93** | 83 | 85 | **76** | 75 | 65 | 60 | **65** | 93 | **78** | 79 |
| NONL | 23.5 | 77.8 | **72** | **75** | **80** | 85 | **93** | 83 | **85** | **76** | 75 | 64 | 60 | 66 | **91** | 81 | 81 |

diverse downstream datasets. As shown in Table 4, *NONL consistently yields strong transfer performance*: it attains the best accuracy on 7/8 datasets and delivers a +5.5% relative improvement in mean performance over CE, whule NTCE also consistently exceeds CE. These results confirm prior works (Papyan et al., 2020; Bartlett et al., 2017; Neyshabur et al., 2018) that explicitly encouraging NC-like geometry produces features that generalize better beyond the pretraining distribution.

**Long-tailed classification.** We next examine robustness to class imbalance using standard evaluation pipelines (Yang et al., 2022). For CIFAR-10-LT and CIFAR-100-LT, we construct long-tailed versions with three imbalance ratios and train all models directly on the imbalanced data. Table 5 shows that our NC-inducing objectives substantially improve minority-class performance: on CIFAR-100-LT, NONL outperforms CE by +3.4%, +7.7%, and +7.0% under increasing imbalance, with gains up to +8.7% across CIFAR-10/100-LT, and also surpasses the ETF+DR baseline (Yang et al., 2022). This suggests that enforcing NC-like geometric structure helps maintain class separability even when minority classes are severely underrepresented, complementing the improvements observed in transfer.

**Out-of-distribution robustness.** Finally, we evaluate robustness to common corruptions on ImageNet-C (Hendrycks & Dietterich, 2019). Models are trained on clean ImageNet-1K only and evaluated on corrupted variants, reporting clean top-1 error and mean Corruption Error (mCE) normalized as in Hendrycks & Dietterich (2019). As summarized in Table 6, our normalized losses reduce mCE compared to CE while also improving clean accuracy. Thus, NC-inducing objectives not only improve in-distribution performance, but also yield representations that are more robust to distribution shift, in line with their benefits for transfer and long-tailed recognition.

# 6 CONCLUSION

In this work, we address the mismatch between the theoretical optima of supervised objectives and their behavior in practice. Constraining learning to the unit hypersphere removes the radial degeneracy of cross-entropy and unifies normalized softmax and supervised contrastive learning (SCL) as a single prototype-contrast paradigm. Building on this view, we propose two objectives (NTCE and NONL) that accelerate convergence to Neural Collapse. Theoretically, we prove SCL already yields an optimal prototype classifier during contrastive training, eliminating the typical linear probing phase. Empirically, across four benchmarks including ImageNet-1K, our methods surpass CE accuracy, reach $\geq$95% on NC metrics, and attain NC geometry in substantially fewer iterations. Overall, supervised learning is recast as prototype-based classification on the hypersphere, narrowing the theory–practice gap while simplifying and speeding up training.

REPRODUCIBILITY STATEMENT

Our approach modifies only loss functions within standard pipelines. Results can be replicated by pluging configutrations from Section A.3 into popular codebases (*e.g.* `https://github.com/HobbitLong/SupContrast`) with minimal effort substituting the original loss with ours.

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

# A  APPENDIX

## A.1  NEURAL COLLAPSE OPTIMALITY OF NORMALIZED OBJECTIVES

We adopt the balanced unconstrained-features / layer-peeled model (UFM/LPM) (Tirer & Bruna, 2022; Yaras et al., 2022). The last-layer features $z_i \in \mathbb{R}^d$ and classifier weights $w_c \in \mathbb{R}^d$ are free

optimization variables. We work with their $\ell_2$-normalized versions

$$\boldsymbol{u}_i = \frac{\boldsymbol{z}_i}{\|\boldsymbol{z}_i\|}, \qquad \hat{\boldsymbol{w}}_c = \frac{\boldsymbol{w}_c}{\|\boldsymbol{w}_c\|},$$

so that $\|\boldsymbol{u}_i\| = \|\hat{\boldsymbol{w}}_c\| = 1$ and

$$S_{ic} := \boldsymbol{u}_i^\top \hat{\boldsymbol{w}}_c.$$

There are $K$ classes and $M$ training samples, and the dataset is balanced: each class $c$ has index set

$$I_c := \{i : y_i = c\} \quad \text{with} \quad |I_c| = n = M/K.$$

We assume $d \geq K$.

**Normalized CE–based losses.** We consider three normalized cross-entropy–based losses with temperature $\tau > 0$:

$$\mathcal{L}_{\mathrm{NF}} = -\frac{1}{M} \sum_{i=1}^{M} \log \frac{\exp(S_{i,y_i}/\tau)}{\sum_{c=1}^{K} \exp(S_{ic}/\tau)}, \tag{8}$$

$$\mathcal{L}_{\mathrm{NTCE}} = -\frac{1}{M} \sum_{i=1}^{M} \log \frac{\exp(S_{i,y_i}/\tau)}{\sum_{j=1}^{M} \exp(S_{j,y_i}/\tau)}, \tag{9}$$

$$\mathcal{L}_{\mathrm{NONL}} = -\frac{1}{M} \sum_{i=1}^{M} \log \frac{\exp(S_{i,y_i}/\tau)}{\sum_{j:\, y_j \neq y_i} \exp(S_{j,y_i}/\tau)}. \tag{10}$$

**Neural Collapse properties.** We say a configuration exhibits *Neural Collapse* (NC) if there exist unit vectors $\boldsymbol{\mu}_1, \ldots, \boldsymbol{\mu}_K \in \mathbb{R}^d$ such that:

(NC1) (Within-class collapse) $\boldsymbol{u}_i = \boldsymbol{\mu}_{y_i}$ for all $i$.

(NC2) (Simplex ETF) the vectors $\{\boldsymbol{\mu}_c\}$ form a centered regular simplex in a $(K-1)$–dimensional subspace:

$$\|\boldsymbol{\mu}_c\| = 1 \quad \text{and} \quad \boldsymbol{\mu}_c^\top \boldsymbol{\mu}_{c'} = -\frac{1}{K-1} \quad \forall c \neq c'.$$

(NC3) (Classifier–mean alignment) $\hat{\boldsymbol{w}}_c = \boldsymbol{\mu}_c$ for all $c$.

At such a configuration the (normalized-feature) class means $\hat{\boldsymbol{\mu}}_c := \frac{1}{n} \sum_{i \in I_c} \boldsymbol{u}_i$ coincide with $\boldsymbol{\mu}_c$ and are therefore unit norm.

**Theorem A.1** (NC optimality of normalized CE–based losses). *In the balanced UFM/LPM setting above with $d \geq K$, every global minimizer of $\mathcal{L}_{\mathrm{NF}}$, $\mathcal{L}_{\mathrm{NTCE}}$, and $\mathcal{L}_{\mathrm{NONL}}$ satisfies NC1–NC3, up to a global rotation and permutation of class labels.*

We now analyze the three losses in turn.

NORMFACE

Yaras et al. (2022) study the constrained UFM problem

$$\min_{H,W} \frac{1}{M} \sum_{i=1}^{M} \mathrm{CE}\big(\tau' W^\top \boldsymbol{h}_i, y_i\big) \quad \text{s.t. } \|\boldsymbol{h}_i\| = 1, \ \|\boldsymbol{w}_c\| = 1, \tag{11}$$

with $\tau' > 0$, where CE is the standard cross-entropy.

**Lemma A.2** (NormFace $\equiv$ Yaras et al.). *Set $\tau' = 1/\tau$, and identify $\boldsymbol{h}_i = \boldsymbol{u}_i$ and $\boldsymbol{w}_c = \hat{\boldsymbol{w}}_c$. Then $\mathcal{L}_{\mathrm{NF}}$ coincides with equation 11, and $\arg\min \mathcal{L}_{\mathrm{NF}}$ equals the set of global minimizers of equation 11 over unit-norm features and weights.*

*Proof.* With $\boldsymbol{h}_i = \boldsymbol{u}_i$, $\boldsymbol{w}_c = \hat{\boldsymbol{w}}_c$ and $\tau' = 1/\tau$, we have $\big(\tau' W^\top \boldsymbol{h}_i\big)_c = S_{ic}/\tau$, so the summand in equation 11 is exactly $-\log \frac{\exp(S_{i,y_i}/\tau)}{\sum_c \exp(S_{ic}/\tau)}$, which is the $i$th summand in $\mathcal{L}_{\mathrm{NF}}$. Averaging over $i$ gives the claim. $\square$

Theorem 3.1 of Yaras et al. (2022) states that, under balanced labels and $d \geq K$, every global minimizer of equation 11 satisfies NC1–NC3. Together with Lemma A.2, this implies that every global minimizer of $\mathcal{L}_{\mathrm{NF}}$ satisfies NC1–NC3.

NTCE

We now show that every global minimizer of $\mathcal{L}_{\text{NTCE}}$ satisfies NC1–NC3. The proof follows the same three-step pattern we later use for NONL: we first reduce to a class-level objective depending only on class means and weights, then view this function as a contrastive loss of La/Lc type from (Koromilas et al., 2024) and apply its respective minimizer characterization of at the class level, and finally lift the resulting structure back to the sample level.

Recall that

$$\mathcal{L}_{\text{NTCE}} = -\frac{1}{M}\sum_{i=1}^{M}\log\frac{\exp(\boldsymbol{u}_i^\top\hat{\boldsymbol{w}}_{y_i}/\tau)}{\sum_{j=1}^{M}\exp(\boldsymbol{u}_j^\top\hat{\boldsymbol{w}}_{y_i}/\tau)} \;=\; \frac{1}{M}\sum_{i=1}^{M}\ell_i^{\text{NTCE}},$$

with per-sample loss

$$\ell_i^{\text{NTCE}} := -\log\frac{\exp(\boldsymbol{u}_i^\top\hat{\boldsymbol{w}}_{y_i}/\tau)}{\sum_{j=1}^{M}\exp(\boldsymbol{u}_j^\top\hat{\boldsymbol{w}}_{y_i}/\tau)}.$$

We again work in the balanced setting $|I_c| = n = M/K$ and $\|\boldsymbol{u}_i\| = \|\hat{\boldsymbol{w}}_c\| = 1$.

**Step 1: reduction to class means.**

**Lemma A.3** (NTCE reduction via class means). *Assume balanced labels, $|I_c| = n = M/K$ for all $c$. For any configuration $\{\boldsymbol{u}_i\}, \{\hat{\boldsymbol{w}}_c\}$ with $\|\boldsymbol{u}_i\| = \|\hat{\boldsymbol{w}}_c\| = 1$ define the normalized-feature class means*

$$\hat{\boldsymbol{\mu}}_c := \frac{1}{n}\sum_{j\in I_c}\boldsymbol{u}_j.$$

*Then*

$$\mathcal{L}_{\text{NTCE}}(\{\boldsymbol{u}_i\}, \{\hat{\boldsymbol{w}}_c\}) \;\geq\; L_{\text{NTCE}}^{\text{cls}}(\{\hat{\boldsymbol{\mu}}_c\}, \{\hat{\boldsymbol{w}}_c\}),$$

*where the class-level loss is*

$$L_{\text{NTCE}}^{\text{cls}} := -\frac{1}{K\tau}\sum_{c=1}^{K}\hat{\boldsymbol{w}}_c^\top\hat{\boldsymbol{\mu}}_c + \frac{1}{K}\sum_{c=1}^{K}\log\left(\sum_{c'=1}^{K}n\exp\left(\hat{\boldsymbol{w}}_c^\top\hat{\boldsymbol{\mu}}_{c'}/\tau\right)\right). \tag{12}$$

*Moreover, the inequality is tight if and only if, for every ordered pair $(c, c')$, the logits $\hat{\boldsymbol{w}}_c^\top\boldsymbol{u}_j$ are constant over $j\in I_{c'}$, i.e.*

$$\hat{\boldsymbol{w}}_c^\top\boldsymbol{u}_j = \hat{\boldsymbol{w}}_c^\top\hat{\boldsymbol{\mu}}_{c'} \quad \text{for all } j\in I_{c'}.$$

*Proof.* Fix a configuration $\{\boldsymbol{u}_i\}, \{\hat{\boldsymbol{w}}_c\}$ and define the class means $\hat{\boldsymbol{\mu}}_c$ as above. Using the balanced labels, write $\mathcal{L}_{\text{NTCE}}$ as an average over classes. For $i\in I_c$ we have

$$\ell_i^{\text{NTCE}} = -\frac{1}{\tau}\hat{\boldsymbol{w}}_c^\top\boldsymbol{u}_i + \log\Big(\sum_{j=1}^{M}\exp(\hat{\boldsymbol{w}}_c^\top\boldsymbol{u}_j/\tau)\Big),$$

so

$$\mathcal{L}_{\text{NTCE}} = \frac{1}{M}\sum_{c=1}^{K}\sum_{i\in I_c}\ell_i^{\text{NTCE}}$$

$$= -\frac{1}{M\tau}\sum_{c=1}^{K}\sum_{i\in I_c}\hat{\boldsymbol{w}}_c^\top\boldsymbol{u}_i + \frac{1}{M}\sum_{c=1}^{K}\sum_{i\in I_c}\log\Big(\sum_{j=1}^{M}\exp(\hat{\boldsymbol{w}}_c^\top\boldsymbol{u}_j/\tau)\Big).$$

The denominator term inside the logarithm depends only on the anchor class $c$, not on $i$, so $\sum_{i\in I_c}$ introduces a factor of $|I_c| = n$. Using $M = nK$ and the definition of $\hat{\boldsymbol{\mu}}_c$ we obtain

$$\mathcal{L}_{\text{NTCE}} = -\frac{1}{K\tau}\sum_{c=1}^{K}\hat{\boldsymbol{w}}_c^\top\Big(\frac{1}{n}\sum_{i\in I_c}\boldsymbol{u}_i\Big) + \frac{1}{K}\sum_{c=1}^{K}\log\Big(\sum_{j=1}^{M}\exp(\hat{\boldsymbol{w}}_c^\top\boldsymbol{u}_j/\tau)\Big)$$

$$= -\frac{1}{K\tau}\sum_{c=1}^{K}\hat{\boldsymbol{w}}_c^\top\hat{\boldsymbol{\mu}}_c + \frac{1}{K}\sum_{c=1}^{K}\log\Big(\sum_{j=1}^{M}\exp(\hat{\boldsymbol{w}}_c^\top\boldsymbol{u}_j/\tau)\Big).$$

For each fixed anchor class $c$, split the denominator over classes:

$$\sum_{j=1}^{M} \exp(\hat{\boldsymbol{w}}_c^\top \boldsymbol{u}_j / \tau) = \sum_{c'=1}^{K} \sum_{j \in I_{c'}} \exp(\hat{\boldsymbol{w}}_c^\top \boldsymbol{u}_j / \tau).$$

For fixed $(c, c')$, the function $f_c(\boldsymbol{x}) := \exp(\hat{\boldsymbol{w}}_c^\top \boldsymbol{x} / \tau)$ is convex in $\boldsymbol{x}$, so by Jensen's inequality over $j \in I_{c'}$,

$$\frac{1}{n} \sum_{j \in I_{c'}} \exp(\hat{\boldsymbol{w}}_c^\top \boldsymbol{u}_j / \tau) = \frac{1}{n} \sum_{j \in I_{c'}} f_c(\boldsymbol{u}_j) \;\geq\; f_c\Big(\frac{1}{n} \sum_{j \in I_{c'}} \boldsymbol{u}_j\Big) = \exp(\hat{\boldsymbol{w}}_c^\top \hat{\boldsymbol{\mu}}_{c'} / \tau).$$

Multiplying by $n$ and summing over $c'$ yields

$$\sum_{j=1}^{M} \exp(\hat{\boldsymbol{w}}_c^\top \boldsymbol{u}_j / \tau) \;\geq\; \sum_{c'=1}^{K} n \, \exp(\hat{\boldsymbol{w}}_c^\top \hat{\boldsymbol{\mu}}_{c'} / \tau).$$

Taking logs and averaging over $c$ gives

$$\mathcal{L}_{\mathrm{NTCE}} \;\geq\; -\frac{1}{K\tau} \sum_{c=1}^{K} \hat{\boldsymbol{w}}_c^\top \hat{\boldsymbol{\mu}}_c + \frac{1}{K} \sum_{c=1}^{K} \log \left( \sum_{c'=1}^{K} n \, \exp(\hat{\boldsymbol{w}}_c^\top \hat{\boldsymbol{\mu}}_{c'} / \tau) \right) = L_{\mathrm{NTCE}}^{\mathrm{cls}}.$$

Jensen's inequality is tight for a given pair $(c, c')$ if and only if the arguments of $f_c$ are constant over $j \in I_{c'}$, i.e. if and only if $\hat{\boldsymbol{w}}_c^\top \boldsymbol{u}_j$ is constant in $j$ on $I_{c'}$. In that case this constant must equal $\hat{\boldsymbol{w}}_c^\top \hat{\boldsymbol{\mu}}_{c'}$. Tightness for all $c, c'$ gives the stated condition. $\qquad\square$

Thus, for any configuration of unit features and weights, the NTCE loss is lower-bounded by the class-level objective $L_{\mathrm{NTCE}}^{\mathrm{cls}}$ depending only on the $K$ class means $\hat{\boldsymbol{\mu}}_c$ and the $K$ classifier weights $\hat{\boldsymbol{w}}_c$, and Lemma A.3 precisely characterizes when this lower bound is tight (blockwise constant logits).

It is convenient to separate out the constant $\log n$ factor, and to view the class means and weights abstractly as unit vectors. Define the *normalized* class-level NTCE loss

$$\tilde{L}_{\mathrm{NTCE}}^{\mathrm{cls}}(\{\hat{\boldsymbol{\mu}}_c\}, \{\hat{\boldsymbol{w}}_c\}) := -\frac{1}{K\tau} \sum_{c=1}^{K} \hat{\boldsymbol{w}}_c^\top \hat{\boldsymbol{\mu}}_c + \frac{1}{K} \sum_{c=1}^{K} \log \left( \sum_{c'=1}^{K} \exp\big( \hat{\boldsymbol{w}}_c^\top \hat{\boldsymbol{\mu}}_{c'} / \tau \big) \right),$$

so that

$$L_{\mathrm{NTCE}}^{\mathrm{cls}} = \log n + \tilde{L}_{\mathrm{NTCE}}^{\mathrm{cls}}.$$

In what follows, we treat the pairs $(\hat{\boldsymbol{\mu}}_c, \hat{\boldsymbol{w}}_c)$ as free variables on the unit sphere and, to lighten notation, write $\boldsymbol{\mu}_c := \hat{\boldsymbol{\mu}}_c$ and $\boldsymbol{w}_c := \hat{\boldsymbol{w}}_c$.

**Step 2: analysis of the class-level problem.** For each class $c$ we view the $c$th summand in $\tilde{L}_{\mathrm{NTCE}}^{\mathrm{cls}}$ as a standard contrastive loss of La/Lc type (Koromilas et al., 2024), with

$$q_c = \boldsymbol{w}_c \quad \text{(anchor)}, \qquad k_c^+ = \boldsymbol{\mu}_c \quad \text{(positive)}, \qquad \{k_c^- = \boldsymbol{\mu}_{c'} : c' \neq c\} \quad \text{(negatives)}.$$

The per-class alignment and contrastive terms are

$$L_{\mathrm{a}}(q_c, k_c^+) = -\frac{1}{\tau} q_c^\top k_c^+, \qquad L_{\mathrm{c}}(q_c, \{k_c^-\}) = \log \Big( \sum_{c'=1}^{K} \exp(q_c^\top k_{c'}^- / \tau) \Big).$$

The La/Lc framework requires $L_{\mathrm{a}}$ to be strictly decreasing in similarity and $L_{\mathrm{c}}$ to be convex and strictly increasing in similarity. These conditions hold here:

- $q_c^\top k_c^+$ enters $L_{\mathrm{a}}$ linearly with a negative coefficient, so $L_{\mathrm{a}}$ decreases as $q_c^\top k_c^+$ increases.
- $L_{\mathrm{c}}$ is a log-sum-exp of the similarities $q_c^\top k_c^- / \tau$, hence convex and strictly increasing in each similarity argument.

Therefore we may invoke the minimizer characterization for La/Lc losses. By Theorem 4.1 and Appendix B.1 of Koromilas et al. (2024), provided $d \geq K$, the global minimizers of $\tilde{L}_{\mathrm{NTCE}}^{\mathrm{cls}}$ over unit vectors satisfy:

- **Perfect alignment:** $\boldsymbol{\mu}_c = \boldsymbol{w}_c$ for all $c$.
- **Simplex ETF structure:** the directions $\{\boldsymbol{\mu}_c\}_{c=1}^K$ form a centered regular simplex equiangular tight frame in a $(K-1)$–dimensional subspace:

$$\|\boldsymbol{\mu}_c\| = 1, \qquad \boldsymbol{\mu}_c^\top \boldsymbol{\mu}_{c'} = -\frac{1}{K-1} \quad \forall c \neq c'.$$

In particular, there exists a simplex ETF $\{\boldsymbol{\mu}_c\}_{c=1}^K \subset \mathbb{R}^d$ such that $\boldsymbol{\mu}_c = \boldsymbol{w}_c$ is a global minimizer of $\tilde{L}_{\mathrm{NTCE}}^{\mathrm{cls}}$, unique up to a global rotation and permutation of the class indices.

**Step 3: lifting back to the sample level.** We now relate these class-level minimizers back to the original sample-level NTCE objective and derive the NC structure of its global minimizers.

*Existence of Neural Collapse minimizers.* Let $\{\boldsymbol{\mu}_c\}_{c=1}^K$ be a simplex ETF and set

$$\hat{\boldsymbol{w}}_c := \boldsymbol{\mu}_c, \qquad \boldsymbol{u}_i := \boldsymbol{\mu}_{y_i} \quad \text{for all } i.$$

This configuration satisfies NC1–NC3 by construction: within each class $c$, all normalized features collapse to $\boldsymbol{\mu}_c$ (NC1), the vectors $\{\boldsymbol{\mu}_c\}$ form a centered simplex ETF (NC2), and $\hat{\boldsymbol{w}}_c = \boldsymbol{\mu}_c$ (NC3). In particular, the feature class means are $\hat{\boldsymbol{\mu}}_c = \boldsymbol{\mu}_c$.

Moreover, for this configuration the Jensen inequalities in Lemma A.3 are tight: for any anchor class $c$ and any class $c'$, we have $\hat{\boldsymbol{w}}_c^\top \boldsymbol{u}_j = \boldsymbol{\mu}_c^\top \boldsymbol{\mu}_{c'}$ for all $j \in I_{c'}$, so the logits are constant within each class. Hence

$$\mathcal{L}_{\mathrm{NTCE}} = L_{\mathrm{NTCE}}^{\mathrm{cls}}(\{\hat{\boldsymbol{\mu}}_c\}, \{\hat{\boldsymbol{w}}_c\}) = \log n + \tilde{L}_{\mathrm{NTCE}}^{\mathrm{cls}}(\{\boldsymbol{\mu}_c\}, \{\boldsymbol{\mu}_c\}).$$

Since $\{\boldsymbol{\mu}_c\}, \{\boldsymbol{\mu}_c\}$ is a global minimizer of $\tilde{L}_{\mathrm{NTCE}}^{\mathrm{cls}}$, this shows that

$$\inf_{\{\boldsymbol{u}_i\}, \{\hat{\boldsymbol{w}}_c\}} \mathcal{L}_{\mathrm{NTCE}} \leq \log n + \inf_{\{\boldsymbol{\mu}_c\}, \{\boldsymbol{w}_c\}} \tilde{L}_{\mathrm{NTCE}}^{\mathrm{cls}}.$$

*Structure of arbitrary global minimizers.* Conversely, let $(\{\boldsymbol{u}_i^\star\}, \{\hat{\boldsymbol{w}}_c^\star\})$ be any global minimizer of $\mathcal{L}_{\mathrm{NTCE}}$, and let

$$\hat{\boldsymbol{\mu}}_c^\star := \frac{1}{n} \sum_{j \in I_c} \boldsymbol{u}_j^\star$$

be the corresponding class means. Lemma A.3 gives

$$\mathcal{L}_{\mathrm{NTCE}}(\{\boldsymbol{u}_i^\star\}, \{\hat{\boldsymbol{w}}_c^\star\}) \geq L_{\mathrm{NTCE}}^{\mathrm{cls}}(\{\hat{\boldsymbol{\mu}}_c^\star\}, \{\hat{\boldsymbol{w}}_c^\star\}) = \log n + \tilde{L}_{\mathrm{NTCE}}^{\mathrm{cls}}(\{\hat{\boldsymbol{\mu}}_c^\star\}, \{\hat{\boldsymbol{w}}_c^\star\}).$$

On the other hand, from the ETF construction above we know that

$$\inf_{\{\boldsymbol{u}_i\}, \{\hat{\boldsymbol{w}}_c\}} \mathcal{L}_{\mathrm{NTCE}} \leq \log n + \inf_{\{\boldsymbol{\mu}_c\}, \{\boldsymbol{w}_c\}} \tilde{L}_{\mathrm{NTCE}}^{\mathrm{cls}}.$$

Since $(\{\boldsymbol{u}_i^\star\}, \{\hat{\boldsymbol{w}}_c^\star\})$ achieves the global infimum, the two displays must be equalities. Therefore:

- $\tilde{L}_{\mathrm{NTCE}}^{\mathrm{cls}}(\{\hat{\boldsymbol{\mu}}_c^\star\}, \{\hat{\boldsymbol{w}}_c^\star\})$ attains the global minimum of $\tilde{L}_{\mathrm{NTCE}}^{\mathrm{cls}}$, so by the La/Lc minimizer characterization we must have, up to a global rotation and permutation of class labels,

$$\hat{\boldsymbol{\mu}}_c^\star = \hat{\boldsymbol{w}}_c^\star \quad \text{for all } c, \qquad \{\hat{\boldsymbol{\mu}}_c^\star\} \text{ form a centered simplex ETF.}$$

- Lemma A.3 must be tight at the minimizer, so the Jensen equalities hold for all $(c, c')$: for every anchor class $c$ and every class $c'$, the logits $\hat{\boldsymbol{w}}_c^{\star\top} \boldsymbol{u}_j^\star$ are constant over $j \in I_{c'}$, equal to $\hat{\boldsymbol{w}}_c^{\star\top} \hat{\boldsymbol{\mu}}_{c'}^\star$.

Let $S := \mathrm{span}\{\hat{\boldsymbol{w}}_1^\star, \ldots, \hat{\boldsymbol{w}}_K^\star\}$, which is the $(K-1)$–dimensional simplex-ETF subspace. Fix a class $c'$ and $j \in I_{c'}$. For every $c \neq c'$, tightness of Jensen gives

$$\hat{\boldsymbol{w}}_c^{\star\top} (\boldsymbol{u}_j^\star - \hat{\boldsymbol{\mu}}_{c'}^\star) = 0.$$

Since $\{\hat{\boldsymbol{w}}_c^\star\}_{c=1}^K$ form a centered simplex ETF in the $(K-1)$–dimensional subspace $S$ and satisfy $\sum_{c=1}^K \hat{\boldsymbol{w}}_c^\star = 0$, any $K-1$ of them are linearly independent and thus span $S$. In particular, the set $\{\hat{\boldsymbol{w}}_c^\star : c \neq c'\}$ spans $S$, so $\boldsymbol{u}_j^\star - \hat{\boldsymbol{\mu}}_{c'}^\star$ is orthogonal to $S$, and hence the orthogonal projection of $\boldsymbol{u}_j^\star$ onto $S$ equals $\hat{\boldsymbol{\mu}}_{c'}^\star$.

But both $\boldsymbol{u}_j^\star$ and $\hat{\boldsymbol{\mu}}_{c'}^\star = \hat{\boldsymbol{w}}_{c'}^\star$ are unit vectors, and $\hat{\boldsymbol{\mu}}_{c'}^\star \in S$. The only way for a unit vector to have a unit-norm projection onto $S$ is to lie in $S$ itself and coincide with its projection, so we must have

$$\boldsymbol{u}_j^\star = \hat{\boldsymbol{\mu}}_{c'}^\star \quad \text{for all } j \in I_{c'}.$$

Thus within each class all features collapse to a single unit direction (NC1), these $K$ directions form a centered simplex ETF (NC2), and the classifier weights align with the class means (NC3). Therefore every global minimizer of $\mathcal{L}_{\text{NTCE}}$ exhibits Neural Collapse, up to a global rotation and permutation of the class labels.

NONL

We finally treat the NONL objective equation 10. The proof proceeds by first bounding the sample-level loss by a class-level objective depending only on class means and weights, then applying the La/Lc minimizer characterization at the class level, and finally lifting this structure back to the sample level to obtain NC1–NC3.

Recall that

$$\mathcal{L}_{\text{NONL}} = -\frac{1}{M} \sum_{i=1}^M \log \frac{\exp(S_{i,y_i}/\tau)}{\sum_{j:\, y_j \neq y_i} \exp(S_{j,y_i}/\tau)} \;=\; \frac{1}{M} \sum_{i=1}^M \ell_i^{\text{NONL}},$$

with per-sample loss

$$\ell_i^{\text{NONL}} := -\log \frac{\exp(\boldsymbol{u}_i^\top \hat{\boldsymbol{w}}_{y_i}/\tau)}{\sum_{j:\, y_j \neq y_i} \exp(\boldsymbol{u}_j^\top \hat{\boldsymbol{w}}_{y_i}/\tau)}.$$

We again work in the balanced setting $|I_c| = n = M/K$ and $\|\boldsymbol{u}_i\| = \|\hat{\boldsymbol{w}}_c\| = 1$.

**Step 1: reduction to class means.** We first show that the sample-level NONL loss admits a lower bound that depends only on the $K$ normalized-feature class means and the $K$ classifier weights.

**Lemma A.4** (NONL reduction via class means). *Assume balanced labels, $|I_c| = n = M/K$ for all $c$. For any configuration $\{\boldsymbol{u}_i\}, \{\hat{\boldsymbol{w}}_c\}$ with $\|\boldsymbol{u}_i\| = \|\hat{\boldsymbol{w}}_c\| = 1$ define the normalized-feature class means*

$$\hat{\boldsymbol{\mu}}_c := \frac{1}{n} \sum_{j \in I_c} \boldsymbol{u}_j.$$

*Then*

$$\mathcal{L}_{\text{NONL}}(\{\boldsymbol{u}_i\}, \{\hat{\boldsymbol{w}}_c\}) \;\geq\; L_{\text{NONL}}^{\text{cls}}(\{\hat{\boldsymbol{\mu}}_c\}, \{\hat{\boldsymbol{w}}_c\}),$$

*where the class-level loss is*

$$L_{\text{NONL}}^{\text{cls}} := -\frac{1}{K\tau} \sum_{c=1}^K \hat{\boldsymbol{w}}_c^\top \hat{\boldsymbol{\mu}}_c + \frac{1}{K} \sum_{c=1}^K \log \left( \sum_{c' \neq c} n \exp\left( \hat{\boldsymbol{w}}_c^\top \hat{\boldsymbol{\mu}}_{c'}/\tau \right) \right). \tag{13}$$

*Moreover, the inequality is tight if and only if, for every ordered pair $(c, c')$ with $c' \neq c$, the "negative" logits $\hat{\boldsymbol{w}}_c^\top \boldsymbol{u}_j$ are constant over $j \in I_{c'}$, i.e.*

$$\hat{\boldsymbol{w}}_c^\top \boldsymbol{u}_j = \hat{\boldsymbol{w}}_c^\top \hat{\boldsymbol{\mu}}_{c'} \quad \text{for all } j \in I_{c'}, \ c' \neq c.$$

*Proof.* Fix a sample index $i$ with label $y_i = c$. Its NONL denominator is

$$D_i^{\text{neg}} := \sum_{j:\, y_j \neq c} \exp(\boldsymbol{u}_j^\top \hat{\boldsymbol{w}}_c/\tau) = \sum_{c' \neq c} \sum_{j \in I_{c'}} \exp(\boldsymbol{u}_j^\top \hat{\boldsymbol{w}}_c/\tau).$$

For each anchor class $c$ and negative class $c' \neq c$, consider the function

$$f_c(\boldsymbol{x}) := \exp(\hat{\boldsymbol{w}}_c^\top \boldsymbol{x}/\tau),$$

which is convex in $\boldsymbol{x}$. Applying Jensen's inequality over the negative-class samples $\{\boldsymbol{u}_j : j \in I_{c'}\}$ gives

$$\frac{1}{n} \sum_{j \in I_{c'}} \exp(\hat{\boldsymbol{w}}_c^\top \boldsymbol{u}_j / \tau) = \frac{1}{n} \sum_{j \in I_{c'}} f_c(\boldsymbol{u}_j) \geq f_c\Big(\frac{1}{n} \sum_{j \in I_{c'}} \boldsymbol{u}_j\Big) = \exp\big(\hat{\boldsymbol{w}}_c^\top \hat{\boldsymbol{\mu}}_{c'} / \tau\big).$$

Multiplying by $n$ and summing over $c' \neq c$ yields

$$D_i^{\text{neg}} = \sum_{c' \neq c} \sum_{j \in I_{c'}} \exp(\boldsymbol{u}_j^\top \hat{\boldsymbol{w}}_c / \tau) \geq \sum_{c' \neq c} n \exp(\hat{\boldsymbol{w}}_c^\top \hat{\boldsymbol{\mu}}_{c'} / \tau).$$

By definition of $\ell_i^{\text{NONL}}$,

$$\ell_i^{\text{NONL}} = -\log \frac{\exp(\boldsymbol{u}_i^\top \hat{\boldsymbol{w}}_c / \tau)}{D_i^{\text{neg}}}$$

$$= -\frac{1}{\tau} \hat{\boldsymbol{w}}_c^\top \boldsymbol{u}_i + \log D_i^{\text{neg}}$$

$$\geq -\frac{1}{\tau} \hat{\boldsymbol{w}}_c^\top \boldsymbol{u}_i + \log \left( \sum_{c' \neq c} n \exp(\hat{\boldsymbol{w}}_c^\top \hat{\boldsymbol{\mu}}_{c'} / \tau) \right) =: \tilde{\ell}_i.$$

This inequality is tight if and only if all Jensen steps above are equalities. For a fixed $(c, c')$ with $c' \neq c$, equality in Jensen requires that the arguments of $f_c$ be constant over $j \in I_{c'}$, i.e. $\hat{\boldsymbol{w}}_c^\top \boldsymbol{u}_j$ is constant on $I_{c'}$. Using the definition of $\hat{\boldsymbol{\mu}}_{c'}$, this constant must then equal $\hat{\boldsymbol{w}}_c^\top \hat{\boldsymbol{\mu}}_{c'}$, giving the stated tightness condition.

Finally, average $\tilde{\ell}_i$ over all samples. Using the balanced labels $|I_c| = n$ and the definition of $\hat{\boldsymbol{\mu}}_c$,

$$\frac{1}{M} \sum_{i=1}^{M} \tilde{\ell}_i = \frac{1}{M} \sum_{c=1}^{K} \sum_{i \in I_c} \left[ -\frac{1}{\tau} \hat{\boldsymbol{w}}_c^\top \boldsymbol{u}_i + \log \left( \sum_{c' \neq c} n \exp(\hat{\boldsymbol{w}}_c^\top \hat{\boldsymbol{\mu}}_{c'} / \tau) \right) \right]$$

$$= -\frac{1}{M\tau} \sum_{c=1}^{K} \sum_{i \in I_c} \hat{\boldsymbol{w}}_c^\top \boldsymbol{u}_i + \frac{1}{M} \sum_{c=1}^{K} |I_c| \log \left( \sum_{c' \neq c} n \exp(\hat{\boldsymbol{w}}_c^\top \hat{\boldsymbol{\mu}}_{c'} / \tau) \right)$$

$$= -\frac{1}{K\tau} \sum_{c=1}^{K} \hat{\boldsymbol{w}}_c^\top \hat{\boldsymbol{\mu}}_c + \frac{1}{K} \sum_{c=1}^{K} \log \left( \sum_{c' \neq c} n \exp(\hat{\boldsymbol{w}}_c^\top \hat{\boldsymbol{\mu}}_{c'} / \tau) \right)$$

$$= L_{\text{NONL}}^{\text{cls}}(\{\hat{\boldsymbol{\mu}}_c\}, \{\hat{\boldsymbol{w}}_c\}).$$

Since $\mathcal{L}_{\text{NONL}}$ is the average of the $\ell_i^{\text{NONL}}$ and each $\ell_i^{\text{NONL}} \geq \tilde{\ell}_i$, we obtain $\mathcal{L}_{\text{NONL}} \geq L_{\text{NONL}}^{\text{cls}}$ with the stated equality condition. $\qquad \square$

As before, it is convenient to separate out the factor $n$ from the logarithm and to treat the class means and weights abstractly as unit vectors. Define the *normalized* class-level NONL loss

$$\tilde{L}_{\text{NONL}}^{\text{cls}}(\{\hat{\boldsymbol{\mu}}_c\}, \{\hat{\boldsymbol{w}}_c\}) := -\frac{1}{K\tau} \sum_{c=1}^{K} \hat{\boldsymbol{w}}_c^\top \hat{\boldsymbol{\mu}}_c + \frac{1}{K} \sum_{c=1}^{K} \log \left( \sum_{c' \neq c} \exp\big(\hat{\boldsymbol{w}}_c^\top \hat{\boldsymbol{\mu}}_{c'} / \tau\big) \right),$$

so that

$$L_{\text{NONL}}^{\text{cls}} = \log n + \tilde{L}_{\text{NONL}}^{\text{cls}}.$$

In what follows we again treat $(\hat{\boldsymbol{\mu}}_c, \hat{\boldsymbol{w}}_c)$ as free unit vectors and write $\boldsymbol{\mu}_c := \hat{\boldsymbol{\mu}}_c$ and $\boldsymbol{w}_c := \hat{\boldsymbol{w}}_c$.

**Step 2: analysis of the class-level problem.** For each class $c$ we can view the $c$th summand in $\tilde{L}_{\text{NONL}}^{\text{cls}}$ as a standard decoupled alignment/uniformity loss of La/Lc type (Koromilas et al., 2024), with:

$$q_c = \boldsymbol{w}_c \quad \text{(anchor)}, \qquad k_c^+ = \boldsymbol{\mu}_c \quad \text{(positive)}, \qquad \{k_c^- = \boldsymbol{\mu}_{c'} : c' \neq c\} \quad \text{(negatives)}.$$

The per-class alignment and contrastive terms are

$$L_{\mathrm{a}}(q_c, k_c^+) = -\frac{1}{\tau} q_c^\top k_c^+, \qquad L_{\mathrm{c}}(q_c, \{k_c^-\}) = \log \Big( \sum_{c' \neq c} \exp(q_c^\top k_{c'}^- / \tau) \Big).$$

As in the NTCE case, $L_{\mathrm{a}}$ is strictly decreasing in similarity and $L_{\mathrm{c}}$ is convex and strictly increasing in the similarities. Therefore we may again invoke the La/Lc minimizer characterization. By Theorem 4.1 and Appendix B.1 of Koromilas et al. (2024), provided $d \geq K$, the global minimizers of $\tilde{L}_{\mathrm{NONL}}^{\mathrm{cls}}$ over unit vectors satisfy:

- **Perfect alignment:** $\boldsymbol{\mu}_c = \boldsymbol{w}_c$ for all $c$.
- **Simplex ETF structure:** the directions $\{\boldsymbol{\mu}_c\}_{c=1}^K$ form a centered regular simplex equiangular tight frame in a $(K-1)$–dimensional subspace:

$$\|\boldsymbol{\mu}_c\| = 1, \qquad \boldsymbol{\mu}_c^\top \boldsymbol{\mu}_{c'} = -\frac{1}{K-1} \quad \forall c \neq c'.$$

In particular, there exists a simplex ETF $\{\boldsymbol{\mu}_c\}_{c=1}^K \subset \mathbb{R}^d$ such that $\boldsymbol{\mu}_c = \boldsymbol{w}_c$ is a global minimizer of $\tilde{L}_{\mathrm{NONL}}^{\mathrm{cls}}$, unique up to a global rotation and permutation of the class indices.

**Step 3: lifting back to the sample level.** We now relate these class-level minimizers back to the original sample-level NONL objective and derive the NC structure of its global minimizers.

*Existence of Neural Collapse minimizers.* Let $\{\boldsymbol{\mu}_c\}_{c=1}^K$ be a simplex ETF and set

$$\hat{\boldsymbol{w}}_c := \boldsymbol{\mu}_c, \qquad \boldsymbol{u}_i := \boldsymbol{\mu}_{y_i} \quad \text{for all } i.$$

This configuration satisfies NC1–NC3 by construction: within each class $c$, all normalized features collapse to $\boldsymbol{\mu}_c$ (NC1), the vectors $\{\boldsymbol{\mu}_c\}$ form a centered simplex ETF (NC2), and $\hat{\boldsymbol{w}}_c = \boldsymbol{\mu}_c$ (NC3). In particular, the feature class means are $\hat{\boldsymbol{\mu}}_c = \boldsymbol{\mu}_c$.

Moreover, for this configuration the Jensen inequalities in Lemma A.4 are tight for all $(c, c')$: for any anchor class $c$ and negative class $c' \neq c$ we have $\hat{\boldsymbol{w}}_c^\top \boldsymbol{u}_j = \boldsymbol{\mu}_c^\top \boldsymbol{\mu}_{c'}$ for all $j \in I_{c'}$, so the negative logits are constant within each negative class. Hence

$$\mathcal{L}_{\mathrm{NONL}} = L_{\mathrm{NONL}}^{\mathrm{cls}}(\{\hat{\boldsymbol{\mu}}_c\}, \{\hat{\boldsymbol{w}}_c\}) = \log n + \tilde{L}_{\mathrm{NONL}}^{\mathrm{cls}}(\{\boldsymbol{\mu}_c\}, \{\boldsymbol{\mu}_c\}).$$

Since $\{\boldsymbol{\mu}_c\}, \{\boldsymbol{\mu}_c\}$ is a global minimizer of $\tilde{L}_{\mathrm{NONL}}^{\mathrm{cls}}$, this shows that

$$\inf_{\{\boldsymbol{u}_i\}, \{\hat{\boldsymbol{w}}_c\}} \mathcal{L}_{\mathrm{NONL}} \leq \log n + \inf_{\{\boldsymbol{\mu}_c\}, \{\boldsymbol{w}_c\}} \tilde{L}_{\mathrm{NONL}}^{\mathrm{cls}}.$$

*Structure of arbitrary global minimizers.* Conversely, let $(\{\boldsymbol{u}_i^\star\}, \{\hat{\boldsymbol{w}}_c^\star\})$ be any global minimizer of $\mathcal{L}_{\mathrm{NONL}}$, and let

$$\hat{\boldsymbol{\mu}}_c^\star := \frac{1}{n} \sum_{j \in I_c} \boldsymbol{u}_j^\star$$

be the corresponding class means. Lemma A.4 gives

$$\mathcal{L}_{\mathrm{NONL}}(\{\boldsymbol{u}_i^\star\}, \{\hat{\boldsymbol{w}}_c^\star\}) \geq L_{\mathrm{NONL}}^{\mathrm{cls}}(\{\hat{\boldsymbol{\mu}}_c^\star\}, \{\hat{\boldsymbol{w}}_c^\star\}) = \log n + \tilde{L}_{\mathrm{NONL}}^{\mathrm{cls}}(\{\hat{\boldsymbol{\mu}}_c^\star\}, \{\hat{\boldsymbol{w}}_c^\star\}).$$

On the other hand, from the ETF construction above we know that

$$\inf_{\{\boldsymbol{u}_i\}, \{\hat{\boldsymbol{w}}_c\}} \mathcal{L}_{\mathrm{NONL}} \leq \log n + \inf_{\{\boldsymbol{\mu}_c\}, \{\boldsymbol{w}_c\}} \tilde{L}_{\mathrm{NONL}}^{\mathrm{cls}}.$$

Since $(\{\boldsymbol{u}_i^\star\}, \{\hat{\boldsymbol{w}}_c^\star\})$ achieves this infimum, the two displays must be equalities. Therefore:

- $\tilde{L}_{\mathrm{NONL}}^{\mathrm{cls}}(\{\hat{\boldsymbol{\mu}}_c^\star\}, \{\hat{\boldsymbol{w}}_c^\star\})$ attains the global minimum of $\tilde{L}_{\mathrm{NONL}}^{\mathrm{cls}}$, so by the La/Lc minimizer characterization we must have, up to a global rotation and permutation of class labels,

$$\hat{\boldsymbol{\mu}}_c^\star = \hat{\boldsymbol{w}}_c^\star \quad \text{for all } c, \qquad \{\hat{\boldsymbol{\mu}}_c^\star\} \text{ form a centered simplex ETF.}$$

- Lemma A.4 must be tight at the minimizer, so the Jensen equalities hold for all $(c, c')$: for every anchor class $c$ and negative class $c' \neq c$, the logits $\hat{\boldsymbol{w}}_c^{\star\top} \boldsymbol{u}_j^\star$ are constant over $j \in I_{c'}$, equal to $\hat{\boldsymbol{w}}_c^{\star\top} \hat{\boldsymbol{\mu}}_{c'}^\star$.

Let $S := \mathrm{span}\{\hat{\boldsymbol{w}}_1^\star, \ldots, \hat{\boldsymbol{w}}_K^\star\}$, which is the $(K-1)$–dimensional simplex-ETF subspace. Fix a class $c'$ and $j \in I_{c'}$. For every $c \neq c'$, tightness of Jensen gives

$$\hat{\boldsymbol{w}}_c^{\star\top}\left(\boldsymbol{u}_j^\star - \hat{\boldsymbol{\mu}}_{c'}^\star\right) = 0.$$

As before, since $\{\hat{\boldsymbol{w}}_c^\star\}_{c=1}^K$ form a centered simplex ETF in $S$ and $\sum_{c=1}^K \hat{\boldsymbol{w}}_c^\star = 0$, any $K-1$ of them are linearly independent and thus span $S$. In particular, the set $\{\hat{\boldsymbol{w}}_c^\star : c \neq c'\}$ spans $S$, so $\boldsymbol{u}_j^\star - \hat{\boldsymbol{\mu}}_{c'}^\star$ is orthogonal to $S$, and hence the orthogonal projection of $\boldsymbol{u}_j^\star$ onto $S$ equals $\hat{\boldsymbol{\mu}}_{c'}^\star$.

But both $\boldsymbol{u}_j^\star$ and $\hat{\boldsymbol{\mu}}_{c'}^\star = \hat{\boldsymbol{w}}_{c'}^\star$ are unit vectors, and $\hat{\boldsymbol{\mu}}_{c'}^\star \in S$. As in the NTCE case, the only way for a unit vector to have a unit-norm projection onto $S$ is to lie in $S$ itself and coincide with its projection, so we must have

$$\boldsymbol{u}_j^\star = \hat{\boldsymbol{\mu}}_{c'}^\star \quad \text{for all } j \in I_{c'}.$$

Thus within each class all features collapse to a single unit direction (NC1), these $K$ directions form a centered simplex ETF (NC2), and the classifier weights align with the class means (NC3). Therefore every global minimizer of $\mathcal{L}_{\mathrm{NONL}}$ exhibits Neural Collapse, up to a global rotation and permutation of the class labels.

*Proof of Theorem A.1.* **NormFace:** Lemma A.2 together with Theorem 3.1 of Yaras et al. (2022) shows that every global minimizer of $\mathcal{L}_{\mathrm{NF}}$ satisfies NC1–NC3.

**NTCE:** Lemma A.3 bounds $\mathcal{L}_{\mathrm{NTCE}}$ by the class-level loss $L_{\mathrm{NTCE}}^{\mathrm{cls}}$, while the La/Lc minimizer characterization (Step 2) identifies the global minimizers of $\tilde{L}_{\mathrm{NTCE}}^{\mathrm{cls}}$ as simplex ETF configurations with $\boldsymbol{\mu}_c = \boldsymbol{w}_c$. Step 3 shows that any global minimizer of $\mathcal{L}_{\mathrm{NTCE}}$ must both attain this class-level minimum and satisfy the tightness conditions in Lemma A.3, which enforces NC1. Together these yield NC1–NC3 for all NTCE minimizers.

**NONL:** Lemma A.4 bounds $\mathcal{L}_{\mathrm{NONL}}$ by the class-level loss $L_{\mathrm{NONL}}^{\mathrm{cls}}$, while the La/Lc minimizer characterization (Step 2) identifies the global minimizers of $\tilde{L}_{\mathrm{NONL}}^{\mathrm{cls}}$ as simplex ETF configurations with $\boldsymbol{\mu}_c = \boldsymbol{w}_c$. Step 3 shows that any global minimizer of $\mathcal{L}_{\mathrm{NONL}}$ must both attain this class-level minimum and satisfy the tightness conditions in Lemma A.4, which again enforces NC1. Together these yield NC1–NC3 for all NONL minimizers.

In all three cases, the resulting NC configuration is unique up to a global rotation and permutation of the class labels. This proves the theorem. $\square$

## A.2 Equivalence of SCL and prototype–softmax minimizers

Here we provide the proof of Theorem 4.2.

*Proof.* Fix $i \in [2M]$ with label $y_i$. Let $\mathcal{C}(i) = \{j \in [2M] : j \neq i, \ y_j = y_i\}$, $\mathcal{B}_c = \{j \in [2M] : y_j = c\}$, $n_c = |\mathcal{B}_c|$, and $\hat{\boldsymbol{\mu}}_c = \frac{1}{n_c} \sum_{j \in \mathcal{B}_c} \boldsymbol{a}_j$.

**(A) SCL lower bound.** By unfolding the SCL loss defined in Equation (2), the per-example loss term can be written as

$$\ell_i^{\mathrm{SCL}} = -\frac{1}{|\mathcal{C}(i)|} \sum_{l \in \mathcal{C}(i)} \frac{\boldsymbol{a}_i^\top \boldsymbol{a}_l}{\tau} + \log \sum_{j \in [2M] \setminus \{i\}} \exp(\boldsymbol{a}_i^\top \boldsymbol{a}_j / \tau).$$

For the first term, using $\frac{1}{|\mathcal{C}(i)|} \sum_{l \in \mathcal{C}(i)} \boldsymbol{a}_l = \frac{n_{y_i} \hat{\boldsymbol{\mu}}_{y_i} - \boldsymbol{a}_i}{n_{y_i} - 1}$ and $\|\boldsymbol{a}_i\| = 1$ gives $-\frac{\boldsymbol{a}_i^\top}{\tau}\left(\frac{1}{|\mathcal{C}(i)|} \sum_{l \in \mathcal{C}(i)} \boldsymbol{a}_l\right) \geq -\frac{\boldsymbol{a}_i^\top \hat{\boldsymbol{\mu}}_{y_i}}{\tau}$.

For the second term, we group by class, subtract the self term and then apply Jensen classwise due to convexity of the exponential function:

$$\sum_{j\in[2M]\setminus\{i\}} e^{\boldsymbol{a}_i^\top \boldsymbol{a}_j/\tau} = \sum_{c=1}^{K} \sum_{l\in\mathcal{B}_c} e^{\boldsymbol{a}_i^\top \boldsymbol{a}_l/\tau} - e^{1/\tau} \geq \sum_{c=1}^{K} n_c\, e^{\boldsymbol{a}_i^\top \hat{\boldsymbol{\mu}}_c/\tau} - e^{1/\tau}.$$

Combining,

$$\ell_i^{\text{SCL}} \geq -\frac{\boldsymbol{a}_i^\top \hat{\boldsymbol{\mu}}_{y_i}}{\tau} + \log\left(\sum_{c=1}^{K} n_c\, e^{\boldsymbol{a}_i^\top \hat{\boldsymbol{\mu}}_c/\tau} - e^{1/\tau}\right) =: \ell_i^\star. \tag{14}$$

Equality in equation 14 holds iff every class-wise sum is collapsed, i.e., $\boldsymbol{a}_j = \hat{\boldsymbol{\mu}}_c$ for all $j \in \mathcal{B}_c$, because the positive-term bound is tight only when $\boldsymbol{a}_i^\top \hat{\boldsymbol{\mu}}_{y_i} = 1$ (so $\boldsymbol{a}_i = \hat{\boldsymbol{\mu}}_{y_i}$) and the classwise Jensen step is tight only when all within-class logits $\{\boldsymbol{a}_i^\top \boldsymbol{a}_l : l \in \mathcal{B}_c\}$ are equal.

**(B) Prototype loss lower bound.** Since $\boldsymbol{a}_i^\top \hat{\boldsymbol{\mu}}_{y_i} \leq 1$ for unit vectors, $e^{\boldsymbol{a}_i^\top \hat{\boldsymbol{\mu}}_{y_i}/\tau} \leq e^{1/\tau}$. Therefore

$$\underbrace{\sum_{c=1}^{K} n_c\, e^{\boldsymbol{a}_i^\top \hat{\boldsymbol{\mu}}_c/\tau} - e^{\boldsymbol{a}_i^\top \hat{\boldsymbol{\mu}}_{y_i}/\tau}}_{=:D_i^{\text{proto}}} \geq \underbrace{\sum_{c=1}^{K} n_c\, e^{\boldsymbol{a}_i^\top \hat{\boldsymbol{\mu}}_c/\tau} - e^{1/\tau}}_{=:D_i^\star},$$

and thus, with the *same* numerator $e^{\boldsymbol{a}_i^\top \hat{\boldsymbol{\mu}}_{y_i}/\tau}$,

$$\ell_i^{\text{proto}} = -\frac{\boldsymbol{a}_i^\top \hat{\boldsymbol{\mu}}_{y_i}}{\tau} + \log D_i^{\text{proto}} \geq -\frac{\boldsymbol{a}_i^\top \hat{\boldsymbol{\mu}}_{y_i}}{\tau} + \log D_i^\star = \ell_i^\star.$$

Averaging over $i$ gives the following inequalities for any batch $\boldsymbol{A}$:

$$L_{\text{SCL}}(\boldsymbol{A}) \geq L_\star(\boldsymbol{A})$$
$$L_{\text{proto}}(\boldsymbol{A}) \geq L_\star(\boldsymbol{A}).$$

**(C) Collapse–simplex makes all three equal.** By Graf et al. (2021, Theorem 2), any SCL global minimizer exhibits class-wise collapse, $\boldsymbol{a}_j = \boldsymbol{\zeta}_{y_j}$, and the directions $\{\boldsymbol{\zeta}_c\}$ form a centered regular $(K-1)$-simplex. Hence $\hat{\boldsymbol{\mu}}_c = \boldsymbol{\zeta}_c$ and $\boldsymbol{a}_i^\top \hat{\boldsymbol{\mu}}_{y_i} = 1$ for all $i$, making both inequalities above tight:

$$L_{\text{SCL}}(\boldsymbol{A}^\star) = L_\star(\boldsymbol{A}^\star) = L_{\text{proto}}(\boldsymbol{A}^\star).$$

Therefore $\min L_{\text{SCL}} = \min L_\star = \min L_{\text{proto}}$, all attained at the collapsed-simplex configurations.

**(D) Equality of argmin sets.** Let $\boldsymbol{A}$ minimize $L_{\text{proto}}$. Then $L_{\text{proto}}(\boldsymbol{A}) = \min L_{\text{proto}} = \min L_\star$, so $L_\star(\boldsymbol{A}) = L_{\text{proto}}(\boldsymbol{A})$, which forces $e^{\boldsymbol{a}_i^\top \hat{\boldsymbol{\mu}}_{y_i}/\tau} = e^{1/\tau}$ for every $i$, i.e., $\boldsymbol{a}_i^\top \hat{\boldsymbol{\mu}}_{y_i} = 1$ and hence $\boldsymbol{a}_i = \hat{\boldsymbol{\mu}}_{y_i}$ (class-wise collapse). Moreover $L_{\text{SCL}}(\boldsymbol{A}) = L_\star(\boldsymbol{A}) = \min L_{\text{SCL}}$, so $\boldsymbol{A}$ also minimizes SCL.

Graf's theorem then implies the class means form a centered simplex ETF. Thus the argmin sets of $L_{\text{SCL}}$ and $L_{\text{proto}}$ coincide (up to rotation and label permutation).

$$\square$$

### A.3 Implementation Details

Experiments are conducted on four standard image classification datasets: *CIFAR10, CIFAR100, ImageNet-100, and ImageNet1K*, following common representation learning benchmarking practices (Khosla et al., 2020; Markou et al., 2024; Wang et al., 2021; Yeh et al., 2022). We use ResNet50 for ImageNet-100/ImageNet1K and ResNet18 for CIFAR10/CIFAR100. All models are trained using SGD optimizer for 500 epochs on ImageNet1K (temperature 0.1) and ImageNet-100 (batch size 1024, temperature 0.2) and 1000 epochs on CIFAR10/CIFAR100. For ImageNet1k in order to enable fair comparison we report for each method its best accuracy for training with batch size 2048, 4096, and 8192. For CIFAR10/100 we set the batch size to 512 and evaluate all 11 temperatures in the set [0.07, 0.1, 0.2, 0.3, 0.4, 0.5, 0.6, 0.7, 0.8, 0.9, 1]. In Table 1 and Table 2 we report for each method the best performing temperature. For Supervised Contrastive Learning we perform the linear probing phase for the typical 90 epochs.

### A.3.1 Classifier Learning Methods (CE, NormFace, NTCE, NONL)

For the family of classifier learning methods, we employ the following hyperparameters across datasets:

**CIFAR10/CIFAR100.** Models are trained for 1000 epochs with batch size 512. We use SGD optimizer with momentum 0.9, weight decay $10^{-4}$, and initial learning rate 0.2. The learning rate follows a cosine annealing schedule throughout training, decaying to a minimum value of $\eta_{\min} = \eta_0 \times 0.1^3$ where $\eta_0$ is the initial learning rate. Data augmentation consists of RandomResizedCrop with scale (0.2, 1.0), RandomHorizontalFlip, and standard normalization with dataset-specific mean and standard deviation values.

**ImageNet-100.** ResNet50 models are trained for 500 epochs with batch size 1024 (256 per GPU with 4 GPUs). We employ SGD optimizer with momentum 0.9, weight decay $10^{-4}$, and initial learning rate 0.1, which is automatically scaled based on the total batch size. We use cosine annealing scheduler with 10 epochs of linear warmup from 0.01 to the target learning rate. After warmup, the learning rate follows a cosine decay to $\eta_{\min} = \eta_0 \times 0.1^3$. Synchronized BatchNorm is enabled across GPUs. Data augmentation includes RandomResizedCrop(224) with scale (0.2, 1.0), RandomHorizontalFlip, and standard ImageNet normalization.

**ImageNet1K.** ResNet50 models are trained for 500 epochs with batch size 2048 (256 per GPU with 8 GPUs). Hyperparameters follow the same configuration as ImageNet-100, with SGD optimizer (momentum 0.9, weight decay $10^{-4}$), initial learning rate 0.1 with automatic scaling based on batch size. We apply 10 epochs of linear warmup followed by cosine annealing to $\eta_{\min} = \eta_0 \times 0.1^3$. Data augmentation and normalization follow ImageNet-100 settings.

### A.3.2 Supervised Contrastive Learning

For supervised contrastive methods, we implement a two-phase training procedure:

**Phase 1: Contrastive Training.**

**CIFAR10/CIFAR100:** Models are trained for 1000 epochs with batch size 512. SGD optimizer is used with momentum 0.9, weight decay $10^{-4}$, and initial learning rate 0.05. The learning rate follows cosine annealing schedule throughout training, decaying to $\eta_{\min} = \eta_0 \times 0.1^3$. We use extensive data augmentation including RandomResizedCrop with scale (0.2, 1.0), RandomHorizontalFlip, ColorJitter(0.4, 0.4, 0.4, 0.1) with probability 0.8, and RandomGrayscale with probability 0.2. Each image generates two augmented views for contrastive learning.

**ImageNet-100:** ResNet50 encoder with 128-dimensional projection head is trained for 500 epochs with batch size 1024. We use SGD optimizer with momentum 0.9, weight decay $10^{-4}$, and base learning rate 0.8 (automatically scaled by batch size). Learning rate follows cosine annealing with 10 epochs linear warmup from 0.01, then decays following a cosine schedule to $\eta_{\min} = \eta_0 \times 0.1^3$. Data augmentation extends CIFAR settings with the addition of Gaussian blur for ImageNet scale images.

**ImageNet1K:** Training spans 500 epochs with batch size 2048 using the same optimizer configuration as ImageNet-100. Base learning rate is set to 0.1 with automatic scaling. We employ cosine annealing with 5 epochs warmup from 0.01, followed by cosine decay to $\eta_{\min} = \eta_0 \times 0.1^3$. The same augmentation pipeline as ImageNet-100 is used.

**Phase 2: Linear Evaluation.** For all datasets, we freeze the learned encoder and train a linear classifier on top of the representations:

**CIFAR10/CIFAR100:** Linear classifier is trained for 100 epochs using SGD with learning rate 5.0, momentum 0.9, and zero weight decay. Learning rate is decayed by factor 0.2 at epochs 60, 75, 90 using a step scheduler.

**ImageNet-100:** Linear evaluation runs for 90 epochs with SGD optimizer, learning rate 2.0, momentum 0.9, and zero weight decay. Learning rate decay by factor 0.2 occurs at epochs 30, 60, 80 using a step scheduler.

**ImageNet1K:** Linear classifier training spans 90 epochs with SGD, learning rate 0.8, momentum 0.9, and zero weight decay. The same step decay schedule as ImageNet-100 is applied.

### A.3.3 Additional Implementation Details

For distributed training on ImageNet datasets, we employ DistributedDataParallel with one process per GPU. Random seed is fixed at 42 for reproducibility. The cosine annealing scheduler is implemented following the standard formulation: $\eta_t = \eta_{\min} + \frac{1}{2}(\eta_0 - \eta_{\min})(1 + \cos(\frac{\pi t}{T}))$, where $t$ is the current epoch and $T$ is the total number of epochs. For experiments with warmup, the warmup period linearly interpolates from the warmup starting learning rate to the initial learning rate before transitioning to cosine annealing. Temperature parameter $\tau$ is searched over the range [0.07, 0.1, 0.2, ..., 1.0] for CIFAR experiments, while ImageNet experiments use the optimal temperature found through preliminary experiments (0.1 for supervised contrastive, 0.2 for classifier learning methods). All models use standard weight initialization and no additional regularization beyond weight decay.

### A.4 Extra Ablation Studies

### A.4.1 Role of the Projection Head

Table 7: **Contrastive Learning Results - Without Projection Head.** Performance comparison across different classifier learning approaches without projection head.

| Classifier Learning | Loss | CIFAR-10 | CIFAR-100 | ImageNet-100 | ImageNet-1K |
|---|---|---|---|---|---|
| Linear Probing | SCL | 95 | 70.6 | 84.1 | 71 |
| Normalized Linear Probing | SCL | 95 | 71.4 | 84.3 | 72.1 |
| Fixed Prototypes | SCL | 95 | 71.4 | 84.7 | 70.1 |

In Table 7 we demonstrate the importance of the projection head in contrastive training. Across three datasets, except on the relatively simple CIFAR-10 benchmark, removing the head consistently reduces accuracy by more than 2 points. At first glance, one might expect the opposite: discarding the head should let the loss act directly on the final encoder embeddings on the unit hypersphere. We hypothesize that the projection head helps primarily by imposing a beneficial dimensionality bottleneck. With ResNet-50, the encoder's representation is 2048-dimensional, whereas the projection head maps it to 128 dimensions. For a $K$-class problem (e.g., $K = 100$), the ideal equiangular tight frame (ETF) geometry lives in a $(K - 1)$-dimensional subspace. Encouraging embeddings to adopt this structure is plausibly easier in a 128-dimensional space than in a 2048-dimensional one, where the optimizer has many more irrelevant directions to explore.

### A.4.2 Effective Hyperparameter ranges

Normalized softmax losses introduce too hyperparameters that originate from contrastive learning (i) temperature, and (ii) need for larger batch size. Here we test whether and how these haperparameters affect the downstream performance.

We conduct hyperparameter optimization experiments on CIFAR-10 and CIFAR-100, evaluating all combinations of 11 temperatures in the range [0.07, 1] and 7 batch sizes in the range [32, 2048]. The results show that different contrastive learning methods exhibit distinct optimal hyperparameter regions with minimal overlap in their peak performance zones across both datasets.

In Figure 1 we can see that normalized softmax losses exhibit the same behavior in terms of downstream performance compared to self-supervised contrastive learning (Chen et al., 2020), which means that there are trustable goto to setups for instance $\tau = \{0.1, 0.2\}$ for small to medium number of classes datasets and $\tau = \{0.07, 0.1\}$ for large. For that reason normalized softmax methods despite introducing extra hyperparameters, this is not a problem in practice

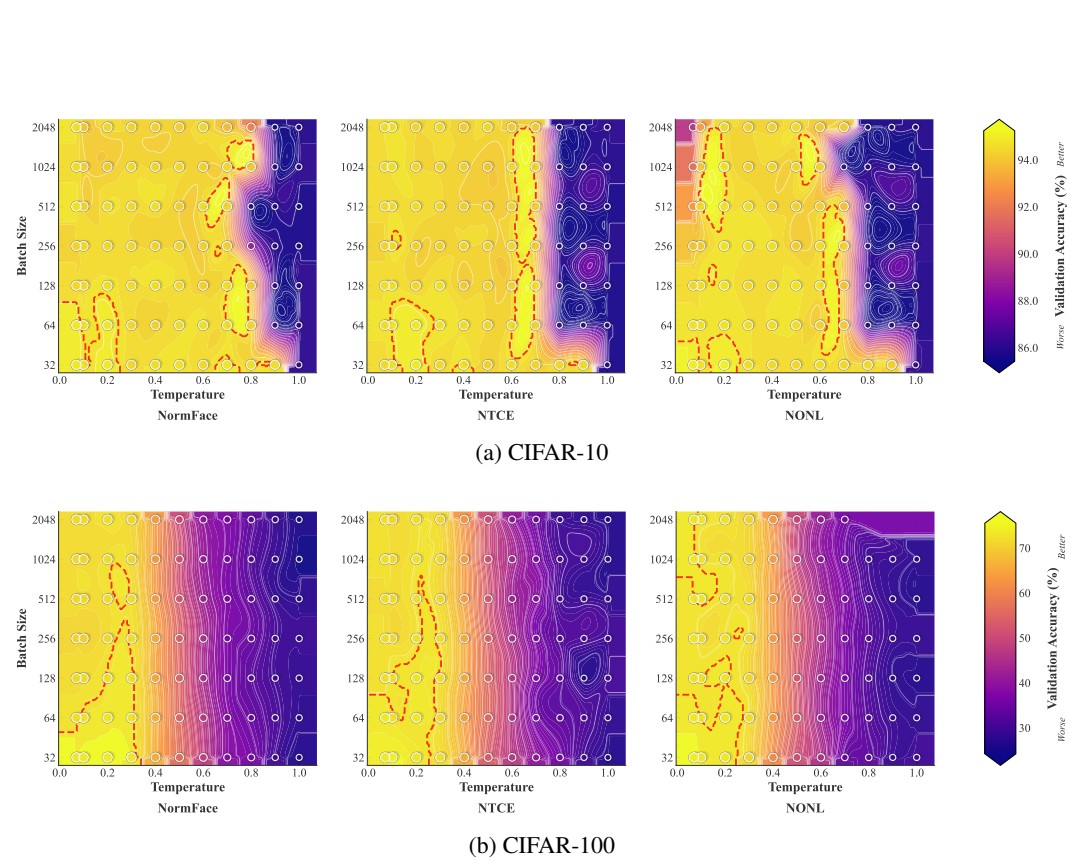

(a) CIFAR-10

(b) CIFAR-100

Figure 1: **Validation Accuracy (%) Phase Diagrams.** Classification accuracy on validation set. Higher values indicate better generalization performance. Each subplot shows the performance landscape across temperature and batch size hyperparameters for different loss functions: Norm-Face, NTCE, and NONL. Brighter regions indicate superior performance. White contour lines indicate iso-performance curves for detailed analysis. Red dashed contours highlight optimal parameter regions (top 10% performance). Scatter points represent individual experimental runs with performance-based sizing. Each dataset uses its own optimal colorbar range. Results originate from grid runs across temperature values in [0.07, 1.0] and batch sizes in 32, 64, 128, 256, 512, 1024, 2048.

| Loss | 2048 | 4096 | 8192 |
|------|------|------|------|
| CE | 75.4 | 75.4 | 75.1 |
| NORMFACE | 75.6 | 76.4 | 76.3 |
| NTCE (ours) | **76.0** | **76.7** | **76.7** |
| NONL (ours) | 75.0 | 76.2 | 76.5 |

Table 8: Top-1 accuracy (%) on ImageNet-1K for different batch sizes and loss functions.

Table 9: ImageNet-100 top-1 accuracy (%) for different backbones. Best results per column are highlighted in **green**. The last row reports the relative improvement of NONL over CE.

| Method | ResNet-50 | ResNet-101 | ResNet-152 | Mean |
|--------|-----------|------------|------------|------|
| CE | 84.4 | 85.3 | 85.5 | **85.1** |
| NormFace | 84.4 | 85.4 | 85.6 | **85.1** |
| NTCE | 84.7 | 85.4 | 85.4 | **85.2** |
| NONL | **84.9** | **85.5** | **85.8** | **85.4** |
| $\Delta$(NONL$-$CE) | +0.6% | +0.2% | +0.4% | **+0.4%** |

### A.4.3 EFFECTIVE HYPERPARAMETER RANGES

Normalized softmax losses introduce two hyperparameters inherited from contrastive learning: the temperature $\tau$, which controls the sharpness of the similarity distribution, and the need for larger batch size $B$, which governs the number of in-batch negatives. We assess their impact by grid–searching $\tau \in [0.07, 1.0]$ (11 values) and $B \in \{32, 64, 128, 256, 512, 1024, 2048\}$ on CIFAR-10 and CIFAR-100 with NormFace, NTCE, and NONL.

Figure 1 shows consistent "sweet spots" across methods: accuracy forms a pronounced band at moderate temperatures, with performance degrading for overly large $\tau$ and, to a lesser extent, for very small $\tau$. The location of this band shifts toward slightly smaller temperatures as the number of classes increases (CIFAR-100 vs. CIFAR-10), mirroring observations in self-supervised contrastive learning (Chen et al., 2020). Within the effective temperature range, performance is comparatively insensitive to $B$, yielding a broad plateau over batch sizes—large batches can help, but are not strictly required.

In practice, these trends provide the same reliable defaults as in self-supervised contrastive learning (Chen et al., 2020): $\tau \in \{0.1, 0.2\}$ works well for small- to medium-class datasets, while $\tau \in \{0.07, 0.1\}$ is preferable for larger-class settings. Thus, although normalized softmax losses expose additional hyperparameters, their effective ranges are narrow and stable, so a small amount of tuning (or even these defaults) is typically sufficient to reach near-peak accuracy.

### A.5 NEED FOR LARGE BATCH SIZE

### A.6 APPLICABILITY TO LARGER ARCHITECTURES

### A.7 TRAINING DYNAMICS

In Figure 2 the training dynamics are denomstrated. While cross-entropy (CE) achieves perfect training accuracy, it fails to reach neural collapse geometry, plateauing at suboptimal metric values. CE's accuracy improvements appear to *derive solely from magnitude and bias adjustments* rather than geometric reorganization. In contrast, our methods *simultaneously optimize all NC metrics throughout training*, converging to proper NC geometry while maintaining optimal accuracy.

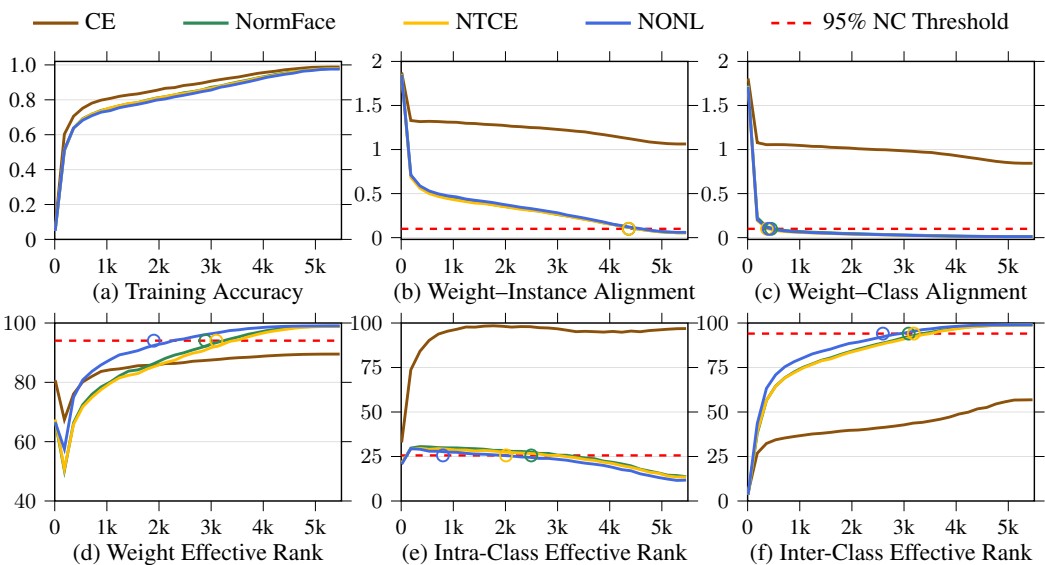

Figure 2: **NC convergence on CIFAR-100.** Six metrics vs. training iterations; red dashed lines mark the 95% NC threshold and circles denote each method's convergence.

