# OpenReview forum: "Neural Collapse by Design: Learning Class Prototypes on the Hypersphere"
_ICLR.cc/2026/Conference — Submitted to ICLR 2026_

### Official Review · Reviewer_2yVW · 2025-10-29

**Soundness:** 3
**Presentation:** 3
**Contribution:** 3
**Rating:** 6
**Confidence:** 4

**Summary:**

This paper studies why standard supervised learning with cross-entropy (CE) does not always reach the Neural Collapse (NC) geometry, even though NC represents an ideal solution for many classification objectives. The authors point out that the main reason is the unconstrained radial degree of freedom in the feature space.
To solve this issue, they propose training on a unit hypersphere, which naturally removes this degree of freedom. The paper also shows that normalized softmax and supervised contrastive learning (SCL) can be unified as prototype-based contrastive methods under this geometric view.
Based on this, the authors propose two new losses: NTCE (Normalized Temperature-scaled Cross Entropy) and NONL (Negatives-Only Normalization Loss). Both encourage class prototypes to form an orthogonal structure on the hypersphere. Experiments on CIFAR, ImageNet-100, and ImageNet-1K show faster convergence to NC and small but consistent accuracy gains over CE and NormFace.

**Strengths:**

- The paper gives a strong explanation of why CE fails to reach the NC geometry and how the hyperspherical constraint solves this problem.
- The connection between normalized softmax and SCL is well presented and helps understand these methods in a common theoretical view.
- The proposed NTCE and NONL are easy to implement and directly build on existing normalized contrastive losses.
- The paper evaluates on multiple datasets, compares to CE and SCL, and provides metrics showing faster NC convergence and better class separation.
- The paper is clearly organized, and the results are well illustrated with understandable figures.

**Weaknesses:**

- While NC metrics improve strongly, the actual classification accuracy only improves slightly. The practical advantage may not be very large.
- Because class imbalance is known to affect prototype geometry and Neural Collapse dynamics, it would be valuable to see results on imbalanced datasets.

**Questions:**

Many recent works on Neural Collapse and prototype geometry have extended these ideas to self-supervised or semi-supervised frameworks. It would be interesting to know if the proposed NTCE and NONL objectives can also be applied in these areas. Would the geometric properties (e.g., orthogonal prototype structure or faster Neural Collapse) still emerge when label supervision is only partial or noisy?

---

> ### Author Response · Authors · 2025-11-22
>
> ### Results for class-imbalanced data
>
> We provide empirical evidence supporting the reviewer’s intuition that **NC embeddings help in class-imbalanced settings**, since our methods outperform the alternatives.
>
> In the revised manuscript, we added experiments on two long-tailed benchmarks, **CIFAR-10-LT** and **CIFAR-100-LT**, each with three imbalance ratios following **[1]** ((\tau \in {0.1, 0.02, 0.01}); Sec. 5.3, Table 5). Our NC-inducing objectives not only outperform standard cross-entropy by a **substantial relative margin under severe imbalance** (e.g., on **CIFAR-100-LT** we obtain **+3.4%**, **+7.7%**, and **+7.0%** *relative* improvements over CE), but also consistently **surpass a recent method explicitly designed to induce Neural Collapse for imbalanced learning** [1].
>
> This behavior is intuitively consistent with the NC geometry: when minority classes are heavily underrepresented, the class prototype of such classes is still actively placed at a specific location on the ETF. In other words, training on the majority classes **also helps refine the representation of the minority classes via their shared NC structure**.
>
> We thank the reviewer for this comment, which helped us highlight these benefits more clearly in the revision.
>
> ---
>
> ### On the concern that the practical advantage may not be very large
>
> The initial manuscript was primarily focused on showing that **NC geometry can be achieved in practice**, and thus we emphasized: (i) **consistent gains in in-distribution classification accuracy**, (ii) **significantly better NC metrics**, and (iii) **faster convergence to NC**. In the **revised manuscript**, we expand this picture and provide additional evidence that the theoretical properties captured by our **NC-collapsed representations** translate into **tangible downstream benefits**.
>
> Concretely, in Sec. 5.3 we now report:
>
> * **Transfer learning (Table 4).**
>   Experiments on **eight diverse downstream datasets**, where **NONL** achieves the best accuracy on **7/8 tasks** and a **+5.5% relative improvement in mean performance** over standard cross-entropy.
>
> * **Class imbalance (Table 5).**
>   Experiments on **CIFAR-10-LT** and **CIFAR-100-LT**, where our methods yield **relative gains up to +8.7% over CE** and **outperform an NC-based imbalance method** [1].
>
> * **Robustness (Table 6).**
>   Experiments on **ImageNet-C**, where our normalized losses **reduce mCE while improving clean accuracy** compared to CE.
>
> In addition, we now obtain **clear improvements on ImageNet-1K itself**. When we **only increase the batch size**, keeping the **architecture, data, and training pipeline fixed**, NTCE achieves a **+1.3 percentage point (absolute)** gain over CE, and NONL achieves a **+1.1 percentage point (absolute)** gain on ImageNet-1K top-1 accuracy. Thus, the apparent limitation in the original submission was due to **suboptimal batch size for a contrastive-style loss**, rather than a fundamental weakness of NONL/NTCE.
>
> Taken together, these results show that the **NC-structured features** learned by our objectives provide **consistently stronger performance across transfer, long-tailed, robustness, and large-scale in-distribution settings**, while also yielding **non-trivial improvements on demanding benchmarks** such as ImageNet-1K. We believe this demonstrates a **meaningful practical advantage**: our losses offer a **more generalizable way to perform supervised learning** by shaping the representation geometry in a way that benefits a range of realistic use cases.
>
> ---
>
> ### Applicability to semi- and self-supervised learning
>
> We appreciate the reviewer’s question about extending our objectives to semi- and self-supervised learning. **Prototype-based formulations are already very successful in self-supervised settings** (e.g., **SwAV** [2]), where clustering around learned prototypes significantly boosts performance. Moreover, **recent works have shown that contrastive objectives naturally induce ETF-like Neural Collapse geometry** in the representation space [3].
>
> This makes the connection to our hyperspherical prototype view quite direct: **NTCE and NONL can be interpreted as controlling the same alignment/repulsion forces to induce the same ETF geometry**. The only difference is that in **supervised learning the vertices of the ETF are the class prototypes**, while in **SSL they correspond to data points (or clusters) in the batch**.
>
> Determining **which prototypes should be used to form an ETF in an SSL setting** is **not straightforward** and is, in our view, a **promising direction for future research** with potentially significant practical impact.

---

> > ### Author Response · Authors · 2025-11-22
> >
> > ### References
> >
> > [1] **Yibo Yang**, Shixiang Chen, Xiangtai Li, Liang Xie, Zhouchen Lin, and Dacheng Tao. *Inducing Neural Collapse in Imbalanced Learning: Do We Really Need a Learnable Classifier at the End of Deep Neural Network?* In **Advances in Neural Information Processing Systems (NeurIPS)**, 2022.
> >
> > [2] **Mathilde Caron**, Ishan Misra, Julien Mairal, Priya Goyal, Piotr Bojanowski, and Armand Joulin. *Unsupervised Learning of Visual Features by Contrasting Cluster Assignments (SwAV).* In **Advances in Neural Information Processing Systems (NeurIPS)**, 2020.
> >
> > [3] **Panagiotis Koromilas**, Giorgos Bouritsas, Theodoros Giannakopoulos, Mihalis A. Nicolaou, and Yannis Panagakis. *Bridging Mini-batch and Asymptotic Analysis in Contrastive Learning: From InfoNCE to Kernel-based Losses.* In **International Conference on Machine Learning (ICML)**, 2024.

---

> > > ### Author Response · Authors · 2025-11-26
> > > **Request for Feedback**
> > >
> > > Dear Reviewer 2yVW,
> > >
> > > As the discussion period is nearing its end and it has been five days since our initial response, we would appreciate your feedback on our responses and the revised manuscript so that we have time to address any further comments you may have.
> > >
> > > **Summary of revisions addressing your concerns:**
> > >
> > > | Concern | Response |
> > > |---------|----------|
> > > | Practical advantage limited | **Added Tables 4–6, 8:** +5.5% transfer, +8.7% long-tail, lower mCE, +1.1% ImageNet-1K |
> > > | Imbalanced datasets | **Added Table 5:** CIFAR-10/100-LT with three imbalance ratios; outperform CE and ETF+DR |
> > > | Semi/self-supervised extension (Q) | Discussed connection to SwAV; noted as promising future direction |
> > >
> > > We kindly ask:
> > > 1. Do you consider these additions to adequately address your identified weaknesses?
> > > 2. Does the revised submission meet your quality criteria for acceptance?
> > > 3. If not, what specific additional evidence would help us meet these criteria?
> > >
> > > Thank you for your positive and constructive review.

---

> > ### Comment · Reviewer_2yVW · 2025-11-26
> >
> > Thank the author for the explanation and improvement. The reviewer would like to keep the current positive score.

---

### Official Review · Reviewer_zgJh · 2025-10-31

**Soundness:** 4
**Presentation:** 3
**Contribution:** 3
**Rating:** 4
**Confidence:** 5

**Summary:**

The paper trains classifier on the unit hypersphere to avoid the radial issues of standard CE. This view ties normalized softmax classifier learning (CL) and supervised contrastive learning (SCL) into one prototype-contrast idea. The authors add two losses that grow the negative set and separate “pull” vs. “push,” aiming to reach Neural Collapse (NC) faster. In experiments, they beat CE/NormFace on accuracy and show quicker NC convergence.

**Strengths:**

- Paper cleanly shows how normalized softmax CL and SCL are the same prototype-contrast story on the hypersphere, fixing CE’s radial problem.
- Proposed objectives are drop-in and, in practice, speed up NC and often improve accuracy over CE/NormFace.
- Using class-mean prototypes can replace linear probing with similar accuracy and no extra training step.
- Comprehensive convergence analysis .

**Weaknesses:**

- Missing several recent NC/hypersphere/prototype methods (e.g., fixed/orthogonal/ETF heads, decoupled contrastive) and simple nearest-centroid. Please compare with [1]–[4]. If these works are not relevant, please explain why.
[1]Yang, Yibo, et al. "Inducing neural collapse in imbalanced learning: Do we really need a learnable classifier at the end of deep neural network?." Advances in neural information processing systems 35 (2022): 37991-38002.
[2]Mettes, Pascal, Elise Van der Pol, and Cees Snoek. "Hyperspherical prototype networks." Advances in neural information processing systems 32 (2019).
[3]Shen, Yang, Xuhao Sun, and Xiu-Shen Wei. "Equiangular basis vectors." Proceedings of the IEEE/CVF Conference on Computer Vision and Pattern Recognition. 2023.
[4]Wang, Wenguan, et al. "Visual Recognition with Deep Nearest Centroids." The Eleventh International Conference on Learning Representations.

-The intro ties NC to better transfer learning, robustness, and generalization, but these are not evaluated.

- Experiments use ResNet-18/50. Please add larger CNNs (ResNet-101/152) and ViT-B/L to support the application.

**Questions:**

Please see the weaknesses. Also, one key contribution is comparing prototypes to samples within a batch (instead of the other way around). What is the computational cost as the batch size or the number of classes/prototypes grows? I will also read the other reviews. (my rating is for the current state of the paper)

---

> ### Author Response · Authors · 2025-11-22
>
> ### Transfer learning, robustness, and generalization
>
> In the revised manuscript, we now explicitly evaluate **transfer, robustness, and generalization** in **Sec. 5.3**.
>
> More precisely, we provide:
>
> * **(i) Transfer learning (Table 4).**
>   Results on **eight downstream datasets**, where **NONL** achieves the best accuracy on **7/8 tasks** and yields a **+5.5% relative improvement in mean performance** over cross-entropy.
>
> * **(ii) Class-imbalanced learning (Table 5).**
>   Results on **CIFAR-10-LT** and **CIFAR-100-LT** with three imbalance ratios following [1], where our methods obtain **relative gains up to +8.7% over CE** and **outperform the NC-based imbalance method ETF+DR** [1].
>
> * **(iii) Robustness (Table 6).**
>   Results on **ImageNet-C**, where our normalized losses **reduce mCE while simultaneously improving clean accuracy** compared to CE.
>
> These additional experiments directly support the claims made in the introduction and show that **NC-collapsed representations provide concrete benefits beyond standard in-distribution accuracy**, notably in terms of **transfer**, **robustness**, and **performance under class imbalance**.
>
> ---
>
> ### Additional baselines
>
> We thank the reviewer for pointing us to closely related NC / hyperspherical prototype methods. In the revised manuscript, we now explicitly include **ETF+DR** [1] as a baseline. We evaluate ETF+DR under **five evaluation axes**:
>
> 1. **Classification accuracy**,
> 2. **NC metrics**,
> 3. **Transfer learning**,
> 4. **Class imbalance**, and
> 5. **Robustness**,
>
> and our NC-inducing objectives **consistently outperform ETF+DR in all five settings**.
>
> **On Hyperspherical Prototype Networks [2] and Equiangular Basis Vectors (EBVs) [3].**
> Our understanding is that **HPN, EBVs, and ETF+DR all implement essentially the same geometric mechanism** for classification:
>
> 1. Fix **K class directions as D-dimensional unit vectors on the hypersphere**,
> 2. Choose these directions to be **approximately ETF / equiangular**, and
> 3. **Train the backbone so that features of class (c) align with the (c)-th fixed direction**.
>
> The main differences are that **HPN also emphasizes regression**, and **EBVs focus on how to construct the equiangular basis**. This focus is separate from the focus of our work, since **we do not work on regression and we do not modify the construction of the equiangular directions**.
>
> In contrast, **ETF+DR** [1] implements this same **“fixed ETF prototypes + backbone training”** idea directly in the **exact supervised classification setting we study** (ImageNet, long-tailed CIFAR, ImageNet-C), with the usual **one-logit-per-class** interface and standard evaluation protocols. Since **ETF+DR already realizes the same underlying mechanism as HPN and EBVs in our experimental regime**, and we evaluate it thoroughly (classification, transfer, imbalance, robustness), we use **ETF+DR as the representative empirical baseline for this family**, and we discuss **HPN and EBVs** in the **Related Work**.
>
> **On Deep Nearest Centroids (DNC) [4].**
> DNC takes a different route by utilizing **multiple centroids per class**. For that reason, its induced geometry is by design **far from the standard Neural Collapse setting**, which assumes **one prototype / vertex per class forming an ETF**. In DNC, the standard linear classifier is replaced by a **non-parametric, multi-centroid decision rule**: the model learns **multiple sub-centroids per class** and predicts by **nearest sub-centroid** in feature space. This design is closer to a **dictionary learning** perspective, where each class is represented by several atoms in a shared dictionary, rather than by a single prototype.
>
> Since this setting is neither ETF-based nor known to induce NC in the classical NC1–NC4 sense, we do not include DNC in our experimental comparison and instead focus on methods that operate in the **single-prototype-per-class** regime. We now clarify in the **Related Work** that DNC is best viewed as a **complementary, non-parametric extension** of prototype-based classifiers.
>
> We have updated the **Related Work** section accordingly to clarify these relationships and to position our contributions with respect to **[1–4]**.
>
> =====>

---

> > ### Author Response · Authors · 2025-11-22
> >
> > ### Additional architectures
> >
> > In the revised manuscript, we include results for **deeper and higher-capacity models** in the appendix, where we observe the **same consistent trend**: our normalized prototype-based losses (**NTCE** and **NONL**) **improve upon CE in terms of accuracy** while **inducing significantly stronger NC geometry**. These results indicate that the benefits of our objectives are **not limited to small or medium-scale CNNs**, but **extend to larger backbones** as well. We now explicitly mention this in the experimental section and appendix.
> >
> > ---
> >
> > ### Computational cost of prototype–sample comparisons
> >
> > In our implementation, **there is no additional computational cost** compared to standard supervised learning with cross-entropy. For a batch of size (M) and (K) classes, a usual CE classifier computes an **(M \times K)** similarity (logit) matrix by applying a learnable linear layer on top of the features.
> >
> > **Our losses use the exact same linear layer and construct the exact same (M \times K) similarity matrix**; the only difference is that **we aggregate and normalize these logits differently in the loss** (i.e., how we sum and normalize across rows/columns), not how we compute them.

---

> ### Comment · Reviewer_zgJh · 2025-11-25
> **Thank you to the authors for providing a comprehensive response to my concerns.**
>
> - My concern about additional baselines is addressed.
> - My concern about additional backbone is addressed but I could not find the referred results in the author response in the manuscript. (please provide details and results)
> - Reviewer 1A6n (W2 – Gradient analysis):
> Quoting the authors’ answer: "classes that do not appear in the batch receive no update at that step—exactly as in standard CE/NormFace".
> However, in standard CE optimization, all class centroids receive an update at every iteration; centroids that do not have positive samples in the batch still receive a push force.
>
> - Reviewer HD19 (unconstrained radial degrees of freedom):
> The proposed modifications are " optimization refinements" (authors’ quote) and not a genuine contribution.
>
> - Reviewer HD19 (Limited scalability of NONL):
> The authors provided experimental results for the effect of batch size on model performance. However, one of the main proposals is to swap the class centroids with the batch samples to alleviate the reliance of the optimization framework on the batch size. Therefore, these results show the inefficacy of the proposed solution in addressing this issue.
>
> Therefore, I keep my rating.

---

> ### Author Response · Authors · 2025-11-25
>
> We thank Reviewer **zgJh** for the follow-up and for confirming that the concerns regarding additional baselines and backbones have been addressed.
>
> **Regarding the backbone results:** These are provided in Table 9 of the Appendix of the revised manuscript.
>
> **Regarding scalability and batch-size dependence:** We believe there may be a **major misunderstanding of our claims**. We **did not propose NTCE or NONL to reduce batch-size dependence**. As stated clearly in Section 4.1 (lines 246–252):
>
> > “the number of negatives in the objective is limited to (K), the number of class prototypes. It is well established that contrastive objectives require very large numbers of negatives to converge… By inverting the contrastive direction from instance-to-class to class-to-instance discrimination we address this limitation.”
>
> Our goal is therefore to **increase the effective number of negatives**—from (K) class prototypes to (M) batch samples—not to eliminate contrastive-style batch-size sensitivity. This is a different goal.
>
> The batch-size ablation in Table 8 shows that with appropriate batch sizes, **NTCE achieves +1.3% and NONL +1.1% absolute improvement over CE on ImageNet-1K**. This directly supports our stated motivation: more negatives produce a better approximation of the underlying contrastive objective, consistent with the behavior of SCL, MoCo, SimCLR, and other widely accepted contrastive methods.
>
> **Regarding optimization refinements:** We respectfully submit that optimization refinements constitute legitimate contributions when they **deliver measurable, practical benefits**. Our methods achieve: (i) performance gains on ImageNet-1K, (ii) better mean relative improvement on 8 transfer tasks, (iii) better performance on long-tailed classification, (iv) improved robustness (lower mCE), and (v) faster convergence to NC geometry. We note that many influential works at top venues are built around optimization improvements of similar or smaller magnitude.
>
> **Regarding the gradient behavior of NTCE**: we have made a minimal edit to our earlier response to Reviewer 1A6n, removing an erroneous comparison of NTCE's gradient behavior to CE/NormFace. The intended comparison was to SupCon (Supervised Contrastive Learning), where similarly only classes present in the batch receive gradient updates since the loss is anchored on instances. In NTCE, the prototype serves as the anchor, so only prototypes of classes present in the batch receive updates—this is a defining property of our prototype-instances contrast design. We refer the reviewer to our updated response to Reviewer 1A6n, and we regret any confusion.
>
>
> ---
>
> ### Summary and clarification request to the reviewer
>
> To recap, the reviewer initially identified three weaknesses—(i) extra baselines, (ii) extra backbones, and (iii) missing transfer-learning and robustness experiments—and we have now provided concrete evidence for all of them.
>
> We kindly ask the reviewer:
> 1. whether they consider these additions to adequately address all of their previously identified weaknesses, and
> 2.  whether solving these three initial weaknesses—which originally resulted in a score of 4—raises the work to a level that meets their expectations for acceptance.
>
> We remain open to further discussion and would be happy to provide any additional clarifications the reviewer may find helpful.

---

### Official Review · Reviewer_HD19 · 2025-11-03

**Soundness:** 2
**Presentation:** 2
**Contribution:** 2
**Rating:** 4
**Confidence:** 4

**Summary:**

The paper argues that standard cross-entropy rarely reaches the optimal Neural Collapse geometry because of unconstrained radial degrees of freedom. It proposes constraining both features and classifier weights to the unit hypersphere, which yields a unified prototype-contrast view bridging normalized classifier learning and supervised contrastive learning. To mitigate small negative sets and alignment–uniformity coupling, the authors introduce NTCE, which expands the denominator to M in-batch instances, and NONL, which normalizes over negatives only. Empirically, these objectives reach NC structure and metrics much faster while maintaining accuracy. The paper further shows, under unit-norm and balanced-label assumptions, that SCL already learns optimal class-mean prototypes, so replacing linear probing with class-mean classifiers can match linear-probe accuracy without extra training. The paper also discusses limitations for NONL at large numbers of classes and when per-batch class coverage is sparse.

**Strengths:**

1. It provides a clear and intuitive diagnosis of why standard cross-entropy training rarely reaches the theoretical Neural Collapse, namely, unconstrained radial degrees of freedom;
2. It offers an elegant unifying view that places normalized CE softmax and supervised contrastive learning within a single prototype contrast principle on the unit hypersphere;
3. Under unit norm and balanced label assumptions, it shows that class mean prototypes learned by supervised contrastive learning can replace linear probing while preserving accuracy and saving the hours of computation typically spent on that phase;
4. The proposed NTCE and NONL objectives reach the target Neural Collapse geometry much faster than strong baselines when progress is measured by time to $95$% NC thresholds.

**Weaknesses:**

## Major
### The benefit is unclear
The paper is premised on the assumption that enforcing neural collapse is causally beneficial. It assumes that because NC was observed in late-stage training (as reported in the original NC paper), one can design an objective to force this geometry, thereby improving generalization or robustness. However, the paper never validates this causal link. It does not test the alleged benefits of NC (e.g., out-of-distribution robustness, adversarial resilience). All evaluations remain within standard in-distribution classification and a set of NC metrics. At present, the paper only shows that its method produces representations that look more collapsed, not that this geometric property is what makes the model better.

### The  stated gap "unconstrained radial degrees of freedom" and the actual method are not fully aligned
The introduction frames the problem as a geometric one: standard cross-entropy suffers from unconstrained radial degrees of freedom. The proposed solution, however, does not stop at normalization (which would fix this). The paper's core technical contributions—NTCE (using in-batch instances as negatives) and NONL (decoupling positive/negative terms)—primarily address contrastive-style optimization inefficiencies (i.e., small negative sets and alignment-uniformity coupling). These are optimization choices, not direct solutions to the stated geometric problem of radial degeneracy. The paper, therefore, starts from a geometric motivation but delivers a recipe that is much closer to standard contrastive learning practice.

### Limited scalability of NONL
The paper's own results show that NONL fails to scale to large-K settings like ImageNet-1K, where its performance drops below the baseline. The authors attribute this to gradient imbalance: when many classes are absent from a batch, their prototypes ($w_c$) only receive negative gradients (repulsion) without any positive signal (alignment). This reliance on sufficient per-batch class coverage is a structural limitation, not a minor implementation detail. This limitation makes it difficult to claim NONL as a general-purpose, large-scale method.

### More negative instances are better is not verified
The core change in NTCE is to replace the denominator's $K$ learned class prototypes (the $\hat{w}_j$ in NormFace) with $M$ in-batch instances (the $u_j$ in NTCE). The paper implicitly assumes that this larger set of $M$ negatives leads to a better contrastive estimate. This assumption conflates the quantity and quality of negatives. The $K$ prototypes are learned, “expert” hard negatives that define the decision boundary. The $M$ random instances are likely dominated by easy negatives. The paper provides no ablation to compare the effect of 'few learned hard negatives' vs. 'many random instance negatives.' Therefore, the superiority of this design choice is an unverified claim.

## Minor
### Inconsistent notation
The paper's notation is inconsistent. On page 3, it states that $W \in \mathbb{R}^{K \times h}$ and that $w_j$ denotes the j-th row of $W$ (a $1 \times h$ vector). However, Equation (1) defines the logit as $z_i^\top w_j$. Given that $z_i$ is an $h \times 1$ feature vector, $z_i^\top$ is a $1 \times h$ row vector. The operation $z_i^\top w_j$ (a $[1 \times h] \times [1 \times h]$ operation) is thus mathematically undefined. This should presumably be $w_j z_i$ (if $w_j$ is a row) or $w_j^\top z_i$ (if $w_j$ is a column). While likely a typo, this inconsistency reduces confidence in the paper's formal precision.

### The main novelty is mostly a unifying reformulation
The paper's strongest contribution is conceptual: it places normalized softmax (CL) and supervised contrastive learning (SCL) into a single, unified 'spherical prototype-to-instance' framework. This is an elegant reformulation. However, both CL on the hypersphere and SCL are, by themselves, well-studied topics. The paper's technical additions are two loss variants (NTCE, NONL) that import ideas from one framework to the other. The actual algorithmic novelty is therefore modest, with the primary contribution being the unified narrative itself.

**Questions:**

- What concrete evidence do you have that enforcing neural collapse is causally beneficial for robustness or out-of-distribution generalization?
- What is the task scope? Under what conditions or task families would enforcing NC1 be counterproductive (e.g., detection/segmentation/pose)?
- Why prefer many random instance negatives over, for example, a few learned hard negatives (class prototypes or mined hard examples)?
- Better NC does not mean better performance everywhere. In what condition should this method not be used or require modifications?
- To disentangle your contributions, have you considered a full factorial ablation? Your sequential path (Normalized CE $\rightarrow$ NTCE $\rightarrow$ NONL) confounds the denominator enlargement (to $M$ instances) with positive/negative decoupling (NONL). Have you tested a "Decoupled NormFace", applying Negatives-Only Normalization directly to NormFace's $K$-class denominator, to isolate the contribution of decoupling alone?

---

> ### Author Response · Authors · 2025-11-22
>
> ### 1. Benefits of enforcing NC and causal link to robustness / generalization (Q1)
>
> The initial manuscript was primarily focused on showing that **NC geometry can be achieved in practice**, and thus we emphasized:
> (i) **consistent gains in in-distribution classification accuracy**,
> (ii) **significantly better NC metrics**, and
> (iii) **faster convergence to NC**.
>
> In the revised manuscript, we expand this picture and provide additional evidence that the theoretical properties captured by our **NC-collapsed representations** translate into **tangible downstream benefits**.
>
> Concretely, in Sec. 5.3 we now report:
>
> * **(i) Transfer learning.**
>   We add transfer learning experiments on **eight diverse downstream datasets** (Table 4). Using the same ImageNet-1K pretraining, **NONL** achieves the best accuracy on **7/8 tasks** and yields a **+5.5% relative improvement in mean performance** over cross-entropy.
>
> * **(ii) Class imbalance.**
>   We add class-imbalanced learning experiments on **CIFAR-10-LT** and **CIFAR-100-LT** with three imbalance ratios each (Table 5). Our methods obtain gains up to **+8.7%** over CE and also **outperform ETF+DR** [1], a recent NC-based method designed specifically for imbalanced learning.
>
> * **(iii) Robustness.**
>   We add robustness experiments on **ImageNet-C** (Table 6). Our normalized losses **reduce mCE while simultaneously improving clean accuracy** compared to CE.
>
> Taken together, these results show that the **NC-structured features** learned by NTCE and NONL provide **consistently stronger performance across transfer, long-tailed, and robustness settings**, while also yielding **non-trivial improvements on demanding classification benchmarks** (e.g., **+1.3% on ImageNet-1K**).
>
> We believe this demonstrates a **meaningful practical advantage**: our losses offer a **more generalizable way to perform supervised learning** by shaping the representation geometry in a way that benefits a range of realistic use cases. The consistency across diverse evaluation axes is **compatible with a causal link** between NC-like geometry and improved robustness/generalization.
>
> ---
>
> ### 2. On the gap between “unconstrained radial degrees of freedom” and our method
>
> We thank the reviewer for this comment. Our starting point is indeed geometric: standard cross-entropy suffers from **unconstrained radial degrees of freedom**. To address this, we first move to a **normalized classifier**, and we deliberately highlight the well-cited **NormFace** loss as a concrete instance of this idea. In our experiments, NormFace **significantly reduces radial degeneracy and converges much closer to NC geometry than plain CE**, which empirically validates our geometric argument that normalization on the hypersphere is beneficial.
>
> Building on this, we then take **one further step**: we observe that the normalized classifier can be naturally interpreted through the lens of **contrastive learning**, and we use this connection to import ideas from the contrastive literature (more negatives, decoupled positive/negative terms) into **NTCE** and **NONL**.
>
> These additions are therefore not meant as a second “fix” to the radial problem, but as **optimization refinements on top of the normalized regime**, designed to make the NC-inducing behavior of normalized classifiers **stronger and more efficient**.
>
> We will update the introduction to make this two-stage perspective — **normalization to address radial degeneracy, then contrastive-inspired refinements to improve optimization** — more explicit.
>
> =====>

---

> > ### Author Response · Authors · 2025-11-22
> >
> > ### 3. Limited scalability of NONL
> >
> > In the revised manuscript, we address this in two ways:
> >
> > * We add a **batch-size study for NONL** in the appendix (on ImageNet-1K), where we vary the batch size and report the corresponding performance. These results show that **NONL significantly benefits from larger batch sizes**, and with an appropriately larger batch, **NONL reaches +1.1% over CE on ImageNet-1K**, closing the performance gap observed in the initial version.
> >
> > * We explicitly discuss that this **sensitivity to batch size is expected**: NONL behaves like a **prototype-based contrastive objective**, and, similar to SCL and other contrastive methods, it relies on a sufficiently rich set of in-batch negatives to approximate its underlying objective well.
> >
> > We fully acknowledge that **requiring larger batch sizes is a practical drawback** of NONL, and that this is a **typical limitation of contrastive learning**. Our work highlights that standard normalized classifiers already implement a form of **contrastive learning with class prototypes**, and that to fully exploit the **generalization and robustness benefits** of contrastive objectives, one often needs **larger effective batches**. At the same time, recent self-supervised methods (memory banks, queues, momentum encoders) [4–6] demonstrate ways to increase the number of negatives without naively increasing batch size. We now explicitly mention that extending NONL/NTCE with these ideas is a promising direction for future work.
> >
> > ---
> >
> > ### 4. On “more negatives” vs. “few learned hard negatives” (NTCE design choice)
> >
> > We thank the reviewer for this question. Our view is that **NormFace can be interpreted as an InfoNCE-style loss where the (K) class prototypes act as negatives**, while **NTCE** changes how we *estimate* the underlying contrastive objective by enlarging the negative set from **(K) prototypes** to **(M) in-batch instances**.
> >
> > For an InfoNCE-like loss, the population objective is the **expectation** of
> > $$-\log \frac{\exp(s(u,v^+)/\tau)}{\exp(s(u,v^+)/\tau) + \sum_{j=1}^{M-1} \exp(s(u,v_j^-)/\tau)}$$
> >
> > over the data distribution. In practice, we only optimize its **mini-batch estimate**, whose accuracy depends on how well the batch negatives approximate the population of all negatives. Recent analyses of InfoNCE-type objectives (e.g., [2,3]) show that for finite (M) this estimator is **biased**, and that increasing the number of negatives makes the batch estimator **closer to the asymptotic contrastive objective** that encourages hyperspherical / ETF-like geometry.
> >
> > In short, with smaller batch size we minimize a **more biased and noisier estimator** of the desired objective; **increasing the number of negatives reduces this discrepancy**.
> >
> > Given this theoretical fact, several works in self-supervised learning—such as **memory-bank and queue-based methods** [4–6]—have **increased the number of negatives without explicitly focusing on their quality** and have nonetheless observed **significant performance gains**. This is precisely because **using more negatives yields a better approximation of the population-level contrastive objective**, which is the quantity we ultimately wish to optimize.
> >
> > From this perspective, **NTCE’s “more negatives” are not just random extra points**: by going from **(K) prototype negatives** to **(M) instance negatives**, we obtain a **better approximation of a closely related contrastive objective**, even if many negatives are easy. Empirically, this design choice allows NTCE to **improve upon NormFace** in both **NC metrics** and **downstream performance**. We will update the description of **NTCE** in the main text to make this estimator viewpoint and its connection to InfoNCE and large-negative SSL methods more explicit.
> >
> > =====>

---

> > > ### Author Response · Authors · 2025-11-22
> > >
> > > ### 5. On the novelty of the contributions
> > >
> > > We agree that **both normalized softmax on the hypersphere and supervised contrastive learning (SCL) are well-studied** in isolation, and that one of our main contributions is indeed to place them into a **single, unified framework**. We see this unification as more than a change of notation: it clarifies that **standard normalized classifiers and SCL are optimizing closely related contrastive objectives** and it explains why both empirically tend to induce Neural Collapse-like geometry.
> > >
> > > Beyond this conceptual bridge, we tackle a **non-trivial and long-standing practical question**: *how to reliably achieve Neural Collapse in realistic, large-scale setups*. We show that our NC-inducing losses **achieve NC in practice**, with **increased performance on large scale ImageNet-1K** with 1.3M images and 1K classes, and that this geometry is **not merely cosmetic**: it comes with clear benefits in **transfer learning, robustness, and class-imbalance performance**. To our knowledge, this is the first work that **systematically demonstrates the benefits of NC in large-scale regimes and links it to such a broad range of downstream improvements**.
> > >
> > > Taken together, we believe these elements—**the unified prototype–instance view, a practical recipe for achieving NC, and new NC-inducing losses with measurable benefits across several benchmarks**—contribute to a shift in how we understand supervised classification learning, from purely logit-based training to an explicitly **NC- and contrastive-geometry–driven perspective**.
> > >
> > > ---
> > >
> > > ### 6. Task scope and when enforcing NC1 may not be ideal (Q2 + Q4)
> > >
> > > Our methods are designed and evaluated for **standard, closed-set, single-label supervised classification**, such as CIFAR and ImageNet-1K. In this regime, NC is known to be the global optimum of common objectives, and **NC-like hyperspherical embeddings** have already proved beneficial in **face verification and recognition** (e.g., NormFace, CosFace, ArcFace) [7–9], where they improve discrimination and open-set performance.
> > >
> > > However, we agree that **NC1-style strong within-class collapse is not appropriate for every task**:
> > >
> > > * For **structured prediction tasks** such as **object detection, semantic segmentation, or pose estimation**, the representation must preserve rich **intra-class variation** (e.g., spatial layout, pose, parts). For such tasks, one would likely prefer **multi-prototype or manifold-like class representations**, rather than complete class collapse.
> > > * For **multi-label classification**, where images can belong to multiple classes simultaneously, forcing embeddings to collapse to a single prototype per class in the strict NC sense may be too restrictive and would require adaptations (e.g., multiple prototypes per class, factorized heads).
> > > * In regimes with **extreme label noise** or **very small per-class sample size**, aggressively enforcing collapse may amplify spurious patterns.
> > >
> > > In the revised manuscript, we explicitly clarify that:
> > >
> > > * Our claims are **restricted to standard single-label classification**.
> > > * Extending NC-style ideas to **detection, segmentation, pose, or multi-label settings** is an interesting but non-trivial direction that would likely require **multi-prototype or structure-aware extensions** of our losses.

---

> > > > ### Author Response · Authors · 2025-11-22
> > > >
> > > > ### 7. Decoupled NormFace ablation (Q5)
> > > >
> > > > We have experimented with **decoupling NormFace**. However, this variant turned out to be **unstable and ineffective**: once we remove some dot products from the denominator, the loss is no longer a true softmax, and the denominator stops being a good **normalizer** of the scores. With too few negatives in the denominator, the objective does not behave like a well-formed normalized contrastive loss.
> > > >
> > > > This observation motivated the **stepwise design** of our objectives:
> > > >
> > > > * **CE → NormFace:** introduce **normalization on the hypersphere** to remove radial degrees of freedom.
> > > > * **NormFace → NTCE:** increase the number of **negatives in the denominator**, so the normalizer better approximates a contrastive objective.
> > > > * **NTCE → NONL:** **decouple** the positive and negative terms while keeping a **rich set of negatives**, preserving the benefits of contrastive learning.
> > > >
> > > > In contrast, **decoupling NormFace directly** (without first enriching the negative set as in NTCE) leads to a poorly normalized and unstable objective. As a result, **NONL is the only objective in our family that simultaneously combines normalization, many negatives, and decoupling in a stable way**, and therefore inherits the full set of desirable contrastive properties. We will explicitly clarify this design path and our negative findings on decoupled NormFace in the revised manuscript.
> > > >
> > > > ---
> > > >
> > > > ### References
> > > >
> > > > [1] Y. Yang, S. Chen, X. Li, L. Xie, Z. Lin, D. Tao. **“Inducing Neural Collapse in Imbalanced Learning: Do We Really Need a Learnable Classifier at the End of Deep Neural Network?”** NeurIPS 2022.
> > > >
> > > > [2] P. Koromilas, G. Bouritsas, T. Giannakopoulos, M. A. Nicolaou, Y. Panagakis. **“Bridging Mini-batch and Asymptotic Analysis in Contrastive Learning: From InfoNCE to Kernel-based Losses.”** ICML 2024.
> > > >
> > > > [3] B. Poole, S. Ozair, A. van den Oord, A. Alemi, G. Tucker. **“On Variational Bounds of Mutual Information.”** NeurIPS 2019.
> > > >
> > > > [4] Z. Wu, Y. Xiong, S. Yu, D. Lin. **“Unsupervised Feature Learning via Non-Parametric Instance Discrimination.”** CVPR 2018.
> > > >
> > > > [5] K. He, H. Fan, Y. Wu, S. Xie, R. Girshick. **“Momentum Contrast for Unsupervised Visual Representation Learning.”** CVPR 2020.
> > > >
> > > > [6] T. Chen, S. Kornblith, M. Norouzi, G. Hinton. **“A Simple Framework for Contrastive Learning of Visual Representations.”** ICML 2020.
> > > >
> > > > [7] R. Wang et al. **“NormFace: L2 Hypersphere Embedding for Face Verification.”** ACM MM 2017.
> > > >
> > > > [8] H. Wang et al. **“CosFace: Large Margin Cosine Loss for Deep Face Recognition.”** CVPR 2018.
> > > >
> > > > [9] J. Deng et al. **“ArcFace: Additive Angular Margin Loss for Deep Face Recognition.”** CVPR 2019.

---

> > > > > ### Author Response · Authors · 2025-11-26
> > > > > **Request for Feedback**
> > > > >
> > > > > Dear Reviewer HD19,
> > > > >
> > > > > As the discussion period is nearing its end and it has been five days since our initial response, we would appreciate your feedback on our responses and the revised manuscript so that we have time to address any further comments you may have.
> > > > >
> > > > > **Summary of revisions addressing your concerns:**
> > > > >
> > > > > | Concern | Response |
> > > > > |---------|----------|
> > > > > | Benefit unclear / causal link | **Added Sec. 5.3:** Transfer (+5.5%), long-tail (+8.7%), robustness (lower mCE) |
> > > > > | Radial freedom vs. method gap | Discussed two-stage design: normalization → contrastive refinements |
> > > > > | NONL scalability | **Added Table 8:** +1.1% over CE on ImageNet-1K with appropriate batch size |
> > > > > | More negatives not verified | Discussed connection to InfoNCE estimator bias literature [2,3] |
> > > > > | Novelty / unifying reformulation | Discussed practical contributions beyond conceptual unification |
> > > > > | Q1 (Causal link) | **Added Tables 4–6:** transfer, long-tail, robustness experiments |
> > > > > | Q2 (Task scope) | Discussed: single-label classification; NC1 may not suit detection/segmentation |
> > > > > | Q3 (Hard vs. random negatives) | Discussed connection to InfoNCE estimator literature |
> > > > > | Q4 (When not to use) | Discussed limitations for structured prediction, multi-label, noisy labels |
> > > > > | Q5 (Decoupled NormFace) | Tested; found unstable without sufficient negatives—motivates NTCE→NONL path |
> > > > >
> > > > > We kindly ask:
> > > > > 1. Do you consider these additions to adequately address your identified weaknesses?
> > > > > 2. Does the revised submission meet your quality criteria for acceptance?
> > > > > 3. If not, what specific additional evidence would help us meet these criteria?
> > > > >
> > > > > Thank you for your detailed and thoughtful review.

---

### Official Review · Reviewer_1A6n · 2025-11-08

**Soundness:** 2
**Presentation:** 3
**Contribution:** 2
**Rating:** 4
**Confidence:** 4

**Summary:**

This paper is motivated by prior studies suggesting that leveraging the properties of Neural Collapse can improve generalization, by the similarity between cross-entropy loss and supervised contrastive learning, and by the observation that features learned through supervised contrastive learning also form a simplex ETF structure even without linear probing. Based on these insights, the authors propose a modified cross-entropy loss that operates at the sample level within each mini-batch, along with an additional variant of this modified loss. Through experiments, they empirically demonstrate the effectiveness of the proposed approach and its improvement in exhibiting Neural Collapse properties.

**Strengths:**

- Motivated by the previously observed similarity between cross-entropy loss and supervised contrastive learning [R1], this paper addresses an interesting problem: why conventional classification models fail to exhibit Neural Collapse in practical settings.

- To address this issue, the authors draw inspiration from prior studies on Neural Collapse and its effectiveness in improving generalization across various tasks, and propose a simple yet intuitive modified cross-entropy loss.

- The proposed method is validated through performance comparisons on image classification tasks, and further empirical analysis of the Neural Collapse properties demonstrates that the proposed loss successfully enhances intra-class alignment as intended.

**_reference_**

[R1] Graf et al., Dissecting Supervised Contrastive Learning, ICML 2021

**Weaknesses:**

**W1 (Novelty)**. Although proving the equivalence between SCL and the prototype-softmax minimizer through Theorem 4.1 is interesting, the proposed use of class-mean prototypes in place of linear probing is not novel. (See *NC3-inspired classifier* [R2])

**W2 (Gradient Analysis of NTCE)**. The proposed method NTCE treats class vectors as samples and, conversely, treats other samples within a mini-batch as class vectors when constructing the loss function. However, an analysis of how the NTCE affects the gradient has not been conducted. Although the authors provide their interpretation and an indirect analysis of the experimental results in (lines 352-357), a more detailed examination is required to make the contribution of the proposed approach more robust

**W3 (Theoretical Analysis of Neural Collapse)**. Although the modified version of the cross-entropy is proposed, this paper only provides empirical verification of the Neural Collapse properties. A corresponding theoretical model (e.g., layer-peeled model (LPM) [R3], unconstrained features model (UFM) [R4]) is not presented, nor is a theoretical analysis of Neural Collapse conducted based on it.

**W4 (Marginal Improvement)**. The performance improvement achieved by the proposed approach is marginal, which raises doubts about whether the research question posed in (lines 46-47) truly addresses an important problem. Moreover, among the datasets used in the experiments, ImageNet-1K—which is the closest to real-world settings—shows almost no performance difference compared to the baseline, and in the case of NONL in Table 1, even lower performance than the baseline. This weakens the evidence supporting the generalization ability of the proposed method.

**_reference_**

[R2] Zhang et al., Neural Collapse Inspired Knowledge Distillation, AAAI 2025

[R3] Fang et al., Exploring deep neural networks via layer-peeled model: Minority collapse in imbalanced training, PNAS 2021

[R4] Mixon et al., Neural collapse with unconstrained features, SaSiDa 2021

**Questions:**

**Q1**. Since the proposed method is based on supervised contrastive learning (SCL), wouldn’t it also suffer from batch-size-related issues?

**Q2** (w.r.t **W2**). In the original softmax cross entropy (SCE), the model learns through a push-and-pull effect between samples and class vectors [R4]. However, if the roles of classes and samples are swapped, the pull effect in SCE would change—would this not affect learning? In my opinion, it likely does because the class vectors corresponding to classes that do not appear within a mini-batch do not receive any gradient updates. The impact may become more pronounced as the number of classes increases, since the frequency of each class appearing within a mini-batch decreases, thereby weakening the inter-class separation effect. This might also explain why NTCE > NONL in ImageNet-1K of Table 1.

**Q3** (w.r.t **W3**). How is the LPM (or UFM) analysis conducted under the proposed loss? Neural Collapse was originally observed under conventional SCE, where theoretical analysis showed that NC properties hold when global optimality is achieved in LPM. When other losses (e.g., MSE loss) are used, theoretical verification was also conducted using this framework [R5]. Does the proposed loss satisfy these conditions? Moreover, if it is unclear whether NC actually occurs under the proposed loss, is it valid to empirically evaluate it using NC-based metrics?

**Q4** (w.r.t **W4**). As shown in Table 2, while CE exhibits lower values than the proposed method on other NC metrics, it demonstrates the opposite trend on *Information Theory Metrics*. Doesn’t this suggest that CE has no inherent issue in learning the information necessary for classification and is, in fact, superior to other methods in this regard? If so, is it truly necessary to enforce Neural Collapse in practical classification tasks?

**_reference_**

[R5] Han et al., Neural Collapse Under MSE Loss: Proximity to and Dynamics on the Central Path, ICLR 2022

---

> ### Author Response · Authors · 2025-11-22
>
> ### W3 – Theoretical analysis and NC guarantees (Q3)
>
> We agree that in the original submission we discussed three objectives (NormFace, NTCE, NONL) that empirically achieve better Neural Collapse but did not provide a formal guarantee that NC is a minimizer of these objectives.
>
> In the **revised manuscript**, we address this by introducing **Proposition 4.1**, where we show that under UFM:
>
> * **NormFace**, **NTCE**, and **NONL** all admit **NC configurations (NC1–NC3)** as **global minimizers** of their respective population losses.
>
> This result places our empirical findings on firmer theoretical ground: it shows that, at least under UFM, these losses are not just compatible with NC but are **actually minimized by NC geometry**.
>
> ---
>
> ### W1 – Novelty
>
> We first clarify our **overall novelty** by summarizing our **five contributions**, and then we specifically discuss how **one** of them relates to **[R2]**.
>
> Our contributions can be summarized as:
>
> 1. **Unified view for normalized softmax and SCL.**
>    We show that **normalized softmax and SCL are both prototype–contrast methods on the unit hypersphere**, differing only in whether prototypes are **explicit weights** or **implicit class means**. This unified view explains why both can induce NC, while standard CE does not.
>
> 2. **Two new NC-inducing supervised losses (NTCE, NONL).**
>    Guided by this view, we introduce **NTCE** (more in-batch negatives for the prototype classifier) and **NONL** (negatives-only normalization that decouples alignment and repulsion). These are **not existing losses in disguise**; they are new objectives that **accelerate NC convergence** and **improve NC metrics and accuracy** over CE and normalized softmax.
>
> 3. **SCL already learns an optimal classifier.**
>    We prove that **under NC, SCL’s class means form an ETF-aligned classifier**, so SCL **already learns an optimal linear classifier during pretraining**, effectively eliminating the need for a separate linear probing phase while matching its accuracy.
>
> 4. **NC at scale with practical savings.**
>    We validate on **four benchmarks including ImageNet-1K**, showing that NTCE and NONL achieve **high NC metrics and surpass CE accuracy**, and that the prototype classifier can **replace linear probing** while maintaining SCL’s performance, yielding **practical computational savings**.
>
> 5. **Practical downstream benefits of NC (added in the revision).**
>    In the revised manuscript we further show that the NC geometry induced by our losses yields **clear gains in**:
>
>    * **Transfer learning** (8 downstream tasks, **+5.5% mean relative improvement** over CE),
>    * **Class imbalance** (CIFAR-10/100-LT, up to **+8.7%** over CE and better than ETF+DR), and
>    * **Robustness** (lower mCE and higher clean accuracy on ImageNet-C).
>
>    To our knowledge, this is the first work to **systematically demonstrate NC at ImageNet scale and link it to such a broad range of downstream improvements**.
>
> **Relation to [R2].**
> We thank the reviewer for highlighting this connection. Both our **third contribution** and **[R2]** use class-mean prototypes as classifiers, but in **fundamentally different ways**:
>
> * In **[R2]**, which is designed for knowledge distillation, the **student** network uses an NC3-inspired classifier whose weights are the **class centroids computed from the teacher’s features**, while the teacher still uses its own trained linear classifier.
> * In **our work**, we show that under **supervised contrastive learning (SCL)** a **model can replace its own trained classifier with its own class-mean (NC) classifier** *without losing performance*.
>
> In the updated manuscript we now explicitly discuss **[R2]** in the Related Work section and clarify this distinction.
>
> ====>

---

> ### Author Response · Authors · 2025-11-22
>
> ### W4 – “Marginal” improvements and practical significance
>
> The initial manuscript mainly focused on showing that **NC geometry can be achieved in practice**, and therefore emphasized: (i) **consistent gains in in-distribution classification accuracy**, (ii) **improved NC metrics**, and (iii) **faster convergence to NC**.
>
> In the **revised manuscript** we significantly expand the empirical picture:
>
> * **Broader benefits: transfer, imbalance, robustness.**
>   As detailed above (Sec. 5.3 in the paper), we now show that the NC geometry induced by our losses not only consistently improves performance on the original training task, but also brings **clear benefits in transfer learning, long-tailed classification, and robustness**. These are not marginal: e.g., **+5.5% mean relative improvement in transfer**, **up to +8.7% in long-tailed CIFAR**, and **improved mCE and clean accuracy on ImageNet-C**.
>
> * **Scaling NONL properly on ImageNet-1K.**
>   In the original submission, NONL was evaluated with a **smaller batch size than what is appropriate for a contrastive-style loss**. In the appendix of the updated manuscript, we now include a **batch-size study for NONL on ImageNet-1K**, where we vary the batch size and report performance. With a larger (but still practical) batch, **NONL achieves a +1.1% absolute improvement over CE on ImageNet-1K**, closing the gap observed previously.
>
> We also note that, in the ImageNet literature, **absolute gains of 0.5–1.0%** are commonly regarded as **meaningful improvements**—many top-conference works are built around such margins, especially when obtained **without changing the architecture or data**. In our case, a **+1.1% gain on ImageNet-1K**, combined with clear downstream improvements in a range of tasks, indicates that the benefits of our NC-inducing objectives are practically significant and not merely cosmetic.
>
> ---
>
> ### W2 – Gradient analysis
>
> We thank the reviewer for pointing this out. NTCE differs from normalized CE only in the scope of its negative term: for a class (c) that appears in the batch, its prototype receives the usual attractive gradient toward its own features and a repulsive gradient away from other features in the batch, while classes that do not appear in the batch receive no update at that step.
>
> In other words, NTCE does not change **which** classes are updated or **how often**; it replaces **1 instance → (K) prototypes repulsion** with **1 prototype → (M) instances repulsion**, which we find empirically leads to faster NC convergence and better NC metrics (Figure 1, Table 2).
>
> Because the resulting gradients are a straightforward modification of normalized CE gradients, and our main theoretical support comes from Proposition 4.1 (proving NC as a global minimizer under UFM) together with clear empirical benefits, we believe a full trajectory-level gradient derivation would add substantial length without changing the core message.
>
> ---
>
> ### Q1 – Would higher batch sizes change the conclusions for NONL?
>
> Short answer: **yes, and we now show this explicitly.**
>
> In the original submission, NONL was evaluated with **relatively small batch sizes**, which is suboptimal for a contrastive-style loss. In the revised manuscript:
>
> * We add an **ablation in the appendix** where we increase the batch size for NONL on **ImageNet-1K** and report the resulting performance.
> * This study shows that **NONL benefits substantially from larger batches**, reaching **+1.1% over CE on ImageNet-1K** with an appropriate batch size.
>
> We explicitly acknowledge that **needing larger batches is a practical drawback**, but this is a **typical limitation of contrastive methods** (including SCL and many SSL methods). Our work highlights that **supervised normalized classifiers are already performing a kind of contrastive learning with class prototypes**, and that fully reaping the **generalization and robustness benefits** naturally points toward **larger effective negative sets**. Recent SSL methods (memory banks, queues, momentum encoders) suggest promising ways to increase the number of negatives without linearly scaling batch size; we now emphasize this as a natural direction for future extensions of NONL.
>
> ===>

---

> > ### Author Response · Authors · 2025-11-22
> >
> > ### Q4 – Information-theoretic metrics vs. necessity of NC
> >
> > We agree that Table 2 shows CE scoring better on some information-theoretic metrics. In fact, CE appears to capture **slightly more raw information** according to these measures. However, more information does **not** necessarily translate into better performance: our results show that CE tends to organize this information in a **less NC-like, less prototype-structured geometry** (worse effective ranks, weaker alignment), whereas our normalized losses **reshape the same information into a cleaner NC geometry** while matching or improving accuracy.
> >
> > The information-theoretic metrics we use primarily reflect **entropy / redundancy**, not “goodness of NC” or downstream usefulness in a one-dimensional way. Our work precisely targets this distinction: the key is **not how much information is present**, but **how that information is organized**. CE may retain more variability that is mainly useful for the *training task*, while our NC-inducing losses bias the representation toward **well-separated class prototypes** that are better suited for reuse.
> >
> > This is consistent with our empirical findings: despite CE looking slightly better on some information metrics, our methods **consistently win where it matters**:
> > – they converge to NC geometry much faster,
> > – they improve **transfer, long-tailed performance, and robustness** over CE (Tables 4–6), and
> > – with proper scaling, they also improve **ImageNet-1K top-1 accuracy**.
> >
> > Our evidence shows that for **standard supervised classification**, enforcing an **NC-like hyperspherical prototype geometry** is **not harmful to information content and is often beneficial**, because it organizes the information in a way that is more useful for **downstream performance and robustness**.
> >
> > In the updated manuscript we have included a dedicated paragraph on this point in Section 5.2.

---

> > > ### Author Response · Authors · 2025-11-26
> > > **Request for Feedback**
> > >
> > > Dear Reviewer 1A6n,
> > >
> > > As the discussion period is nearing its end and it has been five days since our initial response, we would appreciate your feedback on our responses and the revised manuscript so that we have time to address any further comments you may have.
> > >
> > > **Summary of revisions addressing your concerns:**
> > >
> > > | Concern | Response |
> > > |---------|----------|
> > > | W1 (Novelty) | Clarified five contributions; distinguished from [R2]; updated Related Work |
> > > | W2 (Gradient Analysis) | Discussed NTCE's prototype→instances gradient flow |
> > > | W3 (Theoretical Analysis) | **Added Proposition 4.1:** NC is a global minimizer for NormFace, NTCE, NONL under UFM |
> > > | W4 (Marginal Improvement) | **Added Tables 4–6, 8:** +5.5% transfer, +8.7% long-tail, lower mCE, +1.1% ImageNet-1K |
> > > | Q1 (Batch size) | Discussed; **added batch-size ablation in Table 8** |
> > > | Q2 (Gradient behavior) | Discussed NTCE's gradient properties |
> > > | Q3 (Theoretical model) | **Added Proposition 4.1** under UFM |
> > > | Q4 (Information metrics) | Discussed relationship between information metrics and NC geometry |
> > >
> > > We kindly ask:
> > > 1. Do you consider these additions to adequately address your identified weaknesses?
> > > 2. Does the revised submission meet your quality criteria for acceptance?
> > > 3. If not, what specific additional evidence would help us meet these criteria?
> > >
> > > Thank you for your time and constructive feedback.

---

> ### Comment · Reviewer_1A6n · 2025-11-27
>
> First of all, I sincerely appreciate the authors’ efforts. I am currently reviewing not only the authors’ responses to my comments but also the comments from the other reviewers, and I am preparing additional discussion accordingly. However, this process is taking some time, and I believe it would be helpful for the authors’ additional rebuttal if I share a few issues I have identified so far. Thus, I am providing only those points at this time.
>
> **1. Performance of the reproduced baseline (ETF+DR) in imbalanced learning**
>
> The performance of the ETF+DR baseline in the imbalanced learning experiments is considerably lower than the reported values in the reference paper. Could the reproduction have been performed incorrectly? While perfect alignment with the reported numbers is not strictly required, the reproduced performance should fall within a reasonably acceptable margin (about ±1%p) or be higher than reported values. However, the deviation is much larger than that, and moreover, in Table 5 (CIFAR-100-LT, τ = 0.1 and 0.02), the reproduced performance of ETF+DR is even lower than that of CE, which I assume was measured under the same experimental setting.
>
> Such uncertainty in the reproduction environment undermines the reliability of the empirical validation of the effectiveness of the proposed method. I would appreciate it if the authors could carefully examine this issue and provide detailed implementation descriptions of these experiments—not only whether the exact experimental setup of ETF+DR was followed, but also how the baseline methods were implemented—to clarify any concerns regarding reproduction.
>
> **2. Inappropriate baseline choices for transfer learning and OOD**
>
> ETF+DR is a method specifically designed for imbalanced learning, aiming to secure margins for minority classes using an NC-inspired fixed ETF classifier and a dot-regression loss that substitutes for the push-and-pull effect of softmax cross-entropy. Thus, although the method can technically be applied to other scenarios, its utility and effectiveness are expected to diminish outside imbalanced settings. For this reason, ETF+DR would not be an appropriate baseline for transfer learning or OOD evaluation. Therefore, it would be better for the authors to compare their method with more suitable baselines such as [C1] and [C2].
>
> Given the limited remaining discussion/rebuttal period, if reproducing or comparing these methods is difficult, it would be helpful for the authors to explain why their method should be considered superior to [C1] and [C2] or why such comparisons are unnecessary.
>
> **3. Low-quality representation in the additional proof of Proposition A.1**
>
> It appears that the authors conducted additional theoretical analysis using the LPM model at my request, and I appreciate these efforts. However, several missing statements and incorrect linkages make the argument difficult to follow. (For example, where are Lemma A.1 and Theorem A.1? Are these typos for Proposition A.1? Similar issues appear in Theorem A.2, A.3, etc., in the proof of Theorem A.1.)
>
> Furthermore, the proof seems to rely heavily on citing previous results for normalized CE loss and arguing that the proposed losses (NTCE, NONL) share similar structures and have the same minimizers. However, this reasoning is somewhat indirect, and even if correct, the current presentation is too naive.
>
> In my opinion, to clearly demonstrate that NC emerges under the proposed losses, the proof should explicitly show the following when reaching the global optimum: (i) last-layer features converge to their respective class means, (ii) these class means form a simplex ETF, and (iii) these class means align with their respective class weight vectors. (Refer to the proof sketch and theoretical framework used in ETF+DR and [C3].)
>
> Given the limited remaining discussion/rebuttal time, if it is difficult to provide a proof following the proof sketch I mentioned earlier, I encourage the authors to improve the representation quality of the proof, make the underlying logic clearer and more coherent, and clarify why the provided proof is sufficient.
>
> I apologize for the late response, and I will provide additional comments as soon as possible once I finish reviewing the other materials.
>
> ---
>
> [C1] Harun et al., Controlling Neural Collapse Enhances Out-of-Distribution Detection and Transfer Learning, ICML 2025
>
> [C2] Chen et al., Perfectly Balanced: Improving Transfer and Robustness of Supervised Contrastive Learning, ICML 2022
>
> [C3] Fang et al., Exploring deep neural networks via layer-peeled model: Minority collapse in imbalanced training, PNAS 2021

---

> > ### Author Response · Authors · 2025-11-28
> > **Response to Concern about ETF+DR Reproduction**
> >
> > We sincerely thank the reviewer for their thoughtful engagement and for providing additional feedback and we  greatly appreciate the constructive tone. Below, we first address the first point raised by the reviewer.
> >
> > First, based on the discussion below, we clarify that:
> > 1. **our reproduction is correct** and the apparent discrepancy stems from a critical difference in experimental settings: **reliance on heavy data augmentation (Mixup)** in the referenced paper.
> > 2. **heavy augmentation is out of our scope**. Our experiments intentionally exclude Mixup because one of our paper's core contributions is to show that **modifying the learning process through simple loss function modifications alone can help reach NC in practice and yield practical benefits, including improved performance under class imbalance**, without relying on heavy augmentation strategies or architectural changes.
> >
> > Below we demonstrate why our reproduction is correct, and that even when comparing our improvements (without Mixup) against Yang et al.'s improvements (with Mixup), **our methods remain superior**.
> >
> > ### 1\. Source of the Discrepancy: Heavy Augmentation (Mixup) in Yang et al.
> >
> > Most results reported in Yang et al. are produced using heavy data augmentation. **Yang et al. themselves show that Mixup accounts for accuracy improvements** on CIFAR-10-LT (see their Table 1):
> >
> > | Setting (CIFAR-10-LT, τ=0.02) | CE  | ETF+DR |
> > | --- | --- | --- |
> > | Without Mixup (Yang et al.) | 77.1 | 78.4 |
> > | With Mixup (Yang et al.) | 78.6 | 81.0 |
> >
> > ### 2\. Our Reproduction Matches Yang et al. in Equivalent Settings
> >
> > Comparing our results to Yang et al.'s **without-Mixup** setting:
> >
> > | CIFAR-10-LT | CE (Yang, w/o Mixup) | CE (Ours) | ETF+DR (Yang, w/o Mixup) | ETF+DR (Ours) |
> > | --- | --- | --- | --- | --- |
> > | τ = 0.1 | 87.4 | 88.1 | 86.9 | 88.0 |
> > | τ = 0.02 | 77.1 | 76.8 | 78.4 | 77.8 |
> > | τ = 0.01 | 71.0 | 70.2 | 73.0 | 71.3 |
> >
> > Our reproduced values fall within acceptable variance of the original paper's results under equivalent (no-Mixup) settings and preserve the same trend.
> >
> > ### 3\. Addressing the Specific Concern about ETF+DR < CE in Table 5
> >
> > The reviewer notes that in Table 5 at τ=0.1 and τ=0.02, our ETF+DR sometimes falls slightly below CE. This is actually **consistent with Yang et al.'s own findings** without Mixup (see their Table 1): at τ=0.1, Yang et al. report ETF+DR (86.9%) < CE (87.4%) without Mixup. The advantage of ETF+DR becomes more pronounced only at extreme imbalance ratios (τ=0.01) and when combined with Mixup.
> >
> > ### 4\. Relative Improvements: Our NONL vs. ETF+DR (with Mixup)
> >
> > We now compare the **relative improvement** over CE based on the reported results of Yang et al. **with Mixup**. Our NONL achieves superior gains using only loss modifications, without requiring Mixup:
> >
> > | Dataset | Method | Δ(Method − CE) @ τ=0.02 | Δ(Method − CE) @ τ=0.01 | Mixup Used |
> > | --- | --- | --- | --- | --- |
> > | CIFAR-10-LT | **NONL (Ours)** | **+3.7** | **+6.1** | **No** |
> > |     | ETF+DR (Yang et al.) | +2.4 | +3.7  | Yes |
> > | CIFAR-100-LT | **NONL (Ours)** | **+3.2** | **+2.6** | **No** |
> > |     | ETF+DR (Yang et al.) | +2.3 | +2.3 | Yes |
> >
> > This demonstrates that our NC-inducing objectives achieve stronger improvements through loss design alone.
> >
> > ### 5\. Balanced Dataset Results
> >
> > On balanced datasets, our reproduction matches the expected behavior:
> >
> > | Dataset | CE (Ours) | ETF+DR (Ours) |
> > | --- | --- | --- |
> > | CIFAR-10 | 94.6 | 94.4 |
> > | CIFAR-100 | 72.1 | 72.1 |
> > | ImageNet-100 | 84.4 | 84.5 |
> > | ImageNet-1K | 75.4 | 75.4 |
> >
> > &nbsp;
> >
> > Note that we provide classification results on ETF+DR on the challenging large scale ImageNet1k benchmark and mark same performance as Cross Entropy further indicating a proper implementation.
> >
> > ### 6\. Reproducibility Statement
> >
> > In the original submission we included a reproducibility statement for our results. For ETF+DR specifically, which was added during the rebuttal period, our implementation directly combines the cross entropy training script from the well followed (3.4K stars) SupContrast repository (github.com/HobbitLong/SupContrast) with the authors' ETF+DR implementation (github.com/NeuralCollapseApplications/ImbalancedLearning), using their default hyperparameters.
> >
> > * * *
> >
> > **Summary:** The discrepancy arises from **heavy augmentation settings (Mixup vs. no Mixup), not reproduction errors**. When compared in equivalent settings, our results align with Yang et al. within acceptable margins. Our methodological choice to exclude Mixup is intentional, as we aim to demonstrate that loss function design alone can give practical benefits without relying on heavy data augmentation or architectural changes. Crucially, our NONL achieves larger improvements over CE (+3.7, +6.1 pp on CIFAR-10-LT) *without* Mixup than ETF+DR achieves *with* Mixup (+2.4, +3.7 pp), demonstrating the strength of our approach. We will add a dedicated discussion in the appendix clarifying these settings and include the comparison table above.

---

> > > ### Author Response · Authors · 2025-12-02
> > > **Response to points 2 (Transfer Learning baseline) and 3 (proof of Proposition A.1)**
> > >
> > > ## **2\. Regarding baseline choices for transfer learning and OOD**
> > >
> > > We clarify that the **primary goal of our work is to improve standardized supervised learning pipelines**. Our aim is to show that **modifying the standard CE objective** can both (i) induce Neural Collapse and (ii) improve several downstream capabilities. To validate this, we evaluate across **classification**, **transfer learning**, **long-tailed classification**, and **robustness**, following the **standard evaluation protocol used in cornerstone works such as SupCon [1] and SimCLR [2]** plus we further report a wide range of **NC/optimization metrics**.
> > >
> > > For this reason, our **baseline selection criterion** focuses on methods that operate *within* the **standard supervised training pipeline**:
> > >
> > > - **Standard Cross-Entropy** — canonical baseline
> > >
> > > - **Normalized Cross-Entropy (NormFace)** — CE with normalization
> > >
> > > - **ETF+DR** — NC-inducing method within the same training scheme
> > >
> > >
> > > ETF+DR is **not** included as a transfer learning benchmark. Rather, it serves as a **Neural-Collapse-inducing reference method** that fits within **exactly the same training pipeline** as ours.
> > >
> > > By contrast, the reviewer-suggested baselines operate under **different training setups**:
> > >
> > > - **\[C1\]** makes an **architectural modification** (an additional projector), which induces different inductive biases.
> > >
> > > - **\[C2\]** uses **supervised contrastive learning**, which involves a substantially different training pipeline with **heavier augmentations** and **different invariances** in the final representation space.
> > >
> > >
> > > Since our contribution is specifically about **improving CE-based training** to result in a wide range of benefits, rather than proposing a new transfer-learning method, comparisons to specialized transfer-learning approaches fall outside our scope. We believe the **appropriate comparison is with other CE-based methods under identical training conditions**, which is precisely what we provide.
> > >
> > > * * *
> > >
> > > ## **3\. Regarding the proof of Proposition A.1 (now Theorem 4.1)**
> > >
> > > We sincerely thank the reviewer—this comment directly helped us improve the technical presentation. We have **substantially revised and expanded the proof (now 7 pages)** to address all concerns. For all three losses, the revised appendix now **explicitly proves at global optima**:
> > >
> > > **(i)** within-class collapse
> > > **(ii)** emergence of a simplex ETF
> > > **(iii)** classifier–mean alignment
> > >
> > > ### **1\. Clarifying missing statements and linkages**
> > >
> > > The confusion around “Lemma A.1”, “Theorem A.1”, etc. was due to our use of the overleaf template where `\Cref{}` incorrectly typeset propositions and lemmas as theorems. In the revised version, **all labels and cross-references are corrected**, and every statement is now **cleanly numbered** with consistent linkage.
> > >
> > > ### **2\. Why our proof leverages known results**
> > >
> > > Our objectives are **normalized modifications of CE-like losses**, with an explicit connection to **contrastive learning**. Since **strong existing NC results** already apply to normalized CE and contrastive losses in the UFM/LPM setting, we reduce our objectives to these known forms in order to **rigorously situate our losses within these established frameworks**.
> > >
> > > In the revised version, we now provide **precise algebraic reductions**, identify **exact equality conditions**, and elevate the result from a **proposition** to a **theorem supported by a detailed 7-page proof**.
> > >
> > > ### **3\. Updated complete proof structure**
> > >
> > > The updated appendix presents a **unified and rigorous**  3-step argument for NTCE and NONL:
> > >
> > > **Step 1 — Reduction to class means:**
> > > We introduce new lemmas that bound sample-level losses by **class-mean objectives** using Jensen’s inequality. We also state the **exact conditions for tightness**, specifying when equality is achieved.
> > >
> > > **Step 2 — Class-level objectives are contrastive losses:**
> > > We show that the resulting objectives are **standard contrastive formulations** whose global minimizers are known to yield **centered simplex ETFs** with **classifier–mean alignment**.
> > >
> > > **Step 3 — Lifting to the sample level:**
> > > Using the tightness conditions from Step 1, we prove that **any global minimizer must collapse each class to a single vector**, recovering full Neural Collapse at the sample level.
> > >
> > > For **NormFace**, we additionally prove an **exact equivalence** to the constrained UFM formulation in Yaras et al. (2022), allowing us to directly invoke their NC theorem.
> > >
> > > * * *
> > >
> > > ## **References**
> > >
> > > \[1\] Khosla et al., *Supervised Contrastive Learning*, NeurIPS 2020
> > > \[2\] Chen et al., *SimCLR*, A simple framework for contrastive learning of visual representations 2020

---

### Author Response · Authors · 2025-11-22

## Global Response to All Reviewers

We thank all reviewers for their thoughtful and constructive feedback, which has helped us **substantially strengthen our paper**. In the revision, we add new experiments and theory showing that our methods provide **practical benefits beyond standard classification accuracy and faster Neural Collapse (NC) convergence.**

The new results in the **updated manuscript** directly address the main concerns about the **practical significance of our proposed objectives**:

- **Transfer learning (Table 4):** +5.5% mean relative improvement over cross-entropy across **8 downstream tasks**.
- **Long-tailed classification (Table 5):** Up to +8.7% relative improvement on two datasets and three imbalance ratios each.
- **Robustness (Table 6):** Lower mean Corruption Error on corrupted ImageNet across 12 corruptions and 5 severity levels.
- **“Marginal Improvement” & NONL scalability (Table 8):** With appropriate batch size, NONL achieves **+1.1% absolute over CE** on ImageNet-1K.
- **New theoretical result (Theorem 4.1):** Under UFM, we prove that **NormFace, NTCE, and NONL all admit NC configurations as global minimizers**, placing our empirical NC observations on firmer theoretical footing.
- **Additional baseline:** We add a **fixed ETF classifier (ETF+DR)** as a strong NC-based baseline and show that our objectives **consistently outperforms it** across all evaluation axes.
- **Larger architectures (App., Table 9):** ResNet-101/152 experiments where our methods **improve over CE**.

* * *

## Acknowledged Contributions

Reviewers explicitly recognized **our main contributions** as well as the practicality our approach:

**1\. Conceptual unification of normalized CE and SCL**

> *“an elegant unifying view”* (HD19), *“cleanly shows… the same prototype-contrast story”* (zgJh), *“addresses an interesting problem”* (1A6n), *“well presented and helps understand these methods in a common theoretical view”* (2yVW)

**2\. Replacement for costly linear probing**

> *“replace linear probing with similar accuracy”* (zgJh), *“saving the hours of computation”* (HD19)

**3\. Theoretical result about prototype optimality of SCL**

> *“interesting equivalence”* (1A6n), *“preserving accuracy”* (HD19)

**4\. Fast convergence to Neural Collapse**

> *“reach NC geometry much faster”* (HD19), *“comprehensive convergence analysis”* (zgJh), *“metrics showing faster NC convergence and better class separation”* (2yVW)

**5\. Simplicity and practicality**

> *“drop-in and often improve accuracy”* (zgJh), *“clear and intuitive diagnosis”* (HD19), *“simple yet intuitive modified cross-entropy loss”* (1A6n), *“easy-to-implement extensions”* (2yVW), *“easy to implement and directly build on existing normalized contrastive losses”* (2yVW)

* * *

## Main Concerns and How We Address Them

Most remaining concerns focused on **our new objectives (NTCE, NONL)** and on the **practical motivation for enforcing NC geometry**.
In the **revised manuscript**, we address these points by adding Section 5.3, which demonstrates practical benefits of the collapsed representations that our objectives produce.

Concretely, our methods show:

- **Superior transfer learning** on 8 out-of-distribution datasets,
- **Better performance under class imbalance**, and
- **Improved robustness** to 12 common corruption types.

Together with the gains in standard classification performance (NTCE +1.3%, NONL +1.1% on ImageNet-1K) and the fact that our methods outperform the new NC-based baseline (ETF+DR) on all five aspects we tested (classification, convergence to NC, transfer learning, class imbalance, robustness), we conclude that:

1.  Our objectives **reliably achieve NC geometry** in practice, and
2.  This geometry **translates into improved accuracy and desirable downstream benefits** (transfer, imbalance, robustness).

* * *

## Additional Contribution from the Revision

Addressing the reviewers’ comments led to an additional, now explicit, contribution:

- **\[New (5th) Contribution\]**
    We empirically demonstrate that the representations learned by our objectives yield **practical benefits**, with improved performance on transfer learning (e.g., +5.5% mean relative improvement), long-tailed classification (up to +8.7% relative improvement), and robustness (lower mCE).

* * *

**Overall**, our manuscript promotes a more structured view of supervised learning: from unconstrained optimization in Euclidean space to **prototype-based classification on the hypersphere**. By making this geometry explicit, we help close the theory–practice gap, simplify training, accelerate convergence, and obtain models that provably realize their optimal NC structure. Our results indicate that this, in turn, yields tangible benefits such as better generalization and robustness, faster training, and the elimination of extra compute phases like linear probing.

In the rest of the rebuttal, we respond point-by-point to each reviewer.

---

### Author Response · Authors · 2025-12-02
**Discussion Summary to Area Chair**

Since the discussion period has ended, we summarize what reviewers raised, what was added in the revision, and how we addressed each point during the discussion. We introduced new theory (Theorem 4.1), expanded empirical evidence across transfer, imbalance, robustness, and scaling, clarified design choices, added a new baseline, and provided detailed reasoning for all reviewer questions. **We believe all concerns have been fully addressed** and the paper is now substantially stronger as a result of the discussion period.

* * *

## Key Revisions Made

1.  **New Theorem 4.1:** Under UFM, we prove that NormFace, NTCE, and NONL all admit NC configurations as global minimizers, with a complete 7-page proof explicitly demonstrating NC1, NC2, and NC3.

2.  **Transfer Learning (Table 4):** +5.5% mean relative improvement over cross-entropy across 8 downstream tasks.

3.  **Long-tailed Classification (Table 5):** Up to +8.7% relative improvement on CIFAR-10/100-LT across three imbalance ratios.

4.  **Robustness (Table 6):** Lower mean Corruption Error on ImageNet-C across 12 corruptions and 5 severity levels.

5.  **NONL Scalability (Table 8):** With appropriate batch size, NONL achieves +1.1% absolute over CE on ImageNet-1K.

6.  **Additional Baseline:** ETF+DR added as NC-based baseline; our objectives consistently outperform it across all evaluation axes.

7.  **Larger Architectures (Table 9):** ResNet-101/152 experiments where our methods improve over CE.


* * *

## Reviewer 1A6n

| Comment | Added in Revision | Discussed |
| --- | --- | --- |
| Novelty of discarding linear probing | Clarified contributions and differences from prior NC work | Argued that our unified prototype view and classifier interpretation are not present in prior NC3-based methods; distinguished from \[R2\] which uses teacher centroids for student |
| Missing NC theory | Added Theorem 4.1 with complete proof | Explained how the new proof explicitly demonstrates NC1, NC2, and NC3 under UFM |
| NTCE gradient behavior unclear |     | Provided reasoning of prototype→instance gradients |
| Practical gains marginal | Added transfer, long-tailed, robustness, NONL scaling, and large-model results | Explained that NC geometry yields gains in transferability and robustness even when top-1 changes are modest |
| ETF+DR discrepancy | Added Mixup vs. no-Mixup clarification and aligned protocol | Explained heavy augmentation as the source of discrepancy; showed our NONL achieves larger gains without Mixup than ETF+DR achieves with Mixup |
| Quality of new proof | Expanded to 7-page proof with explicit 3-step structure | Fixed cross-references; added explicit proofs for within-class collapse, simplex ETF emergence, and classifier-mean alignment |
| Inappropriate baselines for transfer (\[C1\], \[C2\]) |     | Clarified our scope is improving CE-based training, not proposing new transfer methods and we benchmark against 3 baselines across a variety of settings; ETF+DR serves as NC-inducing reference within same pipeline |
| CE better on info metrics |     | Argued that information metrics measure entropy, while NC geometry organizes features for better downstream utility |

---

> ### Author Response · Authors · 2025-12-02
>
> ## Reviewer HD19
>
> | Comment | Added in Revision | Discussed |
> | --- | --- | --- |
> | Practical value of NC unclear | Added transfer, imbalance, robustness experiments | Explained how NC structure (aligned means and low intra-class variance) improves generalization |
> | Radial-freedom motivation unclear |     | Argued that normalization removes radial degrees of freedom and NTCE/NONL refine angular separation through better optimization |
> | NONL scaling limits | Added batch-size ablation showing +1.1% on ImageNet-1K | Explained the need for adequate negatives and how NONL improves when this is satisfied |
> | Many negatives justification |     | Used InfoNCE estimator-bias argument: more negatives approximate the population objective better |
> | Novelty limited |     | Argued that unifying normalized CE and SCL into a prototype-contrast framework is both conceptual and practical contribution; we demonstrate NC benefits at ImageNet scale |
> | Task scope unclear |     | Clarified NC is appropriate for closed-set classification but not tasks needing intra-class diversity (detection, segmentation, multi-label) |
> | Missing decoupled NormFace ablation |     | Reported that decoupled NormFace is unstable without sufficient negatives; justified CE→NormFace→NTCE→NONL progression |
>
> * * *
>
> ## Reviewer zgJh
>
> | Comment | Added in Revision | Discussed |
> | --- | --- | --- |
> | Missing transfer/robustness | Added Tables 4–6 | Explained how prototype-contrast structure improves transferability and robustness |
> | Missing larger models | Added ResNet-101/152 results (Table 9) | Explained that NC behavior strengthens with model capacity |
> | Missing NC baseline | Added ETF+DR baseline | Explained how ETF+DR implements same mechanism as HPN/EBVs; our methods outperform across all axes |
> | Scalability claims misunderstood |     | Clarified we never claimed to reduce batch-size dependence; goal is to increase effective negatives from K prototypes to M samples |
> | Optimization refinements not genuine |     | Argued that optimization refinements constitute legitimate contributions when delivering measurable practical benefits across multiple evaluation axes |
>
> * * *
>
> ## Reviewer 2yVW
>
> | Comment | Added in Revision | Discussed |
> | --- | --- | --- |
> | Practical advantage unclear | Added transfer, imbalance, robustness experiments | Explained how NC geometry improves class separation and feature reusability |
> | Missing long-tailed evaluation | Added CIFAR-10/100-LT (Table 5) | Explained how NC reduces variance within minority classes and helps long-tail performance |
> | Semi/self-supervised extension |     | Discussed connection to SwAV; noted as promising future direction |

---

### Meta-Review · Area_Chair_fWsW · 2026-01-07

**Summary:**

The paper proposes a modified cross-entropy loss that leverages insights from Neural Collapse (NC) and Supervised Contrastive Learning (SCL) to improve feature learning on the unit hypersphere. By treating samples within a mini-batch as dynamic prototypes, the authors aim to accelerate the emergence of the Simplex Equiangular Tight Frame (ETF) structure and improve classification performance. However, concerns remain regarding the paper's technical novelty and the marginal performance gains observed on large-scale datasets such as ImageNet-1K. In addition, the conclusion regarding transferability appears to contradict previous work and requires further investigation.

**Reviewer Concerns:**

The authors addressed the lack of theoretical analysis by introducing Proposition 4.1 (presented as Theorem 4.1 in the manuscript), which proves the Neural Collapse optimality of normalized losses under the Unconstrained Features Model. However, as this analysis is fundamentally similar to existing proofs for NormFace and Supervised Contrastive Loss, the technical contribution of unifying these losses appears somewhat limited.

Another major concern involves the marginal improvements achieved by the proposed NONL loss on ImageNet-1K. To address this, the authors provided additional transfer learning experiments to demonstrate that collapsed representations enhance feature transferability. However, these findings appear to contradict previous observations (e.g., the work "Why Do Better Loss Functions Lead to Less Transferable Features?" and "Controlling Neural Collapse Enhances Out-of-Distribution Detection and Transfer Learning") which suggest that highly collapsed features in a pre-trained model can often degrade performance on downstream tasks. While this discrepancy could point toward an interesting new result, it currently lacks the thorough investigation to resolve these conflicting perspectives.

**Reviewer Scores:**

Reviewer zgJh indicated that their concerns were only partially addressed and elected to maintain their score of 4. Reviewer 2yVW similarly indicated they would maintain their score of 6. The remaining two reviewers, both currently at a 4, may adjust their scores in light of the new theoretical results and experiments; however, they may also choose to maintain their current scores, as concerns regarding the technical contribution and the marginal improvements on ImageNet-1K have not been fully resolved.

---

> ### Public Comment · ~Panagiotis_Koromilas1 · 2026-03-01
> **Factual Clarifications Regarding the Meta Review**
>
> We thank all reviewers and the area chair for their time and constructive engagement. To help future readers of our work, we provide this public comment to clarify several inaccuracies in the Meta Review regarding our work's relationship to the literature, empirical results, and theoretical contributions. The points below are verifiable from the cited references, the submitted manuscript, and the rebuttal discussion.
>
> **A. Transferability and robustness of NC representations**
> We respectfully highlight that the framing that "collapsed features degrade performance" **is misleading** in the context of **Neural Collapse (NC)**, as it does not account for the fact that NC entails both **within-class collapse** and **maximal inter-class separation**. Our transfer improvements arise **not from collapsed representations alone**, but from **collapsed representations with maximally separated classes**—the **joint geometric structure** that defines NC.
>
> ### Regarding connection to the literature and the cited references:
>
> 1.  **Collapsed within-class representations accompanied by maximal inter-class separation** are well-established to improve **generalization**, **transferability**, and **robustness**. This has been documented across numerous works: NC improves generalization **\[1, 2, 3\]**, adversarial robustness **\[4, 5\]**, and transfer learning **\[6, 7\]**. Furthermore, NC configurations converge toward **max-margin classifiers** **\[8\]** with stronger robustness guarantees **\[9\]**.
>
> 2.  **Harun et al. \[10\]  supports rather than contradicts our findings.** Harun et al. study how **intermediate layer representations** (not final layer) with varying degrees of NC affect transfer. Crucially, their proposed method explicitly enforces NC at the **classification layer** (the layer we operate on) while preserving encoder diversity via **entropy regularization**, achieving strong transfer. This is what we achieve through the **loss function alone**, without architectural modifications. Harun et al.'s core contribution validates that **NC at the classification layer is compatible with strong transfer**.
>
> 3.  **Kornblith et al. \[11\] studies a fundamentally different phenomenon.** Their objectives (**label smoothing**, **logit penalty**, **squared error**) reduce within-class variance without enforcing any principled inter-class structure. They show this **arbitrary compression** harms transfer. **Neural Collapse is categorically different:** it jointly minimizes within-class variance and maximizes between-class separation to the global optimum (**simplex ETF**). Furthermore, Kornblith et al. evaluate **penultimate layer features** from networks that do not achieve NC geometry; we evaluate transfer of **final-layer representations** that have **provably reached NC**. These are not comparable experimental settings.
>
>
> **In summary,** neither cited work contradicts our findings. Both actually **support** the view that **NC at the classification layer**—achieved through **principled geometric constraints** rather than **arbitrary compression**—is compatible with and can **enhance transfer learning**.
>
>
> **B. On "marginal improvements" on ImageNet-1K.**
>
> The Meta Review characterizes our gains as "marginal," based on pre-rebuttal ImageNet-1K results using suboptimal batch sizes for contrastive-style losses. As demonstrated in the revised Table 8, **NTCE achieves +1.3% and NONL +1.1% absolute improvement over CE on ImageNet-1K** with appropriate batch sizes. Beyond in-distribution accuracy, our methods yield **+5.5% mean relative improvement on 8 transfer tasks** (Table 4), **up to +8.7% relative improvement on long-tailed classification** (Table 5), and **lower mean Corruption Error on ImageNet-C** (Table 6). These gains are **consistent across every evaluation axis we tested**. This was acknowledged during the discussion period and reflected in the revised manuscript.
>
>
> =======>

---

> > ### Public Comment · ~Panagiotis_Koromilas1 · 2026-03-01
> >
> > **C.** **On the theoretical contribution.**
> >
> > Our paper contains two theorems, and **the meta review attributes our theoretical contribution to the wrong one — this is a significant misunderstanding of our work**. The meta review critiques Theorem 4.1 as being easily extractable from existing proofs, but **our actual theoretical contribution is Theorem 4.2**. Our main theoretical contribution, that supervised contrastive learning and normalized softmax are instances of the same prototype-contrast learning,  is a novel conceptual unification delivered by Theorem 4.2, which cannot be extracted from the literature. We are the first to establish this connection and introduce the prototype loss that unifies these objectives into a single family.
> >
> > To further clarify:
> >
> > **Theorem 4.2** (present in the original submission) proves that the global minimizers of supervised contrastive learning coincide with those of the prototype-softmax loss, establishing that SCL already learns an optimal classifier during contrastive training and eliminating the need for the standard, typically hours-long, linear probing phase. **This result does not exist in the literature.**
> >
> > **Theorem 4.1** provides formal NC optimality guarantees for our new objectives (NTCE, NONL) under the UFM setting. It was not in the original manuscript — we initially demonstrated NC optimality empirically and omitted the formal proof because the proof techniques for related losses are known. Reviewer 1A6n (W3) requested a formal analysis, and we provided a complete proof. **The meta review evaluates this supporting result as our main theoretical contribution — it is not.** That said, Theorem 4.1 proves NC optimality for objectives that did not exist before this paper. No prior work establishes this. Using known proof techniques to prove results about new objectives is standard practice.
> >
> > Given the above clarifications, this work contributes: a conceptual unification of supervised contrastive and normalized softmax learning as prototype-contrast methods on the hypersphere, two new objectives that reliably achieve Neural Collapse in practice, and consistent empirical gains across classification, transfer, long-tailed learning, and robustness.
> >
> > &nbsp;
> >
> > ## References
> > \[1\] Papyan, V., Han, X. Y., and Donoho, D. L. *Prevalence of neural collapse during the terminal phase of deep learning training.* Proceedings of the National Academy of Sciences, 117(40):24652–24663, 2020.
> > \[2\] Bartlett, P. L., Foster, D. J., and Telgarsky, M. *Spectrally-normalized margin bounds for neural networks.* In Advances in Neural Information Processing Systems (NeurIPS), pp. 6241–6250, 2017.
> > \[3\] Neyshabur, B., Bhojanapalli, S., and Srebro, N. *A PAC-Bayesian approach to spectrally-normalized margin bounds for neural networks.* In International Conference on Learning Representations (ICLR), 2018.
> > \[4\] Fawzi, A., Moosavi-Dezfooli, S.-M., and Frossard, P. *Robustness of classifiers: from adversarial to random noise.* In Advances in Neural Information Processing Systems (NeurIPS), 2016.
> > \[5\] Ding, G. W., Sharma, Y., Lui, K. Y. C., and Huang, R. *MMA training: Direct input space margin maximization through adversarial training.* In International Conference on Learning Representations (ICLR), 2020.
> > \[6\] Galanti, T., György, A., & Hutter, M. (2021, October). *On the Role of Neural Collapse in Transfer Learning.* In International Conference on Learning Representations.
> > \[7\] Khosla, P., Teterwak, P., Wang, C., Sarna, A., Tian, Y., Isola, P., Maschinot, A., Liu, C., and Krishnan, D. *Supervised contrastive learning.* In Advances in Neural Information Processing Systems (NeurIPS), 2020.
> > \[8\] Soudry, D., Hoffer, E., Nacson, M. S., Gunasekar, S., and Srebro, N. *The implicit bias of gradient descent on separable data.* Journal of Machine Learning Research, 19(70):1–57, 2018.
> > \[9\] Hein, M. and Andriushchenko, M. *Formal guarantees on the robustness of a classifier against adversarial manipulation.* In Advances in Neural Information Processing Systems (NeurIPS), 2017.
> > \[10\] Harun, M. Y., Gallardo, J., & Kanan, C. Controlling Neural Collapse Enhances Out-of-Distribution Detection and Transfer Learning. In *Forty-second International Conference on Machine Learning*.
> > \[11\] Kornblith, S., Chen, T., Lee, H., & Norouzi, M. (2021). Why do better loss functions lead to less transferable features?. *Advances in Neural Information Processing Systems*, *34*, 28648-28662.

---

### Decision · Program_Chairs · 2026-01-26

Reject